# Evaluating Human-Language Model Interaction

**Mina Lee    Megha Srivastava    Amelia Hardy    John Thickstun    Esin Durmus**
**Ashwin Paranjape    Ines Gerard-Ursin[§]    Xiang Lisa Li    Faisal Ladhak**
**Frieda Rong    Rose E. Wang    Minae Kwon    Joon Sung Park    Hancheng Cao**
**Tony Lee    Rishi Bommasani    Michael Bernstein    Percy Liang**

**Stanford University    [§]Imperial College London**

**Reviewed on OpenReview:** `https://openreview.net/forum?id=hjDYJUn9l1`

## Abstract

Many real-world applications of language models (LMs), such as writing assistance and code autocomplete, involve human-LM *interaction*. However, most benchmarks are *non-interactive* in that a model produces output without human involvement. To evaluate human-LM interaction, we develop a new framework, Human-AI Language-based Interaction Evaluation (HALIE), that defines the components of interactive systems and dimensions to consider when designing evaluation metrics. Compared to standard, non-interactive evaluation, HALIE captures (i) the interactive process, not only the final output; (ii) the first-person subjective experience, not just a third-party assessment; and (iii) notions of preference beyond quality (e.g., enjoyment and ownership). We then design five tasks to cover different forms of interaction: social dialogue, question answering, crossword puzzles, summarization, and metaphor generation. With four state-of-the-art LMs (three variants of OpenAI's GPT-3 and AI21 Labs' Jurassic-1), we find that better non-interactive performance does not always translate to better human-LM interaction. In particular, we highlight three cases where the results from non-interactive and interactive metrics diverge and underscore the importance of human-LM interaction for LM evaluation.

## 1 Introduction

Language models (LMs) have rapidly advanced, demonstrating unprecedented capabilities for generation and striking generality for tackling a wide range of tasks (Brown et al., 2020; Rae et al., 2021; Chowdhery et al., 2022). However, these models are at present primarily evaluated *non-interactively*: given an input text, a model generates a completion with the focus solely on the quality of the completion.[1] Almost all benchmarks, even those with a diverse range of tasks, such as GEM (Gehrmann et al., 2021) and HELM (Liang et al., 2022), encode this non-interactive view.

Our goal is to evaluate human-LM interaction instead of just model completions. LMs are already deployed in *interactive* settings, where humans work with them to brainstorm ideas (e.g., Jasper, Copy.ai), paraphrase sentences (e.g., Wordtune, QuillBot), reformulate queries (Nogueira & Cho, 2017), autocomplete sentences (Chen et al., 2019), write code (e.g., CoPilot, TabNine), and so forth. We anticipate that the adoption rate will continue to accelerate as LMs become more capable and novel use cases are discovered (Bommasani et al., 2021, §2.5).[2] But as a community, if we explicitly or implicitly optimize for non-interactive performance, will this improve interactive performance?

---

[1]There are specific tasks such as dialogue that are inherently interactive (Paranjape et al., 2020; Thoppilan et al., 2022; Shuster et al., 2022; OpenAI, 2022). We are interested in building a unified evaluation framework for human-LM interaction (where dialogue is a subset) that is inspired by, but also extends beyond dialogue systems.

[2]For instance, over 300 applications were built within a year of the release of GPT-3 (Brown et al., 2020), and ChatGPT amassed 1 million users in five days after launching (New York Times, 2022).

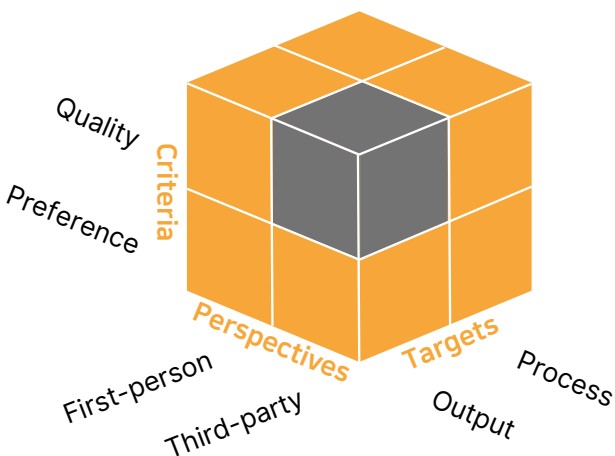

Figure 1: **Dimensions in human-LM interaction evaluation.** We propose a framework, HALIE, that expands on non-interactive evaluation along three dimensions: (i) we capture the full *process* in addition to the final *output* (targets); (ii) we capture the *first-person* subjective experience of users interacting with the LM in addition to the perspective of a *third-party* (perspectives), and (iii) we consider notions of *preference* beyond *quality* (criteria). These dimensions interact to define the full space of evaluation metrics (i.e., a 2x2x2 cube 🧊 ), which goes beyond standard, non-interactive evaluation (i.e., gray cell).

We develop an evaluation framework, **H**uman-**AI** **L**anguage-based **I**nteraction **E**valuation (HALIE) that defines the components of human-LM interactive systems and evaluation metrics, putting interaction at the center of LM evaluation. Concretely, we first make explicit that end users interact with a *system* (as opposed to a raw LM), which consists of an LM, a user interface (UI), and a system logic which constructs prompts and invokes the LM. The distinction between an LM and a system is important because, unlike non-interactive evaluation, interactive evaluation is fundamentally a function of the user and the system, which acts like the glue between the user and the LM. HALIE formally defines the system logic with states and actions. A *state* of the system is textual contents in the UI (e.g., dialogue history and the current text in a user input box), and an *action* is taken by a user (e.g., clicking the "send" button or typing into a text box) through the UI, which subsequently triggers the system to update the state. We define an *interaction trace* to be the resulting sequence of state-action pairs. Note that the exact choice of prompts (Wei et al., 2022; Khattab et al., 2023) and design of UIs (Clark et al., 2018; Buschek et al., 2021; Ippolito et al., 2022) can have significant impact on human-LM interaction; we follow the most standard practice whenever possible in our paper and leave the exploration to future work.

To evaluate human-LM interaction, HALIE defines metrics on interaction traces, which are organized along the following three dimensions (Figure 1)—targets, perspectives, and criteria: (1) *targets* include more than just the final output, and cover the entire interaction process (e.g., user queries and edits); (2) *perspectives* are not limited to third parties, but the users who interact with LMs to capture first-person experiences; and (3) *criteria* include not only quality (e.g., accuracy), but also preference (i.e., attitudinal measures of humans such as enjoyment). In contrast, non-interactive benchmarking devises automatic metrics or third-party human annotations (perspectives) to solely evaluate the quality (criteria) of outputs (targets).

We design five tasks ranging from goal-oriented to open-ended (Figure 2) to capture a variety of different interactions, and construct an interactive system for each task by defining states and actions. With these systems, We evaluated four state-of-the-art LMs: three variants of OpenAI's GPT-3 (Brown et al., 2020; Ouyang et al., 2022)—TextDavinci (text-davinci-001), TextBabbage (text-babbage-001), and Davinci (davinci)—and AI21 Labs' Jurassic-1 (Lieber et al., 2021)—Jumbo (j1-jumbo). We choose these models to study the impact of model size (TextDavinci vs. TextBabbage), instruction tuning (TextDavinci vs. Davinci), and implementation details regarding training data, architecture, and other factors (Davinci vs. Jumbo). While the impact of these differences on non-interactive benchmark performance has been studied

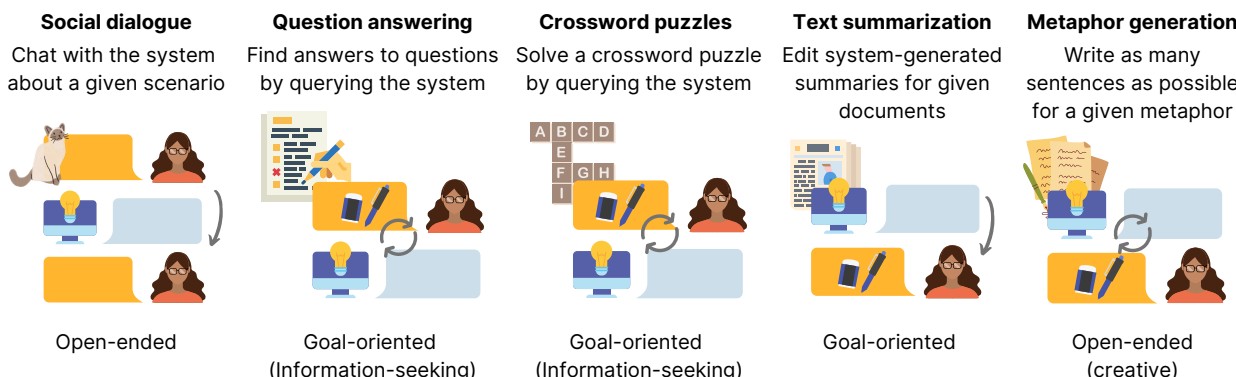

Figure 2: **Five tasks and human-LM interaction in the context of accomplishing the tasks.** We study tasks ranging from goal-oriented to open-ended and cover the common usages of LMs reported by Ouyang et al. (2022). When designing human-LM interaction for these tasks, we make use of various interaction styles, such as strict turn-taking (e.g., social dialogue) and iterative query reformulation (e.g., question answering). See Section 3 for details and screenshots of the systems we built for these tasks.

quite extensively (Brown et al., 2020; Ouyang et al., 2022; Liang et al., 2022), we want to study their effect in interactive settings.

From the 1015 interaction traces we collected (in English), we observe that better non-interactive performance does not always lead to better human-LM interaction; even the worst-performing model in non-interactive settings can perform as well as or better than other models in interactive settings. In Section 3, we present the results from five tasks and highlight the cases where non-interactive and interactive metrics diverge along the three dimensions. We release our interaction traces, their replay links, and system interfaces at https://github.com/stanford-crfm/halie. We summarize our contributions as follows:

- We propose an evaluation framework, HALIE, to structure and explore the multidimensional space of human-LM interaction.

- We instantiate our framework in five tasks. By studying various tasks and bringing them under one roof, we showcase the adaptability and versatility of our framework.

- The results obtained through the framework challenge conventional, non-interactive evaluation metrics and highlight the need for interactive evaluation.

- We provide guidelines for applying the framework to new scenarios and publicly release all data.

## 2 Framework

In this section, we introduce our framework, HALIE, for evaluating human-LM interaction. Concretely, we describe tasks and system construction for studying human-LM interaction, as well as interaction traces that we use to represent the interaction process. Lastly, we propose dimensions and metrics for evaluating human-LM interaction.

### 2.1 Solving Tasks Interactively

We study human-LM interaction in the context of *tasks*, which provide a structured context for facilitating and evaluating these interactions. For a task to be *interactive*, the interaction between a user and an LM needs to be specified. This involves detailing the space of possible input and output for the user and LM, as well as any intermediate steps or back-and-forth exchanges that constitute the task.

Figure 2 shows five tasks we study in this work: social dialogue, question answering, crossword puzzles, text summarization, and metaphor generation (see Section 3 for a detailed description of each task). When

---

**Algorithm 1:** Generate an interaction trace

---

$s_0 \leftarrow$ task-specific contents
**for** $t = 1, 2, ...$ **do**
    User takes an action $a_t$
    **if** $a_t$ finishes the interaction **then**
        **break**
    **else**
        System updates the state $s_{t+1} \leftarrow$ `Transition`$(s_t, a_t)$
**end**
**return** $[(s_1, a_1), (s_2, a_2), ...]$

**Function** `Transition`$(s_t, a_t)$:
    **if** $a_t$ requires querying an LM **then**
        $p = $ `CreatePrompt` $(s_t)$
        Fetch completions $c = $ `QueryLM`$(p)$
        **return** `ShowCompletions` $(s_t, c)$
    **else if** $a_t$ fills out a survey **then**
        **return** $s_t$ with results of survey
    **else**
        `// (e.g.,` $a_t$ `edits user input)`
        **return** $s_t$ updated with $a_t$
    **end**

---

designing the tasks, we aim to strike a balance between incorporating the elements that exist in real-world usage and making them concrete to conduct large-scale user studies. To incorporate real-world usage, we base our selection on the common query types from Ouyang et al. (2022, Table 1). These query types are abstractions of actual tasks (e.g., generation, open question answering, brainstorming, and chat), which we then concretize into the well-specified tasks for our experiments. Furthermore, we choose the tasks that span the spectrum from goal-oriented (e.g., question answering) to open-ended tasks (e.g., metaphor generation), to help us better understand human-LM interaction in various contexts.

## 2.2 Constructing an Interactive System

For each task, we build a *system* that facilitates human-LM interaction in the context of the task. In this work, we underscore the distinction between an LM and a system: users interact with a *system* (as opposed to a raw LM), which consists of an LM, a user interface (UI), and a system logic which constructs prompts and invokes the LM.

**Language model.** We consider an LM to be a black box that takes a text *prompt* and decoding parameters as input and stochastically returns a set of text *completions* (`QueryLM` in Algorithm 1).

**User interface.** In this work, we follow the most standard design and practice for UIs and focus on the performance gap between different LMs given the same interface. The exploration of UI design for human-LM interaction (Clark et al., 2018; Buschek et al., 2021; Ippolito et al., 2022) is another dimension of benchmarking that we leave to future work.

**System logic.** Given an LM, we define a *system logic* to be a set of states, potential user actions, and a transition function that specifies how states get updated, borrowing the terminology and intuitions from Markov decision processes.[3] A *state* $s_t$ captures everything the system needs to know about the current interaction at time $t$. This is primarily the visual state—the text contents of each text field in the graphical

---

[3] Note that in standard reinforcement learning (RL), the goal is to optimize a policy (e.g., the way a user interacts with an LM), whereas we are evaluating naturally occurring patterns in a policy (e.g., the way a user interacts with and accommodates

| Dimensions | | | Tasks | | | | |
| --- | --- | --- | --- | --- | --- | --- | --- |
| | | | Social dialogue | Question answering | Crossword puzzles | Text summarization | Metaphor generation |
| Targets | Perspectives | Criteria | | | | | |
| Process | First-person | Preference | Reuse | Ease | Enjoyment | | Enjoyment |
| Process | First-person | Quality | | Helpfulness | Helpfulness | Improvement | Helpfulness |
| Process | Third-party | Preference | | | Queries | | |
| Process | Third-party | Quality | | Queries | | Edit distance | Queries |
| Output | First-person | Preference | Interestingness | | | | Satisfaction |
| Output | First-person | Quality | Specificity | Fluency | Fluency | Consistency | Helpfulness |
| Output | Third-party | Preference | | | | | Interestingness |
| Output | Third-party | Quality | | Accuracy | Accuracy | Consistency | Aptness |

Table 1: We define a set of metrics for evaluating human-LM interaction across 5 tasks (see Table 21 for the full list); each metric can be characterized along three dimensions (targets, perspectives, and criteria). Note that some metrics, such as the number of *queries* from users, can be viewed as proxies for different quality (e.g., efficiency) or preference (e.g., enjoyment) metrics depending on the task.

interface (e.g., dialogue history and the current text in user input), but also includes any hidden dependencies, such as the history of previous dialogues. Given a state, a user can take one of a set of *actions*. For example, the user may type to modify user input or click a button.

Given a state $s_t$ and an action $a_t$, the system produces the updated state $s_{t+1}$ through a *transition function* (`Transition` in Algorithm 1). When the action is typing to modify user input, the update is a straightforward rendering of textual changes. When an action requires querying an LM, the system constructs a prompt (`CreatePrompt` in Algorithm 1), queries the underlying LM with the prompt (`QueryLM` in Algorithm 1), and updates the state with the completions from the LM (`ShowCompletions` in Algorithm 1). For example, in a dialogue system (Figure 3), a user may write "Thanks!" in user input and click the "send" button, which will trigger the system to construct a prompt (e.g., concatenating current dialogue history and user utterance), fetch a completion from the LM, and show it to the user. At the end of interaction, we get a sequence of state-action pairs, similar to Lee et al. (2022), which we call *interaction trace*. Algorithm 1 shows the process of generating an interaction trace as a result of human-LM interaction.

Note that there are many important design considerations regarding the system logic that can directly affect the dynamics of human-LM interaction. First, `CreatePrompt` determines how much control users have over prompts. For example, `CreatePrompt` can simply take the content of user input, or enforce a pre-defined prompt regardless of the user input. It can also decide how much each query depends on past interactions, by providing previous interactions as part of the prompt. Second, decoding parameters in `QueryLM` influence the nature of completions (Holtzman et al., 2020). For instance, one of the decoding parameters, `temperature`, controls the tradeoff between diversity and quality (Hashimoto et al., 2019), which can change the dynamics of human-LM interaction (Lee et al., 2022). Third, `ShowCompletions` controls how model completions are shown to users. For instance, how many completions are shown at a time, and how intrusive are they? These decisions can have a trade-off between efficiency and ideation (Buschek et al., 2021), or lead to distraction or cognitive load (Clark et al., 2018).

## 2.3 Evaluating Human-LM Interaction

To evaluate interaction traces, we expand on non-interactive evaluation along three dimensions and define metrics characterized by those dimensions.

**Dimensions.** Figure 1 shows three dimensions in human-LM interaction evaluation. The first dimension is *targets* (i.e., what to evaluate). Standard evaluation only considers the final *output* as a target, which is often simply a model completion $c$, but can be a result of more complicated processes (e.g., sample multiple

---

to an LM over time) rather than trying to optimize it. Also, in standard RL, there is a single reward function that matters. In our framework, we evaluate a policy in a multi-dimensional way accounting for different targets, perspectives, and criteria.

completions and use heuristics to construct the final output). In contrast, we consider the entire interaction *process*, which is represented as an interaction trace $[(s_1, a_1), (s_2, a_2), ...]$ in HALIE.

The second dimension is *perspectives* (i.e., whose perspective is reflected). Non-interactive evaluation solely depends on a *third-party* perspective; automatic metrics can be computed on an interaction trace without human intervention; likewise, human evaluation is done by third parties who did not interact with LMs in any way. In the interactive setting, however, evaluation should reflect the *first-person* perspective of the user who actually takes actions $a_t$ to interact with an LM through a system.

The last dimension is *criteria* (i.e., how to evaluate). Standard evaluation focuses on *quality*, which tend to be objective and include metrics like accuracy and fluency (defined for outputs) or helpfulness and edit distance (defined for processes). On the other hand, interactive evaluation adds attitudinal measures of a human, which we define as *preference*. These measures tend to be subjective and often captured by metrics like enjoyment and satisfaction. Note that preference criteria apply to both first-person and third-party perspectives. One example of preference metrics with third-party perspectives is creative story writing: some readers may not like certain kinds of stories (low preference), although they think that they are good (high quality). Also, sometimes metrics can reflect both preference and quality when human preference is aligned with the quality metrics of interest.

Note that each dimension has been extensively studied in subareas within NLP (e.g., dialogue systems). In HALIE, we present the three dimensions together as a set of orthogonal dimensions to consider and argue that it facilitates a holistic design of evaluation metrics for interactive systems.

**Metrics.** For each task, we define a set of metrics that cover the space defined by the three dimensions. Some metrics are a function of interaction traces (e.g., number of queries users made), whereas some metrics only depend on outputs (e.g., accuracy in solving a crossword puzzle). Some metrics are defined based on survey responses from users (e.g., satisfaction over the final output) and third-party evaluators (e.g., consistency of a summary given a document).

Table 1 shows the mapping from the space to the metrics in the five tasks and example metrics (see Table 21 for the full list of metrics). Note that non-interactive evaluation only covers one combination of the dimensions (i.e., outputs for targets, third-party for perspectives, and quality for criteria) corresponding to one gray cell in Figure 1. In contrast, our evaluation considers all combinations, corresponding to 🧊 in Figure 1.

### 2.4 Guidelines

Our framework is designed to be a reference for researchers when designing the evaluation of human-LM interaction for a new scenario. In particular, we believe that the framework can help researchers consider various components and dimensions of human-LM interaction that are easy to overlook (e.g., design considerations associated with `CreatePrompt`, `QueryLM`, and `ShowCompletions` in the transition function, as described in Section 2.2). Here, we provide the guideline for applying the framework to a new task and discuss when the framework may *not* be applicable.

#### 2.4.1 How to apply the framework to a new task

Instantiating the HALIE framework involves defining an interactive task, constructing an interactive system, and designing a set of metrics for a new scenario of interest. As a running example, we will use the case study on interactive story writing in Clark et al. (2018) to describe how one might apply HALIE to evaluate human-LM interaction in this particular setting. Note that this is a hypothetical scenario and we only highlight parts of the work (but not all) for brevity.

**Defining an interactive task.** To define an *interactive* task, it is essential to communicate the nature of the interaction between a user and an LM that is being studied. This involves detailing the space of possible input and output for the user and LM, as well as any intermediate steps or back-and-forth exchanges that constitute the task.

For interactive story generation, the researchers decide to consider a setting where an LM writes every other sentence in a story. Concretely, they define the task to be writing a ten-sentence long story by taking turns between a user and an LM, writing sentence by sentence in a linear fashion. In other words, every other sentence is first generated by an LM and optionally edited by the user. This turn-taking continues until a story reaches ten sentences. In this task, the space for user output and model output is restricted to an English sentence. The intermediate steps are defined as strict turn-taking. It is worth noting that there are limited back-and-forth exchanges in this task, as the user cannot go back and edit sentences from previous turns once they are submitted.

If a task is relatively new and it is unknown how users might interact with LMs, we recommend running small pilots to gain insights into user goals and preferences before defining a task. In Clark et al. (2018), the researchers restrict the output space to be sentence-level based on the preliminary study.

**Constructing an interactive system.** Once a task is defined, we need to define states, actions, and transition functions for the task and build an interactive system based on them. By specifying these components, researchers can easily communicate how their interactive systems operate not just on the surface (i.e., frontend), but also behind the scenes (i.e., backend; even with demos, it is often hard to communicate this aspect without a formal specification).

In the case of interactive story generation, suppose the researchers decide to show sentences from previous turns (story history) and provide an input box for a user to write and edit a sentence (user input) in the user interface. These elements constitute a state of the system (e.g., {story history, user input}). Then, the researchers specify available actions, such as clicking the "Add Line to Story" button and the "Submit Story" button (see Figure 3, 4, 7, 11, and 13 for more examples of states and actions). With states and actions, they define transition functions, which determine how each (state, action) pair connects to the next state (e.g., when a user clicks the "Add Line to Story" button, the system adds the current sentence to the story, updating the story history, and empties the user input box for the next sentence).

**Designing evaluation metrics.** When designing metrics, we encourage researchers to first consider all combinations of the three dimensions presented in the framework (targets, perspectives, and criteria), select combinations that are relevant to the task, and then detail how each combination can be measured in their systems, while taking into account the unique characteristics of the task and corresponding interaction.

For interactive story writing, the researchers come up with an idea to measure the usefulness of a model output, when thinking about a metric for process (target), first-person (perspective), and preference (criteria). Concretely, they decide to measure the usefulness by asking users while conducting open-ended interviews (an alternative for crowdsourcing can be a Likert scale next to each turn for a model output). To account for the unique characteristics of the task, they decide to see the relationship between the usefulness of a model output and the location of the sentence within a story where the model output is presented (e.g., in the 2nd turn vs. 10th turn).

### 2.4.2 When *not* to apply the framework

We note that our framework is designed to support the evaluation of human-LM interaction, where we assume the presence of a human user, an LM, interaction between the user and LM, and an evaluation goal. Therefore, the framework may not be applicable if an LM is evaluated by another LM (as a proxy for human judgment), and not a user. Similarly, in a scenario where there is a user, but the user has no way to interact with models (e.g., a user merely reads model outputs and evaluates them, as opposed to influencing model generation by providing inputs or modifying model outputs), we consider it as human evaluation, rather than human-LM interaction, and therefore consider it out of scope. Lastly, HALIE is designed for scenarios where a single user interacts with an LM. Therefore, when there are multiple users and/or LMs, it may require extending the framework.

# 3 Experiments

In this section, we first present key findings from our experiments (Section 3.1), and then describe details of the experiments for five tasks (Section 3.2—3.6). Our experiment design was reviewed and approved by the institution's Institutional Review Board (IRB).

## 3.1 Summary of Findings

### 3.1.1 Targets: Output vs. Process

One of the research questions we want to answer is the following (**RQ 1**): If we optimize for non-interactive performance (i.e., output), will this improve interactive performance (i.e., process)? In other words, if we select a model that performs the best on a leaderboard, can we assume this model will result in the best interactive performance? The answer based on our observation is "not always."

Consider a highly goal-oriented task like question answering (QA; Section 3.3) where non-interactive model performance, such as accuracy, is likely to determine the interactive experience and overall human-LM performance. Even for this task, we observe that the worst-performing model in non-interactive settings can perform as well as or better than other models in interactive settings. Specifically, `TextBabbage` had the worst accuracy as a zero-shot QA system, but achieved the best performance as an interactive LM assistant for the Nutrition category (Figure 5). For the US Foreign Policy category, `Jumbo` performed worst as a zero-shot QA system and `Davinci` performed best, but as an LM assistant, `Jumbo` outperformed `Davinci` (Figure 5). These counter-intuitive results, although in the minority, suggest that if we select models using only non-interactive performance, we may not achieve the best interactive performance.

For a more open-ended, creative task like metaphor generation (Section 3.6), the notion of "best" performance is more nuanced. One common approach to approximate the usefulness of models is to measure user "effort" given model *outputs*, measured by edit distance between the original model output and the final output edited by a human user (Roemmele & Gordon, 2018; Clark & Smith, 2021). Intuitively, if users find the model output to be helpful for writing, they would keep the generated text. Under this metric, `TextDavinci` performed the best (i.e., generated the outputs that required the fewest edits overall as shown in Table 8), closely followed by `Davinci` in our experiments. Turning to metrics based on the *process*, however, paints a slightly different picture. We approximated user effort with the total time spent in writing a metaphorical sentence and the number of queries users made per sentence. Under these two metrics, `Davinci` required the least effort from users (Table 8). Therefore, while one might expect that specific model performance measured based on the output would directly translate to that measured based on the process, we showed that this might not be the case.

### 3.1.2 Perspectives: Third-party vs. First-person

When designing a system, we care about how users experience the system, such as how useful it is to them, as opposed to the usefulness defined and judged by others. This leads to our next research question (**RQ 2**): Do the results of first-person evaluations differ from those based on the metrics defined by third parties? If so, what are the differences, and why do they occur?

In text summarization (Section 3.5), we observe a discrepancy between the model performance judged by human annotators (i.e., third-party) and by the users who directly interacted with the models (i.e., first-person). Another notable case observed in crossword puzzles (Section 3.4) was that users sometimes perceived models to be more helpful than they actually are. We measured the perceived helpfulness of a model based on a post-survey question and found that users significantly preferred `TextDavinci` over other models (Figure 8). However, assistance from `TextDavinci` led to statistically significantly higher crossword letter accuracy only for one of the five crossword puzzles; with another puzzle, users actually performed *worse* with both instruction-tuned models (`TextDavinci` and `TextBabbage`). One hypothesis for this behavior is that these models can provide confident and fluent misinformation, which can be incorrectly perceived as helpful.

### 3.1.3 Criteria: Quality vs. Preference

When assessing the performance of a model, it is important to differentiate between *quality* and *preference*; a model may perform better than another according to a specific metric (e.g., fluency), but users may care less about the metric and prefer the "inferior" model (e.g., because it is more creative). This leads us to ask the last main research question (**RQ 3**): Does the perceived quality of a model correlate with user preference? Specifically, given the current benchmarking practice of using a set of widely accepted metrics (e.g., fluency, coherence, and engagement for dialogue systems (Thoppilan et al., 2022; Smith et al., 2022)), what are the potential downsides of comparing models based on their performance aggregated over many metrics? Do users prefer models that perform better according to the majority of the metrics?

In dialogue (Section 3.2), users evaluated `TextDavinci` to be the best LM according to most metrics, including fluency, sensibleness, humanness, interestingness, and reuse (Table 2). However, users expressed similar *inclination* to continue interacting with `Davinci` which achieved the best specificity (Table 2). Given that `Davinci` was perceived to be worse than `TextDavinci` on all metrics except *specificity* and *inclination*, we hypothesize that users expressed an inclination to interact with `Davinci` because its responses were the most specific to what users had said. From this, we underscore the importance of understanding user needs and preferences and reflecting them for model selection.

## 3.2 Social Dialogue

Authors: Amelia Hardy, Ashwin Paranjape, Ines Gerard-Ursin, Esin Durmus, Faisal Ladhak, Joon Sung Park, Mina Lee

Dialogue is a natural and flexible way to engage in communication, which makes it an intuitive and popular mode of interaction for LMs. In this section, we evaluate human-LM interaction in the context of open-ended dialogue about social situations (henceforth social dialogue).

**Task.** We consider the task where given a social *scenario*, users converse with a system (or chatbot) about the scenario. Users initiate the conversation and take turns with the chatbot to carry out the conversation until users choose to finish it. We randomly select ten scenarios from the `EmpatheticDialogues` (Rashkin et al., 2019) and `CommonsenseDialogues` datasets (Zhou et al., 2021) after manually filtering out potentially sensitive and upsetting scenarios from the validation and test sets of the datasets.

**System logic.** Figure 3 shows a dialogue system *state* consisting of scenario, dialogue history, and user input and possible *actions*: pressing a key to modify user input, clicking the "send" button to submit their input, and finishing the dialogue. When users click the "send" button, the system creates a prompt with five dialogue examples for in-context learning and the current dialogue history, invokes an LM with the prompt, fetches a completion, and shows the completion in the dialogue history in the interface (`ShowCompletions`).

In the dialogue system, `CreatePrompt` creates a prompt by concatenating four example dialogues for in-context learning and the current dialogue history. While doing so, we omit the scenario information as the scenario is only known to the user. For `QueryLM`, we use `top_k` = 50 and `temperature` = 0.9 for decoding parameters and use HTML tags to delineate a conversation and the turns within.

**User study procedure.** We recruited a total of 189 crowd workers (or users) on Amazon Mechanical Turk. For each of the scenarios and models, we had three users each produce a dialogue. To ensure each dialogue is long enough, the interface allowed users to finish the conversation only after taking more than ten turns or exceeding 250 words in total. At the end of the conversation, users completed a survey about their experience.

**Survey questions.** We ask users to evaluate the *interestingness*, *sensibleness* (i.e., whether a chatbot response makes sense), and *specificity* (i.e., whether a chatbot response is specific to what a user had said) of a chatbot, following the survey questions proposed by Thoppilan et al. (2022). We also ask users to evaluate the *fluency* of the chatbot and their *inclination* to continue talking with the chatbot, as proposed by Smith et al. (2022). We ask dataset-specific questions regarding empathy (`EmpatheticDialogues`) and

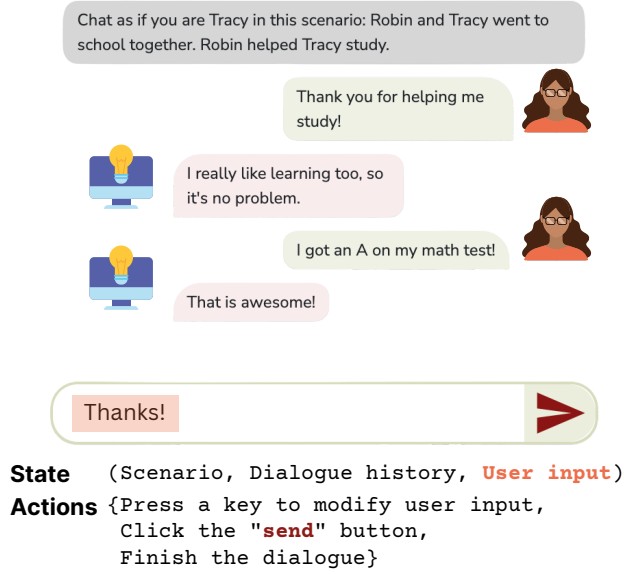

**State** (Scenario, Dialogue history, **User input**)
**Actions** {Press a key to modify user input,
Click the "**send**" button,
Finish the dialogue}

Figure 3: [**Social dialogue**] A dialogue system's *state* consists of a scenario, dialogue history, and user input. When users take an *action* (e.g., clicking the "send" button), the system updates the state (e.g., updating the dialogue history with a model completion).

| Model | Fluency | Sensibleness | Specificity | Humanness | Interestingness | Inclination | Reuse |
|---|---|---|---|---|---|---|---|
| | | | | (/100%) ↑ | | | (/5) ↑ |
| TextDavinci | $93 \pm 1.0$ | $94 \pm 1.0$ ** | $83 \pm 1.6$ * | $37 \pm 2.0$ | $36 \pm 2.0$ | $91 \pm 1.2$ | $4.09 \pm .14$ ** |
| TextBabbage | $90 \pm 1.4$ | $84 \pm 1.7$ * | $81 \pm 1.8$ * | $29 \pm 2.1$ | $30 \pm 2.1$ | $88 \pm 1.5$ | $3.35 \pm .16$ * |
| Davinci | $92 \pm 1.3$ | $89 \pm 1.4$ | $92 \pm 1.3$ ** | $24 \pm 2.0$ | $27 \pm 2.0$ | $91 \pm 1.3$ | $3.80 \pm .17$ |
| Jumbo | $89 \pm 1.3$ | $86 \pm 1.5$ | $84 \pm 1.5$ | $24 \pm 1.8$ | $32 \pm 2.0$ | $87 \pm 1.4$ | $3.21 \pm .20$ * |

Table 2: [**Social dialogue**] Users perceived TextDavinci to have the best *fluency*, *sensibleness*, *humanness*, *interestingness*, and *quality*, but they expressed the similar *inclination* to continue interacting with Davinci whose responses were most *specific* to what users had said. For the first six metrics, the numbers indicate the percentages of system responses under each metric (0–100%). The numbers for *reuse* indicate the ratings of each model after completing a dialogue (1–5). The means, standard errors, and statistical significance[4] are shown in the table.

commonsense understanding (CommonsenseDialogues), which are labeled as *humanness* collectively. Finally, after a dialogue, we ask if users are willing to talk to the chatbot again using a 5-point Likert scale: *reuse*.

For most questions, we ask for a turn-level annotation. To reduce users' workload, we negated those questions where we expected the majority of turns to be annotated positively. For all questions, if users decided to mark none of the utterances, they had to explicitly mark a checkbox acknowledging it.

- **Fluency** (turn-level; binary; reversed scores for the negated question): Which responses did NOT sound human?

- **Sensibleness** (turn-level; binary; reversed scores for the negated question): Mark responses where the chatbot did NOT make sense.

- **Specificity** (turn-level; binary; reversed scores for the negated question): Mark the responses that were NOT specific to what you had said, i.e., responses that could have been used in many different situations. For example, if you say "I love tennis" then "That's nice" would be a non-specific response, but "Me too, I can't get enough of Roger Federer!" would be a specific response.

- **Humanness** (turn-level; binary): Which responses did you feel an emotional connection to? (`Empathetic-Dialogues`); Which responses made you feel the chatbot understood social contexts and situations? (`CommonsenseDialogues`)

- **Interestingness** (turn-level; binary): Mark the responses that were particularly interesting or boring

- **Inclination** (turn-level; binary; reversed scores for the negated question): Which responses made you NOT want to talk with the chatbot again?

- **Reuse** (dialogue-level; 5-point Likert scale): Would you want to talk to this chatbot again?

- **Feedback** (dialogue-level; free-form; optional): Is there anything else you would like to say about the conversation?

**Results.** For social dialogue, we specifically care about models' ability to show an emotional connection to users, by being empathetic and having commonsense about social situations (*humanness*), in addition to being *fluent* and *sensible*, and *specific* to what they have said. Independently, users may have their own subjective views on whether system responses are interesting and make them want to continue interacting with the system, which are measured by preference metrics *interestingness* and *inclination*, respectively.

❶ **Instruction tuning improves performance on most quality metrics, but not specificity.** Table 2 shows that the instruction tuned TextDavinci scored highest on most metrics, including *fluency*, *sensibleness*, and *humanness*. In particular, users evaluated TextDavinci to have considerably higher *humanness* than Davinci. And the instruction-tuned TextBabbage model, despite its much smaller size, was reported to have higher humanness than Davinci or Jumbo.

However, instruction tuning decreased the *specificity* of model completions as shown in the significant drop in performance of TextDavinci compared to Davinci (Table 2). From this observation, we hypothesize that general instruction-tuning makes models better aligned with most metrics at the cost of being more generic, although previous work suggests that dialogue-specific fine-tuning improves all metrics including specificity Thoppilan et al. (2022).

The following are two sets of completions generated by TextDavinci and Davinci (`top_k` = 50, `temperature` = 0.9, `num_completions` = 5) to contrast their *specificity* (i.e., how specific it is to what users had said).

```
User:  I have a cat.

Chatbot (TextDavinci):
1.  What is your cat's name?
2.  What is your cat's name?
3.  What is your cat's name?
4.  What's your cat's name?
5.  What's your cat's name?

Chatbot (Davinci):
1.  Tell me more about your cat.
2.  You have a cat?
3.  This sentence is false.
4.  You have a what?
5.  What kind of cat?

User:  I am going home for Christmas.

Chatbot (TextDavinci):
1.  Really?  That sounds like fun!
2.  That sounds like a lot of fun!
3.  Have a safe trip!
```

---

[4]Results are denoted by [*] if models had a significant effect relative to TextDavinci, [*] if significant relative to TextBabbage, [*] if significant relative to Davinci, and [*] if significant relative to Jumbo at the $p = 0.05$ level using a Tukey-Kramer test.

```
4.  Have a safe trip!
5.  I hope you have a safe and happy trip!

Chatbot (Davinci):
1.  Where do you live?
2.  I am going home for Christmas.
3.  What is your name?
4.  What are your plans for Christmas?
5.  How are you traveling?
```

Although some of Davinci's responses are inappropriate (#3 in the first example) or repetitive (#2 in the second example), overall, they tend to be more specific and diverse to the user utterance.

❷ **Users may prefer to interact with a more specific LM.** Although users evaluated TextDavinci to be the best LM according to most metrics, they expressed similar *inclination* to continue interacting with Davinci during dialogue. Given that Davinci was perceived to be worse than TextDavinci on all metrics except *specificity* and *inclination*, we hypothesize that users expressed an inclination to interact with Davinci *because* its responses were the most specific to what users had said. We recommend further study of the relationship between specificity and inclination to interact with an LM, and in particular, whether specificity affects *reuse* of the LM once the initial novelty of this interactive experience has faded.

### 3.3 Question Answering

Authors: Megha Srivastava, John Thickstun, Rose E. Wang, Minae Kwon, Mina Lee

Question answering is a canonical task in NLP. We consider an *interactive* version of question answering where a user is asked to answer a question given access to an LM that they can query. In practice, users tend to query information systems repeatedly, refining and reformulating their queries in response to the system's output (Huang & Efthimiadis, 2009; Jansen et al., 2009; Jiang et al., 2013).

**Task.** Given a sequence of multiple-choice questions (or *quiz*), users try to select the correct answer with advice from an LM. As an example, consider this multiple-choice question:

```
Before Nixon resigned how many believed he should be removed from office?

A. 79% B. 98 % C. 33 % D. 57%
```

Users can query the system multiple times with free-form text inputs; they might start with the question verbatim, re-phrase the question to increase specificity (e.g., "Did 79 percent of people believe Nixon should have been removed from office?"), or find a different way to query, perhaps bringing in their own prior knowledge (e.g., "Nixon approval rating after watergate").

We use questions from the Measuring Massive Multitask Language Understanding (MMLU) dataset Hendrycks et al. (2021). Concretely, we first chose five diverse subjects from the dataset (Global facts, Nutrition, US foreign policy, College chemistry, and Miscellany), and selected 6 questions from each subject to construct a pool of 30 questions. We constructed quizzes by randomly selecting ten questions from the pool and adding one attention check question in the middle.

**System logic.** Figure 4 shows a *state* of our QA system consisting of a multiple-choice question, user input, and system output. Users can take the following *actions*: pressing a key to modify user input, clicking the "generate" button to query the system, selecting one of the multiple choices, clicking the "next" button to submit their answer, and finishing the quiz. When users click the "generate" button, the system creates a prompt with the user input, invokes an LM with the prompt, fetches a completion, and shows the completion in the system output box in the interface (ShowCompletions).

In the QA system, CreatePrompt creates a prompt by simply copying and pasting user input from the interface. Note that we do not include a multiple-choice question as part of the prompt. For QueryLM, we use temperature $= 0.5$ and max_tokens $= 100$ for decoding parameters.

| Model | Accuracy (/100%) ↑ | Time (min) ↓ | Queries (#) ↓ | Ease | Fluency (/5) ↑ | Helpfulness |
|---|---|---|---|---|---|---|
| TextDavinci | $69 \pm 2.2$ *** | $1.36 \pm .13$ | $1.78 \pm .06$ ** | $4.53 \pm .08$ *** | $4.35 \pm .07$ *** | $4.60 \pm .07$ *** |
| TextBabbage | $52 \pm 2.8$ * | $1.77 \pm .33$ | $2.57 \pm .13$ * | $4.09 \pm .12$ * | $3.84 \pm .12$ *** | $3.84 \pm .12$ *** |
| Davinci | $48 \pm 2.7$ * | $2.09 \pm .14$ | $2.66 \pm .12$ * | $3.73 \pm .13$ * | $3.22 \pm .11$ ** | $3.52 \pm .13$ *** |
| Jumbo | $54 \pm 2.9$ * | $1.67 \pm .09$ | $2.32 \pm .11$ | $3.87 \pm .14$ * | $3.17 \pm .11$ ** | $3.26 \pm .14$ *** |

Table 3: [**Question answering**] Performance averaged across all questions conditioning on the use of AI assistance. Users assisted by TextDavinci achieved the highest *accuracy* while requiring the least effort (*queries* and *ease*) and being perceived to be the most *fluent* and *helpful*. The numbers indicate means and standard errors, and the markers denote statistical significance,[4] conditioning on the use of AI assistance; when the assistance was provided, users queried the system 86% of the time.

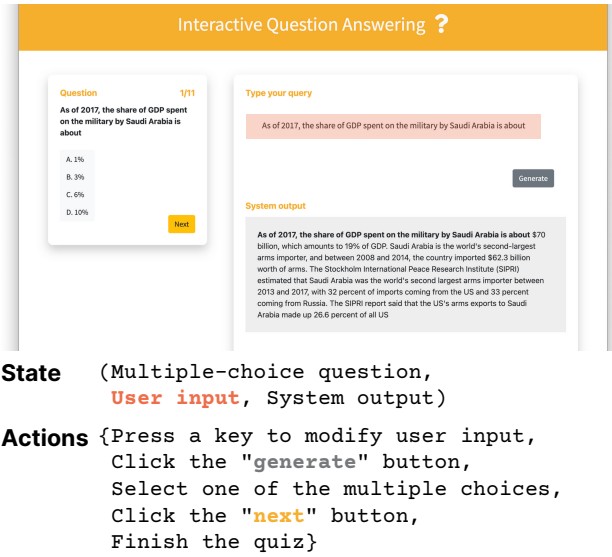

**State** (Multiple-choice question, **User input**, System output)

**Actions** {Press a key to modify user input,
Click the "**generate**" button,
Select one of the multiple choices,
Click the "**next**" button,
Finish the quiz}

Figure 4: [**Question answering**] The system's *state* consists of a multiple-choice question, user input, and system output. When users take an *action* (e.g., clicking the "generate" button), the system updates the state (e.g., updating the system output with a model completion).

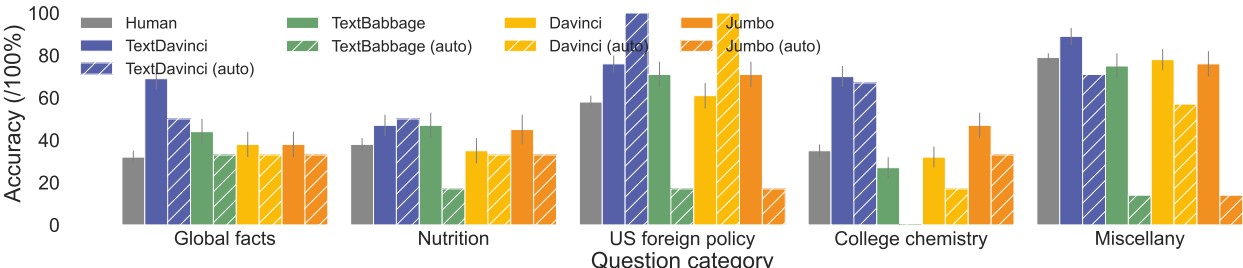

Figure 5: [**Question answering**] **Per-question accuracy broken down by category.** With Text-Davinci, human-LM interaction results in improved accuracy for Global Facts, College Chemistry, and Miscellany, but not Nutrition and US foreign policy. The shaded bars indicate the performance of models as zero-shot, non-interactive QA systems when given a question text verbatim, and gray bars indicate user performance without AI assistance. The results are averaged across questions for which users were provided with AI assistance.

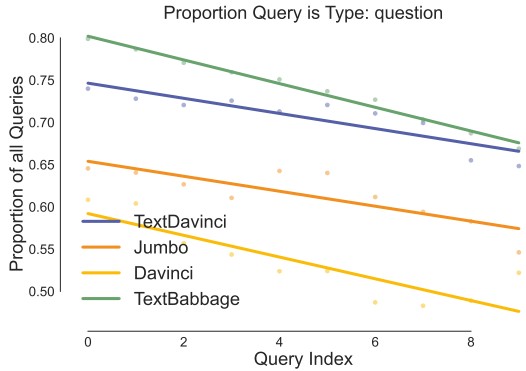 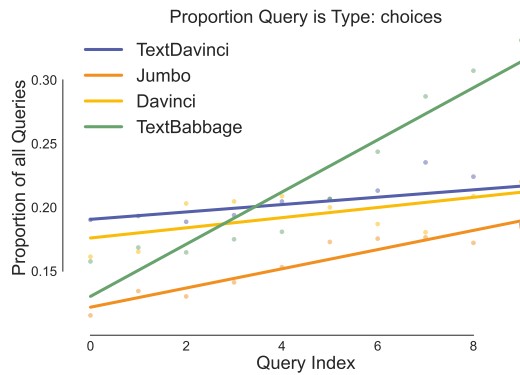

(a) Proportion of prompts that are framed as questions decreases over time for all models.

(b) Proportion of prompts that include answer choices increases for all models, but particularly for `TextBabbage`.

Figure 6: [**Question answering**] Quantitative analysis of user accommodation shows that users decrease framing their prompts as questions over time, and gradually increase using the LM to verify answer choices over time, supporting user survey responses.

**User study procedure.**   We recruited 342 crowd workers (users) on Amazon Mechanical Turk. Each user answered half of the questions with assistance from a single LM (treatment) and answered the other half without assistance (control). We instructed users not to use search engines to answer questions, and alerted users ("Please do not switch tabs or open a new window during your participation.") when we detected the tab switching. At the end of each quiz, users completed a survey about their experience.

**Survey questions.**   We asked users for first-person evaluation of *fluency*, *helpfulness*, and *ease of interaction* of the system on a 5-point Likert scale. Also, we asked users to describe why they found the system helpful or unhelpful, provide adjectives to describe the system, and explain whether their interaction changed over the course of answering questions. Concretely, the following questions were asked at the end of each quiz:

- **Fluency** (5-point Likert): How clear (or fluent) were the responses from the AI Assistant?

- **Helpfulness** (5-point Likert): Independent of its fluency, how helpful was having access to the AI Assistant compared to not having access?

- **Helpfulness** (free-form): Why did you find the AI Assistant helpful or unhelpful?

- **Ease of interaction** (5-point Likert): How easy was it to interact with the AI Assistant?

- **Change** (free-form): Did the way you chose to interact with the AI Assistant change over the course of answering questions? If so, how?

- **Feedback** (free-form; optional): Any feedback or comments?

**Results.**   For the goal-oriented QA task, third-party evaluation is characterized by *accuracy* (i.e., how many questions users get correct answers for) and *efficiency* (i.e., how much effort users put in to find an answer) (Dibia et al., 2022). Because the bulk of a user's time in interactive tasks is spent reacting to system outputs, we count the number of *queries* needed to answer each question as a proxy measurement for efficiency.

❶ **Users with LM assistance generally outperform the user or LM alone, but not always.** Figure 5 shows human-LM accuracy, zero-shot LM accuracy,[5] and human-LM interactive accuracy broken down by question category. Users assisted by `TextDavinci` outperformed users alone in all categories. Users

---

[5]For fully-automated models, we report deterministic results with `temperature = 0`.

with LM assistance generally outperformed an LM alone, with the notable exception of the US Foreign Policy category, where both `TextDavinci` and `Davinci` achieved better accuracy as zero-shot QA systems than as interactive assistants. For the Global Facts and Miscellany categories, users with LM assistance significantly outperformed zero-shot LMs. In these contexts, users are able to figure out how to query the LM and when to trust it.

❷ **Non-interactive performance does not always lead to better human-LM interaction.** Further analysis of Figure 5 shows that for the Nutrition category, `TextBabbage` had the worst accuracy as a zero-shot QA system, but achieved the best performance as an interactive LM assistant (tied with `TextDavinci`). For US Foreign Policy, `Jumbo` performed worst as a zero-shot QA system and `Davinci` performed best, but as an LM assistant `Jumbo` outperformed `Davinci`. We speculate that this might be due to different use cases of LMs in interactive settings (e.g., retrieving relevant information as opposed to directly answering questions). This supports the broader point that non-interactive performance is an imperfect proxy for evaluating human-LM interaction.

❸ **Instruction tuning improves accuracy and efficiency for human-LM interaction.** Table 3 shows that `TextDavinci` was the most efficient and accurate tool (1.78 queries/question with 69% accuracy). In contrast, `Davinci` was least efficient, requiring an average of 1.5x as many queries to answer a question with only 48% accuracy. Despite its smaller size, `TextBabbage` performed better than `Davinci` on most metrics, demonstrating the effectiveness of instruction tuning for this task. Instruction-tuned models were also perceived most favorably in first-party survey evaluation.

❹ **Users have different intentions.** We examine user accommodation, or change in behavior in response to the AI system, in two ways: qualitatively, through a free-response survey question, and quantitatively, via measuring the change in query strategies over time. Example responses to the survey question *"Did the way you chose to interact with the AI Assistant change over the course of answering questions? If so, how?"* indicate that users discovered strategies such as including potential answers in their prompts, or phrasing their prompts as declarative statements, over the course of time, as shown below:

> `TextDavinci` User: I tried a few question formats and found that simply repeating the
> question followed by the answer choices (but not preceded by the answer choices) worked
> so I continued to do that.
>
> `TextBabbage` User: I initially didn't include the multiple choice options because I
> didn't know how helpful it would be to include them, but after a couple questions with
> the AI not giving definitive answers without them, I started including them.
>
> `Davinci` User: I felt I could a better response if instead of asking a question I typed
> in the beginning of a thought. For example, instead of saying "what causes cancer?"
> it's better to type in "cancer is caused by...".
>
> `Jumbo` User: It seemed like providing an unfinished sentence as a prompt often provided
> more useful results than a simple question.

To further examine these trends quantitatively, we categorized all user prompts using the below taxonomy in order to measure how the distribution of prompt type changed for users over time:

- **Question**: User input ends with "?" or starts with one of the following tokens {"who", "what", "where", "how", "which", "why", "when", "whose", "do", "does", "did", "can", "could", "has", "have", "is", "was", "are", "were", "should"}, following Pang & Kumar (2011)

- **Close**: User input has normalized similarity $s$ where $70 < s < 100$ with the question text

- **Keyword**: User input consists of less than five words, similar to a search engine query

- **Exact**: User input has normalized similarity $s = 100$ with the question text

- **Completion**: User input consists of the start of sentences the LM should fluently complete, defined by ending with one of the following tokens created from the manual analysis: {"is", "was", "by", "may", "cause", "are", "of", "about", "the", "to", "their"}

- **Choices**: User input contains answer choices provided in the question text

- **Command**: User input commands the language model to perform a task, such as "list" or "finish the sentence"

- **Meaning**: User input provides definition or information about meaning, such as "synonym of" or "words for"

- **Others**

We observe that users do indeed gradually decrease the use of questions as prompts for all models (Figure 6a), while increasing the inclusion of answer choices in their prompts, especially for `TextBabbage` (Figure 6b), quantitatively supporting user survey responses. We additionally found that users were less likely to copy the exact `MMLU` question text for all models over time, further supporting our motivation of interactive LM evaluation of question answering where users engage in query reformulation.

### 3.4 Crossword Puzzles

Authors: Megha Srivastava

Crossword puzzles have been studied as a challenging task for AI systems (Ginsberg, 2014; Littman et al., 2002; Wallace et al., 2022; Rozner et al., 2021). Unlike multiple-choice QA, solving a crossword puzzle is a *generative* task requiring open-ended responses to clues. The crossword puzzle task also provides additional structure, whereby a user can check whether a candidate answer satisfies the lexical constraints of the puzzle. Finally, clues are often not straightforward (e.g. "Christmas in Chamonix"), and a user might need to reformulate the query to find the desired information.

**Task.** A crossword puzzle requires solving a series of word clues (e.g., "Oscar the Grouch's home") that correspond to either rows ("Across" clues) or columns ("Down" clues) of white squares on a rectangular grid. As users solve one clue and places letters in squares, they reveal partial information (e.g., shared letters) that can affect their strategy for solving future clues. In our setting, users try to complete an entire crossword puzzle with the ability to query an LM-based "AI Teammate" via a chat-based interface. Users may message the AI Teammate with free-form text, where each individual message is used as the full prompt to the underlying LM.

**System logic.** Figure 7 shows a *state* of our crossword system, which we adapt from *Down for a Cross*, an open-source software for the multi-player crossword puzzles. Users can take the following *actions*: pressing a key to modify user input, pressing the "enter" key to submit input, selecting a square in the puzzle, entering a letter into a square, selecting a clue from the list, and finishing the session. While attempting to solve different crossword clues (`ShowCompletions`), users (with the chat ID: `You`) are able to repeatedly interact with an LM (with the chat ID: `AI`) via zero-shot prompts in a dialogue-based chat interface. Although we display the chat history to aid players in remembering past information, only the most recent prompt from the player is sent to the LM. The same LM is fixed throughout the course of the task, which ends either after 30 minutes or when all clues are correctly solved.

In the crossword puzzle system, `CreatePrompt` creates a prompt by simply copying and pasting user input from the interface, without any further context. For each of the four LMs we study, `QueryLM` uses `temperature` = 0.5 and `max_tokens` = 100 for decoding parameters.

**User study procedure.** We recruited 350 workers (users) on Amazon Mechanical Turk, split across each of the four language models and five puzzles (LOVE, SIT, NYT-1, NYT-2, and ELECT). Each user was provided a link to the interface, and asked to engage with the puzzle (e.g., solve clues, interact with the LM) for at least 30 minutes. Afterwards, each user completed a survey about their experience interacting with the AI Teammate.

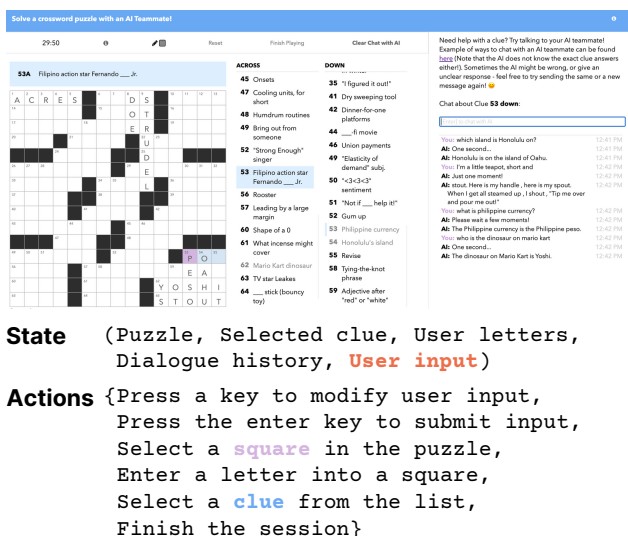

**State** (Puzzle, Selected clue, User letters, Dialogue history, **User input**)

**Actions** {Press a key to modify user input,
Press the enter key to submit input,
Select a **square** in the puzzle,
Enter a letter into a square,
Select a **clue** from the list,
Finish the session}

Figure 7: [**Crossword puzzles**] The system's *state* consists of a crossword puzzle, selected clue (can be none), user letters entered in the puzzle, dialogue history, and user input. When users take an *action* (e.g., pressing the enter key after writing user input), the system updates the state (e.g., updating the dialogue history with a model completion).

**Survey questions.** After playing with provided crossword puzzles, users are then asked to rank different qualities of the AI assistant on a 5-point Likert scale, including *fluency*, *helpfulness*, *enjoyment*, and *ease of the interaction* with the AI Teammate. Users are additionally asked to describe why they found the teammate helpful or unhelpful, what adjectives best describe the assistant, and whether their interaction changed over the course of answering questions. Concretely, the following questions were asked at the end of the session:

- **Fluency** (5-point Likert): How clear (or fluent) were the responses from the AI Teammate?

- **Helpfulness** (5-point Likert): Independent of its fluency, how helpful was your AI Teammate for solving the crossword puzzle?

- **Helpfulness** (free-form): Why did you find the AI Teammate helpful or unhelpful?

- **Ease** (5-point Likert): How easy was it to interact with the AI Teammate?

- **Enjoyment** (5-point Likert): How enjoyable was it to interact with the AI Teammate?

- **Change** (free-form): Did the way you chose to interact with the AI Teammate change over the course of solving the crossword puzzle? If so, how?

- **Description** (free-form): What adjectives would you use to describe the AI Teammate?

- **Feedback** (free-form; optional): Any feedback or comments?

**Results.** Solving a crossword puzzle is simultaneously an open-ended goal-oriented task and a form of entertainment—therefore, third-party evaluation should consider both *accuracy* and *engagement* of the human-LM interaction. Because some users may enjoy solving crossword puzzles independently, we count the number of *queries* used over the course of the task as a proxy measurement for user preference to engage with a given LM.

| Model | Accuracy (letter) (/100%) ↑ | Accuracy (clue) | Fluency | Helpfulness (/5) ↑ | Ease | Enjoyment |
|---|---|---|---|---|---|---|
| TextDavinci | $63 \pm 2.9$ * | $53 \pm 3.4$ * | $3.65 \pm .10$ ** | $3.14 \pm .12$ *** | $4.35 \pm .10$ ** | $2.91 \pm .20$ *** |
| TextBabbage | $47 \pm 3.3$ * | $38 \pm 3.5$ * | $3.14 \pm .13$ ** | $2.27 \pm .14$ * | $3.78 \pm .15$ ** | $2.19 \pm .22$ ** |
| Davinci | $55 \pm 3.5$ | $46 \pm 3.6$ | $2.26 \pm .11$ ** | $1.92 \pm .10$ * | $3.32 \pm .14$ ** | $1.92 \pm .17$ ** |
| Jumbo | $56 \pm 2.8$ | $45 \pm 3.1$ | $2.30 \pm .10$ ** | $2.20 \pm .10$ * | $3.08 \pm .15$ ** | $1.66 \pm .18$ * |

Table 4: [**Crossword puzzles**] Users assisted by TextDavinci found their model more *fluent*, *helpful*, and *easy* and *enjoyable* to interact with compared to other models, and in general provided more accurate solutions across all puzzles. However, while users with Davinci and Jumbo performed worst on the self-reported survey metrics, users with TextBabbage had the worst *accuracy*, suggesting a disconnect between first-person preference and automated quality metrics. The numbers indicate means and standard errors, and the markers denote statistical significance.[4]

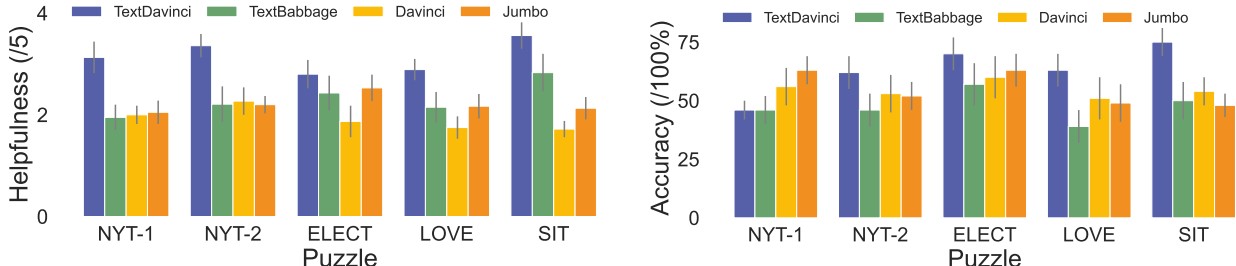

Figure 8: [**Crossword puzzles**] Although across all puzzles, users assisted by TextDavinci were significantly more likely to report the model as helpful (left), they did not always provide significantly more accurate puzzle solutions (right). In fact, for the NYT-1 puzzle, users assisted by TextDavinci were significantly less likely to provide accurate solutions, despite being more likely to find the model helpful.

❶ **Helpfulness perceived by users exceeds accuracy improvements.** We measure the perceived *helpfulness* of a model based on a post-task survey question with a 5-point Likert scale. Figure 8 shows that users significantly preferred TextDavinci over other models with respect to helpfulness for all puzzles. Interestingly, this perception of helpfulness is not necessarily reflected in the overall interaction *accuracy*: as shown in Figure 8, assistance from TextDavinci led to statistically significantly higher crossword letter accuracy only for the SIT crossword puzzle; for the NYT-1 puzzle, users actually performed *worse* with both instruction tuned models (TextDavinci and TextBabbage). One hypothesis for this behavior is that these models are capable of providing confident and fluent misinformation, which can be incorrectly perceived to be helpful.

Furthermore, the variance in model performance between puzzles strongly suggests that different puzzles' distributions over clue types may require different capabilities from LMs. In Table 5, we examine performance across 6 clue category types proposed in Wallace et al. (2022), and find that LMs, and in particular Text-Davinci, were most useful for clues about factual knowledge, but struggle with wordplay and phrases.

❷ **Short prompts exacerbate misinformation and toxicity.** We observed significant amounts of misinformation across all models. Misinformation was particularly pernicious using TextBabbage, which achieved relatively positive responses in user-reported survey responses but the lowest objective accuracy for puzzle solutions provided by users (Table 4). We include a set of examples of misinformation from TextBabbage and TextDavinci below. While we believe misinformation played a big role in the difference between user-perceived and actual performance on the task, the overall effect of misinformation might be more nuanced: the structure of crossword puzzles provides constraints that may help users more quickly identify incorrect model suggestions.

| Model | Knowledge | Definition | Commonsense (/1) ↑ | Phrase | Wordplay | Cross-Reference |
|---|---|---|---|---|---|---|
| TextDavinci | 0.78 ± .07 | 0.61 ± .08 | 0.63 ± .08 | 0.47 ± .09 | 0.42 ± .09 | 0.42 ± .11 |
| TextBabbage | 0.60 ± .09 | 0.47 ± .09 | 0.51 ± .08 | 0.28 ± .08 | 0.35 ± .09 | 0.25 ± .09 |
| Davinci | 0.59 ± .09 | 0.50 ± .09 | 0.55 ± .09 | 0.44 ± .09 | 0.37 ± .09 | 0.29 ± .11 |
| Jumbo | 0.59 ± .08 | 0.52 ± .09 | 0.56 ± .08 | 0.39 ± .09 | 0.43 ± .09 | 0.23 ± .09 |

Table 5: [**Crossword puzzles**] Per-model clue accuracy for different clue categories across all puzzles and users.

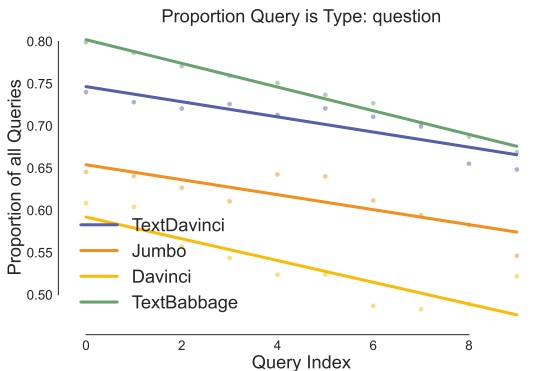

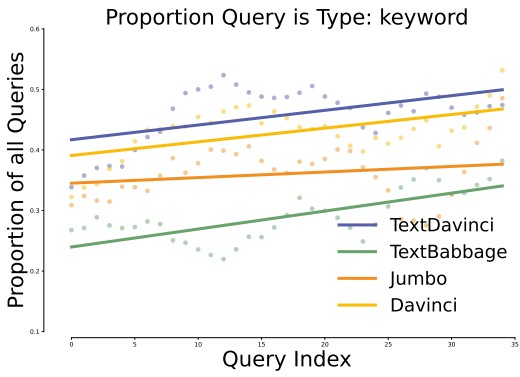

(a) Proportion of prompts that are framed as questions decreases over time for all models.

(b) Proportion of prompts that consist of short keywords increases for all models.

Figure 9: [**Crossword puzzles**] Quantitative analysis of user accommodation shows that users decrease framing their prompts as questions over time, similar to the QA task, but gradually increase using prompts consisting of few short keywords, similar to crossword clue text.

Here, we provide several examples of misinformation observed from the TextDavinci and TextBabbage models, which we note may be particularly serious due to the "confidence" and high fluency of these model's completions.

```
Prompt:  Who won the World Cup in 2018?
TextBabbage:  The United States

Prompt:  Which US Vice President resigned from office?
TextBabbage:  George H.W. Bush

Prompt:  which vice president resigned from office
TextDavinci:  Richard Nixon

Prompt:  Who are the two vice presidents who resigned from office?
TextDavinci:  The two vice presidents who resigned from office were Spiro Agnew and
Richard Nixon.
```

Additionally, we observed instances of toxic content in generated crossword outputs. Explicit responses often occurred when users, perhaps influenced by the style of crossword clues, provided short keyword prompts: e.g., "japanese car," "nick wilde," or "synonym of delayed." Short, ungrammatical prompts may more closely match toxic patterns in the training data (e.g., pornographic video captions or social media comments). Other safety concerns we observed included generating personal phone numbers and YouTube accounts belonging to non-celebrities, as well as discussing more sensitive topics. Longer prompts that setup the appropriate context might mitigate toxic outputs, but its effectiveness merits further study.

Here are several instances of toxic content in completions we observed. Although toxic content was more common for Davinci (12) and Jumbo (8), than TextBabbage (5) and TextDavinci (3), we still found serious examples of toxic content generated from TextBabbage and TextDavinci. Furthermore, models occasionally generated text that *escalated* the sensitivity of the player's prompt, such as in the following examples:

> **Prompt:** who leads a parade
>
> **TextDavinci:** of the Ku Klux Klan A man who leads a parade of the Ku Klux Klan is a Klansman.
>
> **Prompt:** Boxer's comeback
>
> **TextDavinci:** In the early 1990s, boxer Mike Tyson was convicted of rape and served three years in prison. After his release, he made a successful comeback, winning several bouts.
>
> **Prompt:** skipping in line called
>
> **TextBabbage:** for a moment of silence for the victims of the Pulse nightclub shooting a moment of silence for the victims of the Christchurch mosque shootings a moment of silence for the victims of the Las Vegas shooting

Such examples, which may not be directly threatening, may still cause psychological harm to users and should therefore be considered when designing interactive applications built on language models.

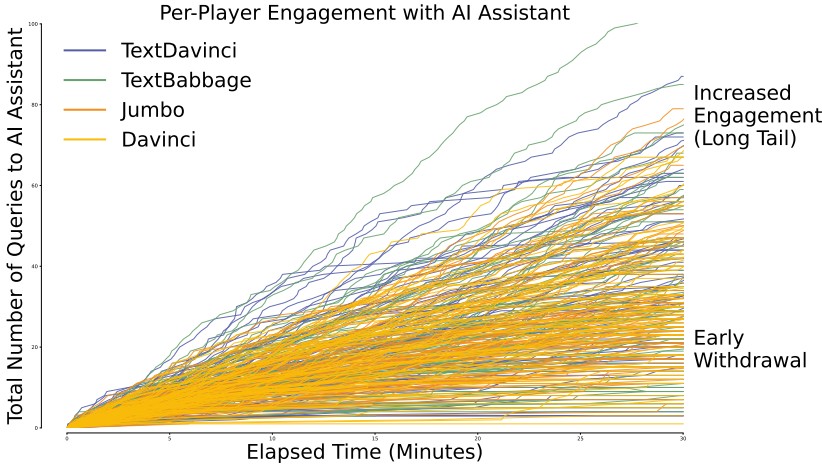

Figure 10: [**Crossword puzzles**] Cumulative queries to the AI Teammate over time for all users. Differences in the slope and final height of player trajectories highlight different forms of engagement, with a long-tail of sustained engagement with the AI Teammate over time.

❸ **Users demonstrate diverse engagement behavior.** The relatively long period of interaction between users and the AI Teammate in the crossword puzzle task provides an opportunity to closely study engagement, which we measure in two ways. First, we ask users in the post-task survey to answer *"How enjoyable was it to interact with the AI Teammate?,"* and show that users found TextDavinci significantly more enjoyable to use (for some puzzles, TextBabbage was also rated highly) in Table 4. This suggests that users tended to enjoy interacting with LMs that were most helpful for the task. However, in Figure 10, we take a closer look at engagement across the entire 30 minutes task length, and observe a diverse set of user behaviors: while some users decided to stop querying the AI Teammate after receiving incorrect responses, others chose to solve clues independently at the start and only query later when stuck. A long tail of users,

mostly interacting with `TextDavinci` and `TextBabbage`, had significant sustained interaction throughout the course of the puzzle, often experimenting with re-phrasing prompts to elicit more helpful responses. Finally, unlike with the use of AI solvers that seek to solve a crossword puzzle perfectly (Wallace et al., 2022), some users found the challenge of figuring out how to best use the AI Teammate itself entertaining:

> `Davinci` Player:  This is an enjoyable task that may not be as fun if the AI would give
> you all the answers!

❹ **Users accommodate their behaviors over time.**   We examine user accommodation, or change in behavior in response to the AI system, in two ways: qualitatively, through a free-response survey question, and quantitatively, via measuring the change in query strategies over time. Example responses to the survey question *"Did the way you chose to interact with the AI Teammate change over the course of solving the crossword puzzle? If so, how?"* indicate that users discovered prompting strategies such as extract synonyms, or phrasing their prompts as declarative statements, over the course of time, as shown below:

> `Jumbo` User:  with some of the questions, it was clear to me that I couldn't even lead
> the AI to an answer even giving hints, things like puns are a perfect example of this.
> multi-word phrases were another.  Where it was the most helpful is in asking for things
> like names of crackers or names of clowns or Ford model cars.  Factual type things that
> didn't require logic.
>
> `TextDavinci` User:  I tried to use a variety of techniques when asking questions, to
> avoid getting misleading answers, and I also learned quickly that I needed to confirm
> the responses by asking the same question multiple times (in different ways)
>
> `TextBabbage` User:  after I got some generic unhelpful responses, I realized I had to be
> more specific with my questions ...  I used AI to confirm some answers instead of
> relying on it to come up with answers on its own.
>
> `Davinci` User:  I learned how to get synonyms.  Instead of [blank] synonym I put [blank]
> thesaurus

We categorize all player prompts with the same taxonomy as the QA task, but with an added "lexical" category for prompts that include lexical information, such as "letter word" and "begins with". We observe that users do indeed gradually decrease the use of questions as prompts for all models (Figure 9a), while increasing their use of short keyword prompts (Figure 9b). Surprisingly, we additionally observe that edit distance between the prompt and clue text (typically short keywords) decreases over time, particularly for `TextDavinci`. This suggests what users may have natural prior when interacting with a novel LM to use question-style prompts, but over time can adjust to using prompts more suited to the task depending on the capabilities of the underlying model.

### 3.5   Text Summarization

Authors: Esin Durmus, Faisal Ladhak, Mina Lee

Text summarization is a long-standing problem in NLP (Luhn, 1958; Mani, 1999; Spärck Jones, 1999; Nenkova & McKeown, 2012). Notably, it has been studied in interactive settings for multi-document summarization, where the users interact with the system to query for additional information to expand upon an initial summary (Avinesh et al., 2018; Shapira et al., 2021; 2022). In contrast, we focus on human-LM interaction for single-document summarization, where users interact with the system to correct LM-generated summaries. In addition, as users correct summaries for a sequence of documents, we provide previous human-edited summaries as examples to the system to improve future summaries via in-context learning.

**Task.**   We consider the task, where given a document and a model-generated summary, users edit the summary to be consistent, relevant, and coherent. We randomly select 964 documents from `XSum` dataset Narayan et al. (2018) and construct 20 summarization *sessions* by randomly choosing ten documents per session without replacement.

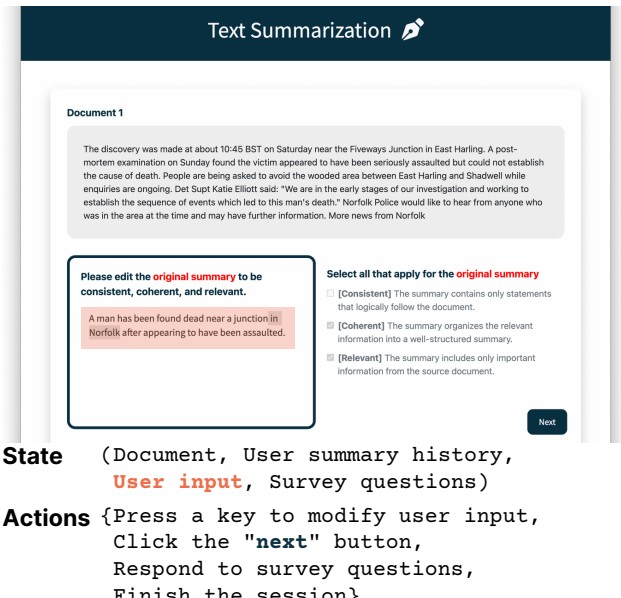

**State** (Document, User summary history,
User input, Survey questions)

**Actions** {Press a key to modify user input,
Click the "next" button,
Respond to survey questions,
Finish the session}

Figure 11: [**Text summarization**] The system's *state* consists of a document, user summary history, user input, and survey questions. When users take an *action* (e.g., responding to survey questions and clicking the "next" button), the system updates the state (e.g., showing the next document and model-generated summary).

**System logic.** Figure 11 shows the system state and actions. A summarization system *state* consists of a document, user summary history (not visible to users), user input, and survey questions. Users can take the following *actions*: pressing a key to modify a model-generated summary, clicking the "next" button, responding to survey questions, and finishing the session.

One notable difference in the summarization system's state is that `CreatePrompt` generates a prompt by concatenating all the previous document and user-edited summary pairs in the user's summary history as part of the prompt. This enables us to study whether an LM can learn from user examples and improve over time, as well as how users behavior change as the models learn or fail to learn from their examples.

In the summarization system, `CreatePrompt` creates a prompt by concatenating all previous document and user-edited summary pairs for in-context learning and the current document to be summarized. For the very first document, we perform one-shot learning, instead of zero-shot to avoid early user dropout, with the following example:

```
Document:  Fire crews and police were called to the property in Savile Road, Halifax,
at 05:37 BST and the body of a man in his 50s was found inside.  West Yorkshire Police
said he had not yet been identified.  Det Insp Craig Lord said:  "Inquiries are ongoing
today with West Yorkshire Fire and Rescue Service to determine the cause of this fire
which has sadly resulted in a man losing his life."

Summary:  A man has died in a fire at a flat in West Yorkshire.
```

For `QueryLM`, we use `temperature` $= 0.3$, `max_tokens` $= 64$, `stop_sequences` $= [$"***"$]$ as decoding parameters and return the first sentence of the model output as a summary.

**User study procedure.** We recruited 39 crowd workers (or users) on Amazon Mechanical Turk. For each model, we collected 20 summarization sessions (80 in total), while allowing the same users to participate multiple times. Upon receiving a model-generated summary for each document, users were asked to edit the summary to be consistent, relevant, and coherent. To encourage users to pay attention to the three quality criteria when editing, we asked users to evaluate the consistency, relevance, and coherence of the original

| Model | Original | Edited (word) | Edit distance ↓ | Revision ↓ | Helpfulness ↑ (/5) | Improvement ↑ |
|---|---|---|---|---|---|---|
| TextDavinci | 17.70 ± .47 | 25.08 ± .89 | 12.38 ± .89 * | 3.00 ± .19 | 4.20 ± .19 * | 2.60 ± .29 |
| TextBabbage | 20.11 ± .43 | 32.61 ± .85 | 16.34 ± .91 * | 3.40 ± .17 | 3.40 ± .28 | 2.15 ± .21 |
| Davinci | 16.96 ± .38 | 25.05 ± .89 | 14.19 ± .89 | 3.25 ± .20 | 3.80 ± .22 | 2.10 ± .20 |
| Jumbo | 14.75 ± .34 | 25.33 ± .82 | 15.12 ± .83 | 3.30 ± .21 | 3.30 ± .25 * | 2.40 ± .28 |

Table 6: [**Text summarization**] Users edited summaries generated by TextBabbage the most, and Text-Davinci the least. The survey responses for the perceived amount of revision (*revision*) accurately reflect the actual edit distance. Overall, users perceived TextDavinci to be most helpful (*helpfulness*) and better improves over time (*improvement*) compared to the other models. The first three metrics refer to the length of model-generated summaries (*original*), human-edited summaries (*edited*), and the Levenshtein distance between them (*edit distance*). The numbers indicate means and standard errors, and the markers denote statistical significance.[4]

| Model | Consistent (/100%) ↑ | Relevant (/5) ↑ | Coherent |
|---|---|---|---|
| TextDavinci | 65 ± 4 * | 4.07 ± .08 * | 4.70 ± .04 * |
| TextBabbage | 89 ± 3 *** | 4.15 ± .06 ** | 4.51 ± .06 * |
| Davinci | 57 ± 4 * | 3.70 ± .09 ** | 4.53 ± .05 |
| Jumbo | 56 ± 4 * | 3.80 ± .07 * | 4.60 ± .04 |

Table 7: [**Text summarization**] **Third-party evaluation of model-generated summaries.** According to third-party evaluation, TextBabbage performed the best with respect to *consistency* and *relevance*, while TextDavinci was rated highest for *coherence*. The numbers indicate means and standard errors, and the markers denote statistical significance.[4]

(model-generated) summary and edited summary before and after editing. At the end of a summarization session, users completed a survey about their overall experience interacting with the system.

**Survey questions.** We ask per-summary and per-session questions to the users. Summary-level questions ask *consistency*, *relevance*, and *coherence* of the original and edited summaries, following Fabbri et al. (2021). Session-level questions evaluated users' overall perceptions of the summarization system with respect to its *helpfulness* and *improvement over time*. At the end of each summarization session, we asked users the following questions:

- **Helpfulness** (5-point Likert): How helpful was having access to the AI assistant as an automated summarization tool?

- **Edit amount** (5-point Likert): How much did you have to edit the generated summaries?

- **Improvement** (5-point Likert): The AI assistant improved as I edited more summaries.

**Third-party human evaluation.** We also asked third-party evaluators to evaluate the consistency, relevance, and coherence of a subset of the summaries generated by LMs. To this end, we randomly sampled 100 documents from our user study and recruited 18 workers (third-party evaluators) on Amazon Mechanical Turk to assess the summaries written for the documents. Each summary was assessed by 3 different evaluators.

**Results.** In text summarization, we ask third-party evaluators to evaluate the quality of summaries with respect to their *consistency* (i.e., all the information in the summary is inferred from the document), *relevancy* (i.e., the summary includes only important information and no excess information), and *coherence* (i.e., the

summary organizes the relevant information in a well-structured manner). For first-person perspectives, we compute *edit distance* from interaction traces, and consider survey responses for *helpfulness* and *improvement* (i.e., the improvement of AI assistance as more summaries are edited).

❶ **There is a discrepancy between metrics based on third-party and first-person perspectives.** Table 6 shows that users found summaries generated by `TextDavinci` to be the most helpful starting point for editing, receiving the smallest *edit distance* and *revision* score along with the highest *helpfulness* score. However, from third-party evaluation, original summaries from `TextBabbage` were rated as the most *consistent* and *relevant* among the four models (Table 7). According to the *density* metric (i.e., the average length of the extractive fragment to which each word in the summary belongs) Grusky et al. (2018), summaries generated by `TextBabbage` are much more extractive (16.11) than those by other models (4.19 for `TextDavinci`, 4.86 for `Davinci`, and 4.09 for `Jumbo`). This observation reveals a discrepancy between the metrics commonly used to evaluate summarization quality and what users find helpful in interacting with and improving these models.

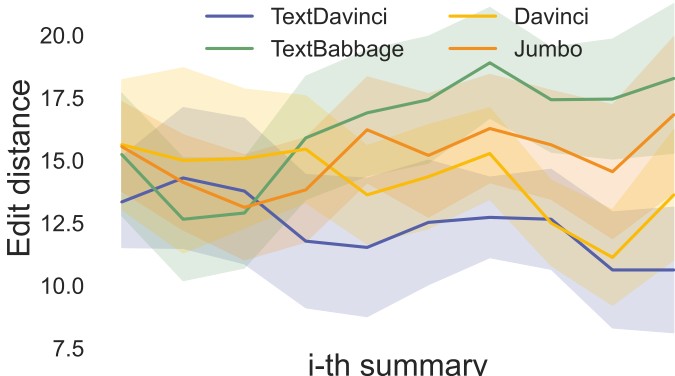

Figure 12: [**Text summarization**] We observe that users edited less with `TextDavinci` and more with `TextBabbage` over time, suggesting that `TextDavinci` is better at in-context learning the user's preferences for summarization. The plot shows the rolling average of edit distance between original and edited summaries over ten summaries (1st through 10th summary) across all summarization sessions, with the window size of 2.

❷ **Users save effort with `TextDavinci` over time, suggesting it is better at in-context learning.** We are interested in how users behavior change over time as LMs succeed or fail to learn from previous examples. To that end, we observed how the *edit distance* between the original and edited summaries change as a summarization session progresses. Figure 12 shows the change of edit distance across ten summaries in a sessions. We observe that users had to edit less with `TextDavinci` over time, whereas they had to edit more with `TextBabbage`. The survey response also indicated that users perceived `TextDavinci` to be best at improving over time (*improvement* in Table 6). From this observation, we hypothesize that `TextDavinci` is better at in-context learning the user's preferences for summarization.

### 3.6 Metaphor Generation

Authors: Frieda Rong, Xiang Lisa Li, Mina Lee

Metaphors are used throughout poetry, journalism, and science education to communicate complex or abstract ideas (Mio, 1997; Lakoff & Turner, 2009; Niebert et al., 2012). Creating metaphors requires divergent, lateral thinking (Glucksberg & McGlone, 2001; Gero & Chilton, 2018). To help with ideation, prior work designed metaphor generation tools where users could query multiple times to get suggestions and showed that these suggestions enabled people to write metaphors that they might never have thought of (Gero &

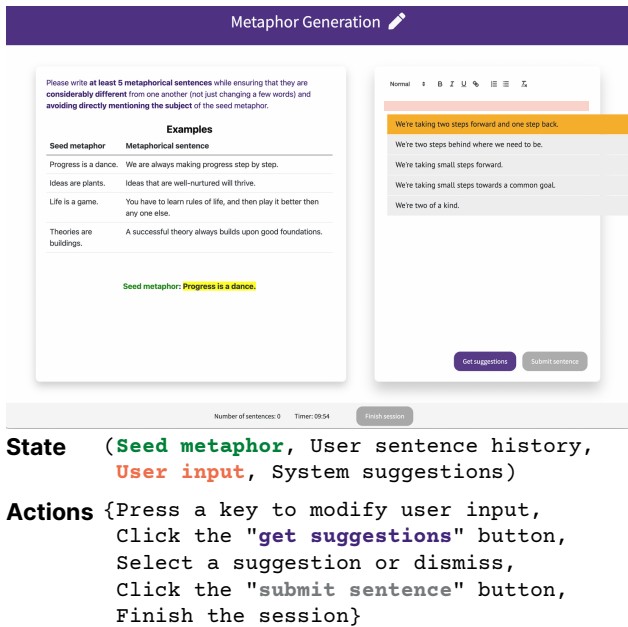

**State** (**Seed metaphor**, User sentence history,
**User input**, System suggestions)

**Actions** {Press a key to modify user input,
Click the "**get suggestions**" button,
Select a suggestion or dismiss,
Click the "**submit sentence**" button,
Finish the session}

Figure 13: [**Metaphor generation**] The system's *state* consists of a seed metaphor, user sentence history, user input, and system suggestions. When users take an *action* (e.g., clicking the "get suggestions" button), the system updates the state (e.g., showing five model completions as suggestions).

Chilton, 2019; Chakrabarty et al., 2022). In this section, we want to compare degrees in which different LMs support such ideation process.

**Task.** Given a seed metaphor, the task is to write metaphorical sentences that evoke the given metaphor in a limited amount of time. For example, for the metaphor "Time is money," we might generate the following metaphorical sentences:

```
How do you spend your time?
That flat tire cost me an hour.
I've invested a lot of time in him.
```

From Lakoff & Johnson (1980), we select 3 metaphors and their corresponding sentences as in-context learning examples, and choose 4 thematically diverse metaphors for evaluation. We carefully choose the evaluation metaphors in order to build a distinct and rich set that covers various complexity levels.

**System logic.** Figure 13 shows our metaphor system *state* consisting of a seed metaphor, user sentence history, user input, and system suggestions. Users can take one of the following *actions*: pressing a key to modify user input, clicking the "get suggestions" button, selecting a suggestion from the list of suggestions or dismiss all of them, and finishing the session. When users click the "get suggestions" button, the system creates a prompt with three in-context examples and the current user input, invokes an LM in the state, fetches five outputs with the prompt, and shows the outputs in the popup box in the interface (`ShowCompletions`).

In the metaphor system, `CreatePrompt` prepends the following three pairs of metaphor and metaphorical sentences to the seed metaphor to generate a prompt for few-shot learning.

```
Metaphor:  Argument is war.
Metaphorical Sentence:  He attacked every weak point in my argument.

Metaphor:  Time is money.
Metaphorical Sentence:  Is that worth your while?
```

```
Metaphor:  Love is a journey.
Metaphorical Sentence:  We'll just have to go our separate ways.
```

We carefully chose these metaphors from Lakoff & Johnson (1980) that are relatively simple and direct without being generic. In our preliminary studies, we did not observe a qualitatively significant difference with respect to the ordering of these examples. For `QueryLM`, we use `temperature` = 0.9, `max_tokens` = 30, `stop_sequences` = ["Metaphor:"] for decoding parameters.

**User study procedure.**   We recruited 32 crowd workers (users) on Amazon Mechanical Turk, and each of them could work on all four seed metaphors if desired. For each seed metaphor, we randomly assigned one of the four LMs. In each session, each user was given 10 minutes to come up with as many metaphorical sentences as they could using the system. In the instructions, we asked users to write sentences that are apt, specific, and imageable along with examples, following the evaluation criteria in Gero & Chilton (2019). At the end of the session, users completed a survey about their experience.

**Survey questions.**   In the survey, we asked about users' perceptions concerning *fluency*, *helpfulness*, *ease* of interaction, *enjoyment*, *satisfaction*, *ownership*, and willingness to *reuse* the system on a 5-point Likert scale, similar to Lee et al. (2022). At the end of each metaphor session, we asked users the following questions:

- **Fluency** (5-point Likert): How fluent were the responses from the AI assistant?

- **Helpfulness** (5-point Likert): During this writing session, the system helped me come up with new ideas.

- **Ease** (5-point Likert): It was easy to write metaphorical sentences with the system.

- **Enjoyment** (5-point Likert): I enjoyed this writing session.

- **Helpfulness** (free-form): What kinds of suggestions did you find helpful and why? (Give a concrete example if possible.)

- **Non-helpfulness** (free-form): What kinds of suggestions did you find not helpful and why? (Give a concrete example if possible.)

- **Editing** (free-form): What were the reasons you made edits, if any, to suggestions from the system?

- **Change** (free-form): Did the way you interacted with the AI assistant change over the course of the writing session? If so, how?

- **Satisfaction** (5-point Likert): I am satisfied with the metaphorical sentences I came up with.

- **Ownership** (5-point Likert): I feel like the metaphorical sentences are mine.

- **Reuse** (5-point Likert): I would be willing to use this system again.

- **Description** (free-form): What adjectives would you use to describe the AI assistant?

- **Feedback** (free-form; optional): Any feedback or comments?

**Third-party evaluation.**   We also asked third-party evaluators to evaluate whether written metaphorical sentences are apt, specific, and imageable, following Gero & Chilton (2019). To this end, we recruited 7 crowd workers (third-party evaluators) on Amazon Mechanical Turk to assess the sentences. Each sentence was assessed by 2 different evaluators.

**Results.**   In metaphor generation, we measure user effort in writing with *time* and the number of *queries* per sentence. Then, we look at user survey responses on *helpfulness*, *satisfaction*, *ease* of interaction, and willingness to *reuse* the system. Lastly, we evaluate the quality of metaphorical sentences (based on how *apt*, *specific*, and *imageable* the sentences are) through third-party evaluation.

| Model | Time (min) ↓ | Queries (#) ↓ | Acceptance (/100%) ↑ | Edit distance (word) ↓ |
|---|---|---|---|---|
| TextDavinci | 0.74 ± .07 | 0.92 ± .07 | 51 ± 4.4 ** | 4.79 ± .52 |
| TextBabbage | 0.73 ± .04 | 0.97 ± .09 | 56 ± 3.6 * | 6.43 ± .73 |
| Davinci | 0.60 ± .05 | 0.77 ± .06 | 71 ± 4.1 ** | 4.83 ± .60 |
| Jumbo | 0.75 ± .06 | 1.03 ± .11 | 68 ± 4.3 * | 5.59 ± .54 |

Table 8: [**Metaphor generation**] **User effort in writing a metaphorical sentence.** Users spent the least effort writing metaphorical sentences with Davinci, taking least *time* and making the fewest *queries* among the four models. The same model achieved the highest *acceptance* rate. However, conditioning on the acceptance of suggestions, users edited the least amount (*edit distance*) when working with suggestions from TextDavinci. The numbers indicate means and standard errors (for writing one metaphorical sentence), and the markers denote statistical significance.[4]

| Model | Helpfulness | Satisfaction (/5) ↑ | Ease | Reuse |
|---|---|---|---|---|
| TextDavinci | 4.21 ± .18 | 4.42 ± .14 | 3.68 ± .22 | 4.42 ± .18 |
| TextBabbage | 3.64 ± .21 | 4.14 ± .20 | 3.82 ± .23 | 4.39 ± .17 |
| Davinci | 4.17 ± .23 | 4.33 ± .20 | 3.94 ± .25 | 4.61 ± .14 |
| Jumbo | 4.13 ± .24 | 4.40 ± .13 | 3.87 ± .24 | 4.47 ± .19 |

Table 9: [**Metaphor generation**] **User survey responses.** Overall, users perceived TextDavinci to be the most *helpful* and *satisfied* with the results from working with the model. However, users perceived Davinci to be the *easiest* to work with and were willing to *reuse* the model the most. Overall, perceived *helpfulness* of models and user *satisfaction* had a strong positive correlation ($r = 0.95$), while *ease* of interaction and users' willingness to *reuse* the system were also positively correlated ($r = 0.74$). The numbers indicate means and standard errors. For these outcomes, no results were found to have statistical significance using a Tukey-Kramer test.

❶ **Users are more willing to reuse models that require less effort.** Table 8 shows that users needed the least effort when working with Davinci, given that they took the least *time* and made the fewest *queries*. Despite the fewest number of queries, and therefore fewest suggestions, the acceptance rate for suggestions was the highest for Davinci and the final metaphorical sentences were rated most highly by third-party evaluators (Table 10). From these observations, we hypothesize that users could query a few times and quickly find acceptable sentences generated by Davinci. According to survey responses, users also perceived the same model as the *easiest* to interact with and were most willing to *reuse* the system when the model was used.

❷ **More satisfying user experiences may not correlate with the quality of outputs.** On the other hand, users found TextDavinci to be most *helpful* and *satisfactory* according to the survey responses (Table 9). However, third-party evaluators considered sentences written with the same model to be the worst among the four models (Table 10). Although it is hard to draw strong conclusions due to the relatively small size of users and third-party evaluators, one possible explanation is that users' value judgment for satisfaction and reuse may depend on different factors (e.g., helpfulness and ease, respectively) based on the context.

## 4 Related Work

We first review the evaluation of LM-based language generation systems while distinguishing the evaluation of model *completions* and *interaction traces*. Then, we briefly recount specialized interactive systems other than LMs. In addition, we provide related work for the five tasks we studied in this paper in Section B.

| Model | Apt | Specific | Imageable | Overall |
|---|---|---|---|---|
| | | (/100%) ↑ | | |
| TextDavinci | $75 \pm 4.0$ | $78 \pm 4.6$ | $75 \pm 4.6$ | $78 \pm 3.4$ |
| TextBabbage | $75 \pm 4.0$ | $79 \pm 3.7$ | $70 \pm 5.0$ | $78 \pm 3.0$ |
| Davinci | $90 \pm 3.3$ | $90 \pm 3.3$ | $83 \pm 5.3$ | $88 \pm 3.0$ |
| Jumbo | $77 \pm 5.5$ | $83 \pm 5.3$ | $72 \pm 5.7$ | $84 \pm 3.9$ |

Table 10: [**Metaphor generation**] **Third-party evaluation on the quality of metaphorical sentences.** The numbers indicate means and standard errors. These metrics were not found to exhibit any statistically significant differences using a Tukey-Kramer test at the $p = 0.05$ level. Conditioning on the use

**Evaluation of model completions.** Evaluation has a long history in NLP (see Spärck Jones & Galliers, 1995; Liberman, 2010). Traditionally, evaluations for generative systems have centered on specific tasks: for example, WMT for machine translation (WMT, 2006), TIPSTER SUMMAC for text summarization (Mani et al., 1999), Dialogue System Technology Challenges (DSTC) (Gunasekara et al., 2020) and Alexa Prize Challenges (Hakkani-Tür, 2021; Gottardi et al., 2022) for dialogue systems. More recently, efforts like GEM (Gehrmann et al., 2021) and GEMv2 (Gehrmann et al., 2022) have consolidated and standardized practices for natural language generation across many tasks.

For LMs, we have seen an analogous trend: LMs were initially evaluated for the probabilities they assigned to unseen corpora like the Penn Treebank (Marcus et al., 1999), One Billion Word Benchmark (Chelba et al., 2013), and WikiText-103 (Merity et al., 2016). However, as LMs have functioned as foundation models (Bommasani et al., 2021) underpinning myriad downstream systems, evaluation practices have shifted to consolidated evaluations for many downstream tasks (Wang et al., 2019a;b; Kiela et al., 2021; Liang et al., 2022). We emphasize that across all the benchmarks used to evaluate generation systems and/or LMs, the prevailing norm has been *non-interactive* evaluation with few notable exceptions (e.g., dialogue benchmarks).

**Evaluation of interaction traces.** While human-LM interaction has been studied within NLP in specific domains, such as dialogue (Hu et al., 2021; Thoppilan et al., 2022; Smith et al., 2022) and story generation (Akoury et al., 2020; Clark & Smith, 2021), these works only consider some aspects of user interaction. Namely, while they target the interaction process (e.g., dialogue history), they often foreground quality and third-party evaluations, with less consideration of preference and first-person experiences. In contrast, our evaluations cover a broader range of evaluation dimensions by working with richer user interactions (e.g., users' keystrokes to type in prompts, button clicks for querying systems, the appearance of a popup box for showing completions to users, and users' cursor movements to edit after getting completions) along with timestamps in *interaction traces*. In the field of Human-Computer Interaction (HCI), interaction traces have been used to reason about the helpfulness of completions (Roemmele & Gordon, 2018; Clark et al., 2018), the time that humans take to work with given completions (Buschek et al., 2021), and the language, ideation, and collaboration capabilities of the systems (Lee et al., 2022). Our goal is to study the process of human-LM interaction through interaction traces and compare interactive performance to non-interactive performance.

**Interactive systems.** Human interaction with user-facing intelligent systems has been explored in many disparate communities (see Bommasani et al., 2021, §2.5). Examples include work in information retrieval and search engines (Salton, 1970; Belkin et al., 1982; Kuhlthau, 1991; Ingwersen, 1992; Marchionini, 2006; Micarelli et al., 2007; White & Roth, 2009; Kelly, 2009; Croft, 2019), recommender systems (Goldberg et al., 1992; Konstan, 2004), voice assistants (Kamm, 1994; Cohen et al., 2004; Harris, 2004), human-robot interaction and assistive technologies (Hadfield-Menell et al., 2016; Mataric, 2017; Xie et al., 2020; Jeon et al., 2020), and broadly as a means for user creativity and expression (Fede et al., 2022; Lee et al., 2022). Relative to these works, we study how the unique aspects of LMs mediate user interaction with language generation systems.

When interaction has been consider in prior work, many works understand interaction to be a feedback signal for model improvement. Examples include work in the active learning (Settles, 2012) and language

learning (Wang et al., 2016) communities, with ter Hoeve et al. (2021) specifically proposing interactive language modeling to improve learning. In contrast to these learning-from-interaction settings, while we do consider the role of in-context learning in text summarization (Section 3.5), our focus is on the broader user experience with LMs (Yang et al., 2019).

## 5    Discussion

Benchmarking LMs with human users in interactive settings introduces both old and new challenges. In this section, we share the challenges and limitations we encountered while designing our tasks and systems as well as working with users. Given these experiences, we discuss potential solutions and paths forward.

### 5.1    General Challenges in Human Evaluation

**Cost.**    Compared to automatic evaluation, human evaluation requires costly human labors, which can be one of the biggest challenges for researchers to adopt human-centric evaluation approaches. Recently, there have been effort to leverage LMs to approximate human judgement, or more generally, simulate human behaviors (Liu et al., 2023; Dubois et al., 2023; Park et al., 2023). It would be interesting to study what kind of human behaviors can be modeled by LMs and how much coverage LMs can have over *diverse* human preferences in various contexts, which we believe is one viable, cost-effective way to extend this work. However, we emphasize that we need to do the evaluation with human users at least once to learn what the ground truth is and see which automatic metrics correlate with the real user evaluation.

**Reproducibility.**    According to Belz et al. (2023), many papers do not specify fundamental details such as the number of users, details of training, instructions, and set of model outputs, thereby severely hindering their reproducibility. We provide details on our methodology, data collection processes, and experimental design in Section 3 and C, and make our code and data publicly available at `https://github.com/stanford-crfm/halie`. We hope this information facilitates other researchers in replicating our study, building upon our findings, and doing additional analyses.

**Subjectivity.**    Human judgments are inherently subjective and can vary among different annotators (i.e., annotator bias). The interpretation of task instructions (i.e., instruction bias), evaluation criteria, and the perception of quality can differ, leading to inconsistencies in the decision-making process (Ethayarajh & Jurafsky, 2022; Parmar et al., 2023). One way to mitigate this variability is to perform within-subject studies. However, to compare four models, each user would need to repeat an experiment four times, which significantly increases the workload. For instance, each experiment (or "HIT" in Amazon Mechanical Turk) in text summarization asks a user to evaluate and edit a model summary 10 times; with four models, it will be 40 times (instead of 10), taking two hours (instead of 30 minutes) in total. Therefore, to streamline our experiments, we made use of between-subject studies as a practical choice, while recruiting a large number of users to reduce the effect of individual differences whenever possible and employing statistical analyses to ensure the robustness of our findings (Appendix D).

### 5.2    General Challenges in Interactive Evaluation

**Study design.**    User study design requires researchers to account for individual difference: we empirically observe significant heterogeneity in how users qualitatively experience and interact with LMs. Due to these individual effects, especially when the number of recruited users is not especially large, it is generally desirable for each user to interact with most/all models (i.e., *within-subjects* study design). Consequently, this would requires the set of models themselves to be decided upon in advance. This is in contrast to many non-interactive settings, where model selection and evaluation do not have much dependency on each other. If each user is expected to interact with multiple models, sequential effects need to be controlled for (e.g., through the randomization of model order across users). Otherwise, results may be subject to undesired confounding from the *novelty effect* (i.e., initial performance improvement due to the increased interest in the new technology) and *user adaptation* (i.e., performance improvement over time due to the increased familiarity of the technology). If possible, we recommend recruiting a large number of diverse users to

allow for more flexibility in selecting which models to evaluate and to alleviate concerns of sequential effects (e.g., by having each user only interact with one model).

**Latency.** The amount of time an LM takes to generate completions, or *latency*, can significantly influence human-LM interaction and how humans perceive models. Guidelines in HCI research recommend that interactive systems respond within 0.1 seconds to appear instantaneous and within 1 second to avoid interrupting the user's flow of thought (Miller, 1968; Card et al., 1991; Nielsen, 1994). The four models in our experiments had similar latency in an acceptable range: on average, it took 0.12s for `TextDavinci`, 0.12s for `TextBabbage`, 0.36s for `Davinci`, and 0.48s for `Jumbo` to generate a completion. With these models, we did not find any significant preference toward faster models. However, we did have to exclude some models that were simply too slow at the time of experiments (e.g., 50x slower than `Davinci`). Meeting these latency standards can be a challenge, exacerbated by the growing scale of existing LMs. With that said, the observed latency can be largely determined by the factors beyond model scale alone (e.g., allocated compute resources, query optimization, API support) as observed by Liang et al. (2022, §4.9). Overall, we underscore that latency is crucial to positive user experience for deploying human-LM interaction in the real world.

**Harms.** LMs are prone to generating toxic, biased, or otherwise undesirable text (Gehman et al., 2020). When users are exposed to this text via interaction, this can cause serious psychological harm. In our experiments, we observe that toxic content can be elicited by seemingly innocuous prompts, even for instruction-tuned models designed to discourage this behavior. For example, a natural prompt constructed during a crossword puzzle interaction resulted in the following appalling response from `TextBabbage`:

```
User:  What is a young pigeon called?
System:  A young pigeon is called a n****.
```

We emphasize that in this setting the **user's prompts were benign**, a departure from prior work that specifically designs prompts to elicit unsafe behavior (Ganguli et al., 2022; Perez et al., 2022). More surprisingly, it was extremely easy to reproduce the completion even with slightly different prompts and decoding parameters (with `TextBabbage` as of June 2022). In our case, where the primary goal is to evaluate LMs, striking the right balance between comprehensive evaluation and stringent safeguards poses a unique challenge. While our experiments include a mild measure (i.e., blocklist) to mitigate potential harms, we acknowledge that a more extensive approach could be pursued in future work, and should be pursued for productionized applications (Kirk et al., 2022). Given the evolving nature of both NLP research and societal considerations, continued exploration of enhanced safeguards should be a collective effort within the community.

### 5.3 Limitations Specific to Our Experiment Design

**Task selection.** In this work, we selected five tasks to highlight different aspects of interaction. These tasks range from goal-oriented to open-ended and cover the common usages of LMs reported by Ouyang et al. (2022, Table 1), such as generation (45.6%), open QA (12.4%), brainstorming (11.2%), chat (8.4%), and summarization (4.2%). However, given the generality of LMs, there are a vast number of tasks and domains (e.g., story generation, copywriting, etc.) that can be considered. One could also imagine taking any non-interactive task and turning it into an interactive version (similar to what we did with question answering), or creating non-traditional tasks to explore new ways of using LMs (similar to what we did with metaphor generation). While we conducted extensive experiments and evaluations on the selected tasks as case studies to demonstrate the usefulness of our framework, future work can further extend it to other tasks and novel ways of interacting with LMs.

**User interface design.** Designing user interfaces (UIs) with a strong focus on user needs and preferences is essential to creating positive user experiences. Prior work has shown how design elements (e.g., who initiates interaction, where suggestions are presented, and how many suggestions are shown at a time) can have significant impact on the dynamics of human-LM interaction (Clark et al., 2018; Buschek et al., 2021; Ippolito et al., 2022; Dang et al., 2023). In our experiments, however, we decided to follow the most standard design and practice for UI design whenever possible and leave the exploration of UIs to future

work. Concretely, we used an existing dialogue interface (Paranjape et al., 2020) and crossword puzzles interface from *Down for a Cross* (an open-source software for the multi-player crossword puzzles) almost as is, while minimally adopting the interface from Lee et al. (2022) for question answering, text summarization, and metaphor generation. Before user studies, we tested out our interfaces with small in-person pilots and iterated on the design, mainly to resolve any confusion users had with the task and functionality. By focusing on the performance gap between LMs given the same interface, we were able to reason about the difference between their non-interactive performance and interactive performance. Future work may investigate how to design UIs that better leverage these strengths, and perhaps narrow down the gap of the different models through the design.

**Model selection.** The choice of LMs in our experiments is susceptible to rapid technological advancements. Since we conducted our experiments, there have been many new models over the last year. In light of these newer models, we acknowledge the nuanced nature of our findings. Particularly, as chat interfaces have grown significantly in popularity, this dominance has prompted adaptations that prioritize interactive aspects, resulting in a more aligned optimization goal than before. Consequently, we speculate that the gap between non-interactive and interactive performance might have been partially bridged due to these adaptations. Future work can investigate the implications of these newer models on our research questions and further contextualize our findings in this ever-changing landscape. Regardless, we believe the validity of our research questions and conclusions remain, and the significance of our work lies in its ability to offer a conceptual framework and insights that extend beyond the specific models under consideration.

**System logic design.** Similar to UI design, the choice of system logic undoubtedly influences the nature of the human-LM interaction for any given LM. In particular, recent work has shown that prompts and even the order of examples presented in the prompts can significantly affect the performance of LMs (Brown et al., 2020; Wei et al., 2022; Lu et al., 2022). In this work, we chose a simple and generic prompt per task and used it for all models, which corresponds to `CreatePrompt` in Algorithm 1 (see Section 3 for exact prompts we used). Likewise, we chose a set of decoding parameters per task and applied it to all models, which corresponds to `QueryLM` in Algorithm 1. Before in-person pilots and user studies, we tested out the four models with the chosen prompts and hyperparameters ourselves to ensure our selection did not unfairly penalize specific models. However, given the sensitivity of these models, we expect that different prompt design and decoding parameter choices can result in different outcomes and interaction behaviors. Thus, there is also ample opportunity to explore other system logics and understand their impacts on human-LM interaction.

**User recruitment.** Due to the large number and size of user studies we ran, we recruited crowd workers from Amazon Mechanical Turk to recruit users in a timely and scalable manner (see Section C for details on crowdsourcing). This may have influenced the results due to a potentially uniform level of expertise among users, and more importantly, different incentives compared to those of actual users in the wild. Therefore, it is important to take into account that the findings presented here may not directly generalize to actual users. Moreover, each task's user study was designed independently, resulting in some tasks having between-subjects study design while the others having within-subjects study design. These decisions were made based on the design of each HIT (Human Intelligence Task; a single, self-contained, virtual task) which determined expected time commitment and difficulty of the task for users, thus affecting our recruitment strategy. We did not prefer either type of study design and acknowledge that both have associated strengths and weaknesses. Whenever possible, we tried to recruit a large number of users (especially for social dialogue, question answering, and crossword puzzles) and performed random assignment of users to groups to provide some assurance that the results are well-calibrated across different groups.

## 5.4 Future Work

**Accommodation.** One important aspect of human-AI interaction to account for is user *accommodation*, or the ability of users to adapt their behavior as they learn more about the strengths and weaknesses of a system and the underlying model. For QA and crossword puzzles, we asked users to answer the survey question "Did the way you chose to interact with the AI Teammate change over time? If so, how?." Oftentimes,

user accommodation reflected underlying properties of the models—for example, in QA, prompts phrased as questions yield successful outputs mainly from `TextBabbage` and `TextDavinci`, so users assisted with `Davinci` or `Jumbo` were more likely to switch to declarative, "fill-in-the-blank" style prompting strategies. Additionally, for some scenarios, we studied user accommodation quantitatively by either measuring how the edit distance or query strategies of user prompts changed over time. We found that task framing can also affect accommodation—while for QA, users engaged in more reformulation over time and became less inclined to copy the provided question verbatim, for crossword puzzles, users became more likely to copy clue text over time, perhaps due to later relying on the LM to understand unfamiliar clue texts. Because of these results, we believe future work should closely study more long-term interactive scenarios, where the time it takes a user to develop familiarity with a tool is outweighed by the overall interaction period.

**Lasting impact.** In our work, the emphasis is on the short-term interaction of users with LMs: we evaluate this localized experience. However, human-LM interaction, and human-AI interaction more broadly, can have more lasting and longitudinal impacts on users. In the context of LMs specifically, we expect interactions with LMs may influence human writing practices, opinions and beliefs, broader perception of language technology and AI systems, and potentially many other aspects of human experience that are mediated by language. That is, we expect these effects could persist to environments where LMs are not present. Existing evidence includes the works of Jakesch et al. (2023), which finds interacting with an opinionated language model influences written opinions and reported attitudes, and Wenker (2022), which finds that machine agency affects the content users write. More established evidence for other language technologies, such as search engines, shows users' knowledge, beliefs, and attitudes can be significantly altered by sustained interaction with these technologies (Allam et al., 2014). Overall, especially given the rapid deployment of LMs, we recommend that future work should actively monitor how these interactions come to affect human language practices (e.g., writing, reading, listening, speaking), culture, well-being, and broader society.

## Acknowledgements

We thank Eric Horvitz for the valuable feedback on the paper, Lama Ahmad and Miles Brundage from OpenAI, and Opher Lieber, Barak Lenz, Dan Padnos and Yoav Shoham from AI21 Labs, for their support and for providing credits to evaluate their models. We thank the Stanford Center for Research on Foundation Models (CRFM) and the Institute for Human-Centered Artificial Intelligence (HAI) for support throughout this project.

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

# A  Author contributions

This project was a team effort, built on countless contributions from everyone involved. To enable future inquiries to be directed appropriately, we listed contributors for each part of the paper below.

**Amelia Hardy** (Social dialogue): Led the social dialogue team through the design, data collection (in-person pilot and crowdsourcing), and analysis. Helped with the third-party human evaluation of metaphor generation.

**Ashwin Paranjape** (Social dialogue): Contributed to the overall framing of the project. Implemented the system and helped with the quantitative analysis of social dialogue.

**Esin Durmus** (Text summarization, Social dialogue): Led the text summarization team through the design, implementation, data collection (in-person pilot and crowdsourcing), analysis, and third-party human evaluation. Helped with running pilots for social dialogue.

**Faisal Ladhak** (Text summarization, Social dialogue): Helped with the text summarization team through the design, implementation, data collection (in-person pilot) and analysis. Helped with running pilots and implementing scenarios for social dialogue.

**Frieda Rong** (Metaphor generation): Led the design and data collection (in-person pilot).

**Hancheng Cao**: Managed AMT account and communication with crowd workers.

**Ines Gerard-Ursin** (Social dialogue): Helped with running in-person pilot and implementing scenarios for social dialogue. Ran statistical analysis for all teams.

**John Thickstun** (Question answering): Contributed to the overall framing and writing process of the project. Helped with the design, implementation, data collection (in-person pilot), and analysis for question answering. Led the writing process for the question answering team, and helped with the writing process for crossword puzzles and related work. Helped with the third-party human evaluation of metaphor generation.

**Joon Sung Park** (Social dialogue): Contributed to the framing and qualitative analysis of social dialogue.

**Megha Srivastava** (Question answering, Crossword puzzles): Contributed to the overall framing of the project. Led the question answering and crossword puzzles teams through the design, implementation, data collection (in-person pilot and crowdsourcing), and analysis.

**Michael Bernstein**: Provided overall guidance on framing the project.

**Mina Lee** (All teams): Led and managed the overall project. Led the overall framing and writing process of the project. Standardized data and reproduced analyses for all teams. Implemented the systems for question answering, text summarization, and metaphor generation. Helped with analysis for text summarization and metaphor generation. Helped with the third-party human evaluation of metaphor generation.

**Minae Kwon** (Question answering): Helped with the design, implementation, and data collection (crowdsourcing) of question answering.

**Percy Liang**: Provided overall guidance on the project. Contributed to the overall framing of the project.

**Rishi Bommasani**: Provided overall feedback on the project. Helped with the writing process for related work.

**Rose E. Wang** (Question answering): Helped with the design, implementation, data collection (crowdsourcing), and quantitative analysis of question answering.

**Tony Lee**: Supported with the use of language models via Stanford CRFM API.

**Xiang Lisa Li** (Metaphor generation): Led the implementation, data collection (crowdsourcing), third-party human evaluation, and analysis of metaphor generation.

# B  Related Work

## B.1  Social Dialogue

Dialogue systems leverage the power of natural language to promise a highly natural mode of interaction with machines in a wide range of domains ranging from information retrieval, transactions, entertainment, and even informal chit-chatting (Glass, 1999). LMs pose an interesting opportunity to boost the development of such systems given their generative capacity and the breadth of knowledge encoded in them that could

enable dialogue systems to be able to react even in open domains. Although we do not evaluate them in this paper, due to time and technical constraints, recent models such as BlenderBot (Shuster et al., 2022) and ChatGPT (OpenAI, 2022) demonstrate significant advancements in this field. Still, achieving a successful dialogue between the users and machines is difficult owing to the frequent rise of unmatched expectations of the systems' capabilities (Clark et al., 2019) and the difference in conversational styles between the interlocutors (Thomas et al., 2020). Additionally, models trained on human-human conversations learn to replicate not only desirable, but also undesirable patterns of behavior (Xu et al., 2020). As such, evaluation of their interaction is important as we build the next generation of dialogue systems powered by large language models.

Our evaluation of dialogue systems is focused on open-domain contexts, in which the conversational agent is tasked with continuing a dialogue with the user in a hypothetical scenario (Langevin et al., 2021; Thoppilan et al., 2022; Smith et al., 2022). Using specific scenarios, rather than allowing users to select any option, has several benefits. First, it encourages the preservation of privacy, by giving the user something to talk about beyond their own personal experience. Second, it enables greater standardization of results, since variance due to selected topic is reduced (Ji et al., 2022). Third, using particular scenarios is consistent with past work in dialogue evaluation (Smith et al., 2022). Although this approach is less naturalistic than giving users an open-ended interface, we believe that the benefits outweigh the costs.

However, there is a lack of standard approach for evaluating dialogues in such contexts (Deriu et al., 2021; Huang et al., 2020; Roller et al., 2021). It is difficult to standardize dialogue evaluation, since for any context, there are many possible appropriate responses (Ji et al., 2022). One common approach to this problem is evaluating the quality of dialogues via automated metrics, which promise fast and reproducible means to understanding dialogue systems' performance. However, a growing literature suggests that such metrics can be limiting in capturing the breadth of possible interactions that could happen in an open domain dialogue (Liu et al., 2016). Additionally, they have been shown to have little correlation with human judgments (Liu et al., 2016; Ji et al., 2022). Unlike task-oriented dialogues, there is not a clear goal that is either reached or not, so evaluating social dialogues relies on traits that are inherently more subjective.

Prior work employed crowd workers as well as experts (e.g., a group of researchers in the same institution as where the study took place, or those trained in dialogue evaluations (Deriu et al., 2020 2020; 2021) to evaluate the quality of dialogue systems, with each group of evaluators offering their unique strengths. Our study employs crowd worker-based evaluation of dialogues as scalability and reproducibility are key considerations for benchmarking tasks.

## B.2 Question Answering

Question answering is a popular task within NLP that is often evaluated with large, non-interactive datasets of question-answer pairs, sometimes accompanied with additional context Rajpurkar et al. (2016); Joshi et al. (2017). The most closely related work to ours is Bowman et al. (2022), which evaluates LM-assisted QA on the MMLU dataset. Whereas Bowman et al. (2022) evaluate interactions between a single LM and a small group of five well-trained users, we consider interactions between four different LM's and hundreds of users. In addition to replicating the finding of Bowman et al. (2022) that human-LM interaction generally outperforms a human or LM alone, we are able to observe statistically significant patterns in the relative performance of human interactions with different LM's. Other recent efforts to design interactive evaluation for question answering, such as Kiela et al. (2021); Bartolo et al. (2020), largely aim to reduce test-set overfitting by asking humans to adversarially identify edge cases, rather than considering how they would naturally choose to interact with models.

To account for more naturalistic ways humans use these systems, researchers have built datasets from actual human-generated search queries, such as MS-MARCO (Nguyen et al., 2016) and NaturalQuestions (Kwiatkowski et al., 2019), thereby capturing a more realistic distribution of the information users would naturally seek from an intelligent assistant. However, while these datasets are useful for measuring model performance in the context of realistic user prompts for potential downstream applications, they do not provide information about the quality (e.g., reported satisfaction) of a user's interaction over multiple queries. Furthermore, evaluations on such datasets do not account for the ability of humans to adapt, such as changing

their query styles over time (i.e., query reformulation) or adjusting how they trust and use system outputs in decision-making.

Query reformulation refers to the iterative process of expressing our requests differently to improve system utility, and arises commonly in the context of search engines; for example, Teevan et al. (2007) found that up to 40% of Yahoo search log queries in a year were reformulations. A significant body of research from the information retrieval community focuses on deriving taxonomies for query reformulation based on analyzing search logs, including lexical-based (e.g., adding words) and more general reformulation strategies (e.g., specialization) (Huang & Efthimiadis, 2009; Lau & Horvitz, 1999). These works have inspired several methods for automatic query reformulation for inferred user goals, including using context-sensitive term-substitutions and reinforcement learning to optimize for retrieved document recall (Wang & Zhai, 2008; Nogueira & Cho, 2017).

Further research has focused on varying querying habits across time and user populations. For example, Pang & Kumar (2011) found, perhaps surprisingly, that the use of natural language question-queries (e.g. starting with "What") has increased over time, despite requiring more user effort, perhaps due to an increase in novice users unfamiliar with formulating keyword-based queries White et al. (2015). Additional research has also demonstrated a strong effect of a system's interface on users' query formats (Spink & Ozmultu, 2002), and that long-term users are more likely to interact longer with a search engine during a session with more complex queries Liu et al. (2013). While effective querying strategies for large language models (e.g., "prompt hacking") have been popularly discussed, to the best of our knowledge we are the first to study these questions in the context of real user data from an LM-based interactive system.

Finally, our task allows us to study how the lack of factuality and faithfulness of outputs from different LMs influence user trust and behavior Jacovi & Goldberg (2020). Several recent works on interpretability and human-AI collaboration have studied the effects of model explanations on human decision-making in classification settings Lakkaraju et al. (2019) finding that explanations that make model outputs appear more justifiable can make users trust even incorrect outputs more (Bansal et al., 2021b). Furthermore, they found that human-AI collaborative contexts place an additional cost of verification of model outputs on users Bansal et al. (2021a) that, if needed frequently, may discourage future model use. Our task allows us to assess the degree to which efforts to align LMs to human preferences (Ouyang et al., 2022), as well as factors such as fluency and tone, can impact user trust over time in a collaborative task.

### B.3 Crossword Puzzles

Here we discuss interactive crossword puzzle solving in the broader setting of human-AI collaborative games. We refer readers to Section B.2 for related work on shared aspects with the interactive question answer task.

Solving crossword puzzles has long been a challenging task for designing more complex AI systems (Ginsberg, 2014; Littman et al., 2002; Wallace et al., 2022; Rozner et al., 2021). However, the increased success of modern text generation models and their ability for interaction with *lay*-people has sparked further development in several AI-based language games, including text-adventure games and AI Charades (Urbanek et al., 2019; Frans, 2021). Cooperative games can help highlight differences between AI-AI and human-AI benchmarking, as shown in Chattopadhyay et al. (2017). Furthermore, they differ from other human-AI collaborative contexts as the goal of the collaboration is not simply completion of a task, but also a general sense of user engagement and enjoyment, which in turn depend more on more abstract properties of LM outputs. For example, players of the text adventure game in Ammanabrolu et al. (2019) found that the creativity of an AI-generated game setting was anti-correlated with its coherence.

Human-AI collaborative games also serve as a rich test-bed for understanding human's mental models of an AI system; for example, humans performing poorly in the cooperative word guessing game in Gero et al. (2020) tended to *overestimate* the knowledge capabilities of the collaborating AI agent; another user study found that users were more likely to believe their teammate was an AI if it failed to adapt their behavior (Ashktorab et al., 2020). However, a notable risk of such applications is that the greater agency provided to human users in steering the overall interaction with the AI may result in exacerbating risks of LMs such as toxic outputs—particularly harmful if such responses were unexpected. One notable example is AI Dungeon

(Hua & Raley, 2020), where player inputs resulted in sexually inappropriate outputs involving children.[6] We observe similar behaviors in our system, as described in the results (Section 3.4).

### B.4 Text Summarization

Text summarization is a well established research direction in NLP (Mani, 1999; Spärck Jones, 1999; Nenkova & McKeown, 2012), where the goal is to compress the salient information in the source document into a coherent and fluent summary (Peyrard, 2019). The advent of pre-trained large LMs has led to dramatic improvements in summarization capabilities, particularly in terms of coherence and fluency (Lewis et al., 2020; Zhang et al., 2020). However, generated summaries are far from perfect and tend to contain information that is not supported by the original document (Cao et al., 2018; Durmus et al., 2020; Maynez et al., 2020; Pagnoni et al., 2021).

Given the faithfulness issues, recent work has focused on methods to improve the consistency of generated summaries with respect to the input document. Prior work has largely focused on improving fine-tuned models with approaches such as reducing noise incorporated from training data (Nan et al., 2021b; Kang & Hashimoto, 2020; Goyal & Durrett, 2021), adding losses aimed at improving consistency (Nan et al., 2021a; Cao & Wang, 2021), and learning discriminators to select from set of candidate summaries Chen et al. (2021); Ladhak et al. (2022). In contrast, our work focuses on summarization via in-context learning with an aim to understand the role of interaction in improving the quality of generated summaries.

Interactive summarization has been studied by prior work in the context of multi-document summarization, where users interact with the system to query for additional information to expand upon an initial summary (Shapira et al., 2021; 2022; Avinesh et al., 2018). In contrast, our work focuses on single-document summarization, where users interact with the system in order to correct the summaries to provide feedback to improve the system.

Our work is closely related to a line of recent work that looks at improving the performance of LMs through human feedback by fine-tuning LMs using reinforcement learning (Ziegler et al., 2019; Böhm et al., 2019; Stiennon et al., 2020; Ouyang et al., 2022) or natural language feedback (Campos & Shern, 2022). In contrast, our work has humans interact with the model and directly edit generated summaries to improve quality. Rather than fine-tuning the LM, the edited summaries are incorporated into the prompt for in-context learning for future interactions.

### B.5 Metaphor Generation

Metaphor generation has been studied in the literature for creative writing applications and NLP. Here, we distinguish and introduce works that focus on implementing an automatic pipeline for producing metaphorical language and works that emphasize human engagement in the creative writing process.

Computational approaches to generating metaphorical language build off of conceptual metaphor theory Lakoff & Johnson (1980), which describes metaphors as mappings between source and target concepts. Approaches to generate metaphorical language can operationalize this framework by training LMs to replace literal phrases in an originally non-metaphorical sentence with figurative ones from the source (Chakrabarty et al., 2021; Stowe et al., 2021b;a), or by searching for connections from the source to the target concept out of a pre-existing knowledge graph of concepts (Gero & Chilton, 2019). More recent works compare the use of pipelines that focus on specific keywords to our strategy of few-shot prompting for generating figurative language in the context of puns (Mittal et al., 2022) and find that decomposing the generation task improves successful generation, although surprisingly the few-shot only approach achieves the highest coherency.

Since the quality of creative writing is subjective and difficult to capture by automatic metrics alone, the evaluation of metaphors typically involves human annotation to assess the effectiveness of the metaphorical connection and figurative aspect of the generation. For instance, works related to lexical substitution or paraphrasing assess *metaphoricity* to determine how well the modified sentence evokes figurative imagery, while checking that the intended meaning is present (Chakrabarty et al., 2021; Stowe et al., 2021b;a), whereas

---

[6]https://www.wired.com/story/ai-fueled-dungeon-game-got-much-darker/

Gero & Chilton (2019) consider more precisely how well the identified metaphorical connection fits the seed metaphor, or mapping, in both *aptness* and *specificity*.

The use of computational metaphor generation to support creative writing has been explored in Gero & Chilton (2019), where users could try out different seed metaphors for a target word and use the system to generate metaphorical connections to inspire their own writing. In such settings, the diversity that the system enables in cooperation with the human user is important: "creative writers do not want tools that will make their writing sound the same as others" (Gero & Chilton, 2019). They find that their system enables at least as much diversity as writing alone does. Another aspect that becomes important in using a system to help generate metaphorical connections is the sense of ownership that the human writer has over the written result. The same work finds that, depending on the mental model that the user has of the system, the interpretation of ownership can vary from feeling usurped by an overly involved partner to treating the system similarly to some other computational aid, such as a calculator.

## C   Crowdsourcing Details

Our experiment was reviewed and approved by the institution's Institutional Review Board (IRB). Before the study, all participants read and agreed to a consent form that included risks, payments, participants' rights, and contact information.

**Qualification.**   We recruited participants through Amazon Mechanical Turk, where we required that they had a HIT approval rate greater than 97%, were located in the United States, and had greater than 10000 approved HITs.

**Payment.**   The payment was determined to be around $18 or more per hour, which exceeds the minimum wage in the United States. For data collection, we paid $3 for social dialogue (10 minutes), $5 for question answering (15 minutes), $10 for crossword puzzles (30 minutes), $10 for text summarization (30 minutes), and $4 for metaphor generation (10 minutes). For third-party evaluation, we paid $0.35 for evaluating one document and summary pair on three criteria and $0.15 for evaluating one metaphorical sentence on three criteria.

### C.1   Social Dialogue

**Instructions.**   We provided users with both general and dataset-specific instructions.

```
General instructions

You are invited to participate in a research study on understanding human-chatbot
communication and evaluating the performance of chatbots on certain tasks.  Our goal is
to learn how to build chatbots that can talk to people naturally, much like another
human being does.  You will chat with a chatbot for around 10-15 minutes.  The chatbot
is designed to discuss a range of social topics (for example music or travel) and you
are encouraged to converse with it naturally, as you would with a friend.  For example,
you may ask questions or introduce your own topics.  Once you have completed your
conversation, you will be asked to answer feedback questions about the interaction.
Data from responses and chatlogs will be analysed in deidentified format and extracts
edited to preserve confidentiality may be featured in any published work resulting out
of the study.

CommonsenseDialogues task instructions

When you start the chat, you will be given a scenario to discuss.  For example, you may
have the scenario "I lost my keys this morning.  It was very stressful." During the
conversation, please talk as though you have experienced the scenario given.  For this
scenario, you might say something like "I had such a stressful morning!  I couldn't
find my keys anywhere." The goal of this task is to evaluate how well the chatbot
understands common social scenarios.
```

```
EmpatheticDialogues task instructions

When you start the chat, you will be given a scenario to discuss.  For example, you may
have the scenario "I lost my keys this morning.  It was very stressful." During the
conversation, please talk as though you have experienced the scenario given.  For this
scenario, you might say something like "I had such a stressful morning!  I couldn't
find my keys anywhere." The goal of this task is to evaluate how well the chatbot
communicates empathetically.
```

**Interface.**   See Figure 3.

## C.2   Question Answering

**Instructions.**   Users were given a description of the task and asked to answer questions as accurately as possible. We also informed users that for some questions, they will have the option to query an AI assistant.

Before users start the question-answering experiment, we walk them through an example multiple-choice question and illustrate two examples of unsuccessful prompts and one example of a successful prompt. These examples were created from initial pilot experiments. The example question we use is:

```
What is another name for the camelopard?

A. Circus
B. Giraffe
C. Cantaloupe
D. Oasis
```

The unsuccessful examples illustrate how prompts result in inconclusive answers from the model. The first unsuccessful example is:

```
Prompt:  What is another name for the camelopard?

Completion:  A: The camelopard is a cross between a camel and a leopard.  The
camelopard is a cross between a camel and a leopard.
```

The second unsuccessful example is:

```
Prompt:  Another name for the camelopard is

Completion:  the "camel-cat."  which is a more common name in the United States.
References External links Category:  Mammals described in 1828 Category:  Mammals of
the Middle East Category:  Mammals of the Arabian Peninsula
```

The successful example illustrates how a prompt results in a conclusive answer from the model for answering the question. We used the following example:

```
Prompt:  Circus, Giraffe, Cantaloupe, Oasis.  Another name for camelopard is

Completion:  giraffe.  In the animal kingdom, giraffes are the tallest land animals.
```

To encourage diversity in prompting, we emphasize that users need not follow the example prompts we outlined and are free to prompt the assistant in whatever manner they choose. We also remind users not to use Google or another search engine and not to switch tabs during the experiment.

**Interface.**   See Figure 14.

### C.3 Crossword Puzzles

**Instructions.** We provided users a description of how to interact with the AI Teammate and overall crossword puzzle interface, as well as examples of prompts they could provide to assist with solving puzzle clues. As in the QA task, these examples were drawn from pilot studies. We include an image of the full instructions provided to crowdworkers in Figure 15, which users were able to access at any point from the crossword interface.

**Interface.** See Figure 7.

### C.4 Text Summarization

**Instructions.** For the main study, we provided users a description of how to interact with the text summarization system before presenting the system interface. Concretely, the instructions included the following seven steps.

```
1. When you click the link below, you will be given a text editor with a document and
   AI-generated summary.
2. First, rate whether the machine generated summary is consistent, coherent and
   relevant.  The descriptions for these aspects are provided in this link.
3. Once you're done rating the machine generated summaries, click "Next" button which
   will allow you to edit the generated summary to improve its consistency, coherence
   and relevance.
4. After editing the summary, click "Next" button to evaluate the edited summary.
   Rate whether the edited summary is consistent, coherent and relevant.
5. You will repeat the steps above for 10 article-summary pairs.  Then, at the end of
   your session, you will be shown a verification code.
6. Copy your verification code below.
7. Finally, answer the survey questions below to complete the task.
```

For the third party evaluation, the following description was provided.

```
In this task, we need your help in evaluating machine generated summaries.  In the
below table, a news article (top) and a summary generated by an AI system (below) are
shown.  Please evaluate the (1) consistency, (2) relevance, and (3) coherence of the
summary based on the article.

(1) We ask you to give a binary decision about whether the summary is consistent with
the new article.  We define a summary to be consistent if the all the information
expressed by the summary can be inferred from the new article.  We now give you two
examples where the first example is inconsistent and the second example is consistent.
For illustration purpose, we omit parts of the article.

Here is an example where the summary is inconsistent because the summary mistakenly
summarizes that the person died on March 5, 1898 when they were actually born on March
5, 1898:
```

- ```
  Article:  The world's oldest person has died a few weeks after celebrating her
  117th birthday.  Born on March 5, 1898, the great-grandmother had lived through two
  world wars, the invention of the television and the first successful powered
  aeroplane flight by the wright brothers [...]
  ```
- ```
  Summary:  The world 's oldest person has died on March 5, 1898.
  ```

```
Here is an example where the output is consistent because all information can be
inferred from the news article:
```

- ```
  Article:  Andy Murray [...]  is into the semi-finals of the Miami Open, but not
  before getting a scare from 21 year-old Austrian Dominic Thiem, who pushed him to
  4-4 in the second set before going down 3-6 6-4, 6-1 in an hour and three quarters.
  [...]
  ```

- Summary: Andy Murray defeated Dominic Thiem 3-6 6-4, 6-1 in an hour and three quarters.

The summary should only contain information from the original article. If the information presented in the summary is true based on general knowledge (e.g. the earth is round) but is not supported by the original article, you should mark it as unfaithful. It is okay for the output to have minor grammatical errors. If you can understand what the output expresses despite the minor grammatical errors and if the information is supported by the article, select consistent. If the output is nonsensical, select inconsistent.

(2) We ask you to judge whether the summary is **relevant** to the news article on a 1 to 5 scale. The summary should include only important information from the source document. If the summary contains redundancies and excess information, you should consider it as less relevant.

Here is an example where the summary is not very relevent because the information summarized is not the main point of the article.

- Article: The world's oldest person has died a few weeks after celebrating her 117th birthday. Born on March 5, 1898, the great-grandmother had lived through two world wars, the invention of the television and the first successful powered aeroplane flight by the wright brothers [...]
- Summary: The first successful powered aeroplane flight was by the wright brothers.

(3) We ask you to judge whether the summary is **coherent** on a 1 to 5 scale. A good summary should organize the relevant information into a well-structured summary. The summary should not just be a heap of related information, but should build from sentences into a coherent body of information about a topic.

Here is an example where the summary is not very coherent because the two sentences don't flow well together.

- Article: The world's oldest person has died a few weeks after celebrating her 117th birthday. Born on March 5, 1898, the great-grandmother had lived through two world wars, the invention of the television and the first successful powered aeroplane flight by the wright brothers [...]
- Summary: The world's oldest person has died at the age of 117. The first successful powered aeroplane flight was by the wright brothers.

**Interface.** See Figure 17.

### C.5 Metaphor Generation

**Instructions.** Users were given a description of the task, examples of good metaphorical sentences for three criteria, and instructions for the metaphor generation system. The task and examples were provided as follows:

Goal: Given a seed metaphor, your goal is to write as many metaphorical sentences as possible (at least 5) in 10 minutes with the help of the AI-powered system!

Examples: Consider a seed metaphor in the form of "[source] is [vehicle]", such as "Love is a storm".

Good metaphorical sentences meet the following three criteria:

Apt: Describe a connection between the concepts. The connection between source's feature and vehicle's feature (e.g., both love and storm can come unexpectedly) should be sound.

- Strong example: Love can come on unexpectedly.
- Weak example: Love is a weather event.

```
Specific:  Describe a connection unlikely to be transferable to other concepts.  The
vehicle's feature (e.g., lasting through the night) should be specific enough to source
(i.e., love) and vehicle (i.e., storm).
```

- Strong example:  Love can last through the night.
- Weak example:  Love is dark.

```
Imageable:  Evoke visual.  It should be easy to visualize a scenario from the sentence.
```

- Strong example:  Love can rain down on our heads.
- Weak example:  Love can scare people.

```
We acknowledge that it might be difficult to satisfy all of the criteria for every
metaphorical sentence.  Therefore, we ask you to do your best to meet most of them
while writing sentences, but it is okay not to satisfy some of them.  However, if all
of your sentences fail to meet these criteria, we reserve the right to reject your HIT.
```

Instructions were provided right above the "Click here to start writing" button to guarantee that users are aware of and understand the task before proceeding to interacting with the system.

```
1. When you click the link below, you will be given a text editor with a seed
   metaphor.
2. In 10 minutes, write as many metaphorical sentences as possible that express the
   seed metaphor.  The timer will start once you start writing.  If you want, you can
   write more than 10 minutes.
3. While writing, you can write on your own from scratch or get suggestions from AI.
     • To get suggestions from AI, click the "get suggestions" button or press the tab
       key, then our AI-powered system will show suggestions (that continue your input
       text if provided).
     • If you do not like any of them, just press the tab key again to get a new set of
       suggestions.
     • If you like a suggestion, click it to add this suggestion to your text!
     • You can also use up/down arrow keys to navigate the suggestions and press the
       enter key to add one to your sentences.
     • Revise your text or suggestions to meet the aforementioned criteria.
4. When you finish writing one metaphorical sentence, click the "add sentence" button.
   Then, write another metaphorical sentence and repeat the process.
5. Once the time is up, click the "finish session" button which will give you a
   verification code.  Copy and paste the code into the survey.
```

For the third party evaluation, the following instruction was provided along with the same set of examples for three criteria.

```
In the table below, you will see a seed metaphor and a metaphorical sentence that
attempts to express the seed metaphor.  Please evaluate whether the metaphorical
sentence is apt, imaginable, and specific with respect to the seed metaphor.  We
consider metaphors of the form "[source] is [vehicle]" (e.g.  Love is a storm).
```

**Interface.**  See Figure 18.

# D   Statistical Analysis

## D.1   Methods for statistical analysis

For the main results presented in the paper, we computed group-wise means and standard errors and used the modified Tukey-Kramer post-hoc test to account for unequal group sizes. For these calculations, we used both the Python scipy and R stats packages Virtanen et al. (2020); R Core Team (2020).

| Model | Sensibleness | Specificity | Humanness (/100%) ↑ | Interestingness | Inclination | Reuse (/5) ↑ |
|---|---|---|---|---|---|---|
| TextDavinci | 94 ± .03 | 82 ± .03‡ | 36 ± .05 | 37 ± .05 | 91 ± .03 | 4.09 ± .23 |
| TextBabbage | 84 ± .03 | 81 ± .04 ‡ | 28 ± .05 | 29 ± .05 | 88 ± .03 | 3.35 ± .24 |
| Davinci | 89 ± .02‡ | 92 ± .02 ‡ | 24 ± .04 ‡ | 27 ± .04 ‡ | 91 ± .02‡ | 3.8 ± .17‡ |
| Jumbo | 85 ± .03 | 83 ± .04 | 25 ± .05 | 31 ± .05 | 86 ± .03 | 3.21 ± .24 |

Table 11: [**Social dialogue**] **Model performance based on user survey responses.** Metrics are denoted by ‡ if models had a significant effect relative to Davinci, at the Bonferroni-corrected significance level of $p = .0125$. The numbers indicate averages and standard errors.

| Model | Time (min) ↓ | Queries (#) ↓ | Accuracy (/100%) ↑ |
|---|---|---|---|
| TextDavinci | 0.77 ± 0.69 | 0.83 ± 0.08‡ | 58 ± 2 ‡ |
| TextBabbage | 0.9 ± 0.73 | 1.14 ± 0.09 | 51 ± 2 |
| Davinci | 0.98 ± 0.51 | 1.1 ± 0.06‡ | 49 ± 2 |
| Jumbo | 1.89 ± 0.73 | 0.93 ± 0.09 | 52 ± 2 |

Table 12: [**Question answering**] **Model performance based on automatic metrics.** The results are derived from linearly regressing the model types against the continuous metrics using an ordinary least squares method, and the resulting values are the means of the metrics for each model type. Metrics are denoted by ‡ if models had a significant effect with a Bonferroni correction (see Appendix D.3 for details).

Given the likely bias in our survey sample, we decided to add robustness to the investigation and check our results while mathematically controlling for error by supplementing our analysis with univariate regressions. We ran linear regression models on all of the models against the different survey metrics using the stats package in R, which fits a linear regression against continuous variables using an ordinary least squares method R Core Team (2020). The numbers reported in the Table 11 —19 are the intercept plus the beta effect estimates associated with each variable.

Significance was assessed at the Bonferroni-corrected level Bland & Altman (1995). We checked the assumptions of the regressions held through manual verification of the residuals and Q-Q plots, and we compared the results of our experiments with the linear regression and validated relationships.

## D.2   Social Dialogue

The metrics analyzed for the dialogue task included *sensibility*, *specificity*, *humanness*, *interestingness*, *inclination*, and *reuse*. The significant results are in Table 11.

In the main body of the paper, it was reported that TextDavinci scored highest on *sensibleness*, *humanness*, *interestingness*, and *reuse*. TextDavinci tied with Davinci on *inclination*, and Davinci scored highest on *specificity*. The linear regression analysis confirms the results. For *inclination*, similar to the main paper, the rounded values are equivalent for TextDavinci and Davinci, and there is no significant measured difference between the two, although Davinci significantly outperforms Jumbo. This could reflect the performances in the previous metrics, and indicate the influence of *specificity* on *inclination*. Further research could be done by diving deeper into the nuances of the survey questions when evaluating dialogue, and investigating the relationships between the metrics. Additionally, this specific experiment could be well suited for a mixed effects analysis and could stand to benefit from further question refinement with the help of user feedback.

## D.3   Question Answering

The metrics analyzed for question answering are elapsed *time*, number of *queries*, and user correctness (*accuracy*). The results can be found in Table 12 and 13.

| Model | Accuracy (/100%) ↑ | |
| --- | --- | --- |
| | No LM | LM Used‡ |
| TextDavinci | 54 ± 2 ‡ | 62 ± 4 ‡ |
| TextBabbage | 47 ± 2 | 55 ± 4 |
| Davinci | 46 ± 2 ‡ | 54 ± 4 ‡ |
| Jumbo | 49 ± 2 | 57 ± 4 |

Table 13: [**Question answering**] **Model performance on accuracy, conditional upon language model use.** Metrics are denoted by ‡ if models had a significant effect with a Bonferroni correction (see Appendix D.3 for details). The ‡ next to the LM-Used column refers to the fact that LM use was associated with an increase in accuracy accross all models. However, there was also statistically significant higher accuracy in some groups over others, suggesting sample bias.

| Model | Fluency | Helpfulness | Ease | Enjoyment |
| --- | --- | --- | --- | --- |
| | | (/5) ↑ | | |
| TextDavinci | 3.65 ± .17 ‡ | 3.16 ± .17 ‡ | 4.35 ± .20 ‡ | 3.42± .20 ‡ |
| TextBabbage | 3.05 ± .17‡ | 2.35 ± .18‡ | 3.85 ± .21‡ | 2.76 ± .21‡ |
| Davinci | 2.24 ± .12 ‡ | 1.90 ± .12 ‡ | 3.26 ± .14‡ | 2.18 ± .14 ‡ |
| Jumbo | 2.34 ± .17 | 2.28 ± .18 | 3.11 ± .21 | 2.23 ± .20 |

Table 14: [**Crossword puzzle**] **Model performance based on user survey responses.** The results are derived from linearly regressing the model types against the metrics using an ordinary least squares method, and the resulting values are the means of the metrics for each model type. Metrics are denoted by ‡ if models had a significant effect relative to Davinci, at the Bonferroni-corrected significance level of $p = .0125$.

The results are derived from linearly regressing the model types against the continuous metrics using an ordinary least squares method, conditioning upon whether a linear model was used, and the resulting values are the means of the metrics for each model type. Overall, using AI assistance was significantly associated with higher accuracy for all of the models; similarly to the main results reported in the paper, TextDavinci performed the best. Additionally, like the main paper, participants using TextDavinci used the least *time*, and had the least *queries* suggesting ease of use. However, there were significant differences in accuracy amongst the study participants assigned to TextDavinci and Davinci, suggesting potential confounding factors and possible limitations to the results. Future research could control for potential confounding by using block randomization when assigning participants to different language models, and thus lend robustness to results.

### D.4  Crossword Puzzle

The metrics analyzed for the crossword puzzle task included *fluency*, *helpfulness*, *ease*, and *enjoyment*. Further analysis was conducted conditioning on the prompts used, to test for prompt-specific error and variability. The linear regression results are in Table 14. None of the prompts besides the arbitrarily chosen intercepts were deemed significant, and therefore the table is not included.

The model results reported in the main results section show TextDavinci outperforming the other models in all metrics. However, TextDavinci only showed *significant* improvement over TextBabbage for *helpfulness* and *enjoyment*, and there was no significant difference between the two models' performances on *fluency* and *ease*. The added linear regression analysis here demonstrates that, when controlling for error, TextDavinci has the best performance metrics to a significant degree above the other models. Additionally, although this analysis yields similar patterns of results for the worst performers, the overlapping confidence intervals and lack of significant differences between Jumbo and Davinci indicate that there is less clarity around the worst performer when using a technique that accounts for error and assumes normally distributed residuals. Upon verifying the linear regression assumptions, the residual and Q-Q plots indicate that there were certain

influential outliers that might be responsible for this lack of clarity. Further research could involve replicating this analysis in different survey samples, including regularization techniques in the statistical evaluation to account for overfitting, and investigating nonlinear relationships between the variables. These techniques might all yield clearer comparisons between the models.

### D.5 Text Summarization

| Model | Time (min) | Edit distance (word) ↓ |
|---|---|---|
| TextDavinci | $1.11 \pm .13$ | $12.38 \pm 1.25$ |
| TextBabbage | $1.22 \pm .13$ | $16.33 \pm 1.25$ |
| Davinci | $1.10 \pm .09$‡ | $14.18 \pm .88$‡ |
| Jumbo | $1.17 \pm .13$ | $15.12 \pm 1.25$ |

Table 15: [**Text summarization**] **Model performance based on automatic metrics.** The results were derived using the same ordinary least squares method and linear regression as described previously for the continuous variables (elapsed time, distance, improvement, edit). Model type was used as the predictor. Metrics are denoted by ‡ if models had a significant effect with a Bonferroni correction (see Appendix D.5 for details).

| Model | Improvement (/5) ↑ | Revision (/5) ↓ | Helpfulness (/5) ↑ |
|---|---|---|---|
| TextDavinci | $2.60 \pm .35$ | $3.00 \pm .27$ | $4.20 \pm .34$ |
| TextBabbage | $2.15 \pm .35$ | $3.40 \pm .27$ | $3.40 \pm .34$ |
| Davinci | $2.10 \pm .25$ ‡ | $3.25 \pm .19$‡ | $3.80 \pm .24$‡ |
| Jumbo | $2.40 \pm .35$ | $3.30 \pm .27$ | $3.30 \pm .34$ |

Table 16: [**Text summarization**] **Model performance based on user survey responses.**.

A linear regression analysis of the summarization task was used to calculate the means and significance for the variables (*elapsed time, edit distance, improvement, edit, helpfulness, original consistency, original coherency, original relevance, edited consistency, edited coherency, and edited relevance*). The results can be found in Table 15, Table 16, and Table 17. Model type was used as the predictor.

We checked the assumptions of the regressions by looking at the residuals and Q-Q plots. In the main paper TextDavinci was found to be the most helpful and improve the most with time, resulting in the lowest Levenshtein edit distance between the model produced and final summary, TextBabbage and Jumbo both scored low - TextBabbage requiring the most editing, and Jumbo producing the worst original summary and being the least helpful. The linear regression confirmed this and additionally highlighted the lack of distinction between the bad performers TextBabbage and Jumbo.

Third party evaluation indicated higher performance from TextBabbage than was found in the main paper, highlighting the importance of including annotation to verify the validity of results. Although the evidence reveals conclusively that users rate their summaries higher after editing, the evidence only slightly points to the superiority of TextDavinci, and the field could benefit from additional experiments in wider samples.

### D.6 Metaphor Generation

The metrics analyzed for metaphor generation included *elapsed time, queries, acceptance, edit distance, helpfulness, satisfaction, ease, reuse* and for third party evaluation *aptness, specificity, and imageability*. The significant results are in Table 18, Table 19 and Table 20. The direction of the results were comparable to the main paper, however no statistically significant relationships were found in the linear analysis besides the differences in acceptance and edit distance. In all other metrics, none of the models besides the arbitrarily

| Model | First-person evaluators | | | Third-party evaluators | | |
|---|---|---|---|---|---|---|
| | Consistency (/100%) ↑ | Relevance (/1) ↑ | Coherence | Consistency (/100%) ↑ | Relevance (/5) ↑ | Coherence |
| TextDavinci | 56 ± 5 | 0.63 ± .05 | 0.77 ± .04 | 65 ± 5 | 4.7 ± .07 ‡ | 4.07 ± .11‡ |
| TextBabbage | 75 ± 5 ‡ | .67 ± .05 | 0.7 ± .04 | 89 ± 5 ‡ | 4.51 ± .07 ‡ | 4.15 ± .11 |
| Davinci | 57 ± 3‡ | 0.65 ± .03‡ | 0.77 ± .03 ‡ | 57 ± 4‡ | 4.53 ± .05‡ | 3.7 ± .08 ‡ |
| Jumbo | 50 ± 5 | .61 ± .05 | 0.74 ± .04 | 56 ± 5 | 4.6 ± .07 | 3.8 ± .11 |

Table 17: [**Text summarization**] **Linear regression evaluating quality metric annotation based on first-person and third-party human evaluation.** Metrics are denoted by ‡ if models had a significant effect with a Bonferroni correction (see Appendix D.5 for details).

| Model | Time (min) ↓ | Queries (#) ↓ | Acceptance (/100%) ↑ | Edit distance (word) ↓ |
|---|---|---|---|---|
| TextDavinci | 0.74 ± 0.08 | 0.92 ± 0.12 | 50.79 ± 5.94 ‡ | 4.78 ± 0.96 |
| TextBabbage | 0.73 ± 0.07 | 0.97 ± 0.11 | 55.68 ± 5.51‡ | 6.42 ± 0.85 ‡ |
| Davinci | 0.6 ± 0.05 | 0.77 ± 0.09 | 71.48 ± 4.24 ‡ | 4.83 ± 0.64‡ |
| Jumbo | 0.75 ± 0.08 | 1.03 ± 0.13 | 68.29 ± 6.28 | 5.59 ± 0.92 |

Table 18: [**Metaphor generation**] **Model performance based on automated metrics.** The results are derived from linearly regressing the model types against the metrics using an ordinary least squares method, and the resulting values are the means of the metrics for each model type. Metrics are denoted by ‡ if models had a significant effect with a Bonferroni correction, $p = 0.0125$ (see Appendix D.6 for details).

| Model | Helpfulness | Satisfaction | Ease | Reuse |
|---|---|---|---|---|
| | | (/5) ↑ | | |
| TextDavinci | 4.21 ± 0.32 | 4.42 ± 0.27 | 3.68 ± 0.35 | 4.42 ± 0.26 |
| TextBabbage | 3.64 ± 0.3 | 4.14 ± 0.25 | 3.82 ± 0.32 | 4.39 ± 0.24 |
| Davinci | 4.17 ± 0.23‡ | 4.33 ± 0.19‡ | 3.94 ± 0.25 ‡ | 4.61 ± 0.19 ‡ |
| Jumbo | 4.13 ± 0.34 | 4.4 ± 0.29 | 3.87 ± 0.37 | 4.47 ± 0.28 |

Table 19: [**Metaphor generation**] **User survey responses.** The results are derived from linearly regressing the model types against the metrics using an ordinary least squares method, and the resulting values are the means of the metrics for each model type. Metrics are denoted by ‡ if models had a significant effect with a Bonferroni correction, $p = 0.0125$ (see Appendix D.6 for details). None of the above metrics were found to be significantly different from each other, except for the reference model.

| Model | Apt | Specific | Imageable |
|---|---|---|---|
| | | (/100%) ↑ | |
| TextDavinci | 77 ± 5 | 74 ± 5 | 73 ± 5 |
| TextBabbage | 80 ± 4 | 74 ± 5 | 72 ± 5 |
| Davinci | 83 ± 3 ‡ | 81 ± 4 ‡ | 74 ± 4‡ |
| Jumbo | 81 ± 5 | 76 ± 5 | 75 ± 6 |

Table 20: [**Metaphor generation**] **Third-party evaluation on the quality of metaphorical sentences.** The numbers indicate means and standard errors. The results are derived from linearly regressing the model types against the metrics using an ordinary least squares method, and the resulting values are the means of the metrics for each model type. Metrics are denoted by ‡ if models had a significant effect with a Bonferroni correction, $p = 0.0125$ (see Appendix D.6 for details).

chosen baseline models performed at a significantly different level from each other in the user survey. This indicates that the experiment found that the means of the models collectively were more than zero. However, the results do not conclusively reveal which ones performed better or worse (confirmed by the overlapping confidence intervals) on most metrics. This means that although the experiment found higher acceptance for `Davinci` suggestions and shorter edit distance, other conclusions are pending further investigation and experiment replication.

| | Targets | | Perspectives | | Criteria | Framework | |
|---|---|---|---|---|---|---|---|
| Metric | Target | Unit | Method | Evaluator | Criteria | Standard | HALIE (ours) |
| Social dialogue | | | | | | | |
| fluency | output | turn | survey | first-person | quality | N | Y |
| sensibleness | output | turn | survey | first-person | quality | N | Y |
| specificity | output | turn | survey | first-person | quality | N | Y |
| humanness | output | turn | survey | first-person | quality/preference | N | Y |
| interestingness | output | turn | survey | first-person | quality/preference | N | Y |
| inclination | output | turn | survey | first-person | preference | N | Y |
| reuse | process | dialogue | survey | first-person | preference | N | Y |
| Question answering | | | | | | | |
| accuracy | output | quiz | auto | third-party | quality | Y | Y |
| time | process | question | auto | third-party | quality | N | Y |
| queries | process | question | auto | third-party | quality | N | Y |
| queries | process | change | auto | third-party | quality | N | Y |
| prompt styles | process | change | auto | third-party | preference | N | Y |
| fluency | output | quiz | survey | first-person | quality | N | Y |
| ease | process | quiz | survey | first-person | quality | N | Y |
| helpfulness | process | quiz | survey | first-person | quality | N | Y |
| Crossword puzzles | | | | | | | |
| accuracy (letter) | output | puzzle | auto | third-party | quality | Y | Y |
| accuracy (clue) | output | puzzle | auto | third-party | quality | Y | Y |
| queries | process | puzzle | auto | third-party | preference | N | Y |
| prompt styles | process | change | auto | third-party | preference | N | Y |
| fluency | output | puzzle | survey | first-person | quality | N | Y |
| ease | process | puzzle | survey | first-person | quality | N | Y |
| helpfulness | process | puzzle | survey | first-person | quality | N | Y |
| enjoyment | process | puzzle | survey | first-person | preference | N | Y |
| Text summarization | | | | | | | |
| improvement | process | session | survey | first-person | quality | N | Y |
| helpfulness | process | session | survey | first-person | quality | N | Y |
| consistency (self) | output | summary | survey | first-person | quality | N | Y |
| relevance (self) | output | summary | survey | first-person | quality | N | Y |
| coherency (self) | output | summary | survey | first-person | quality | N | Y |
| consistency | output | summary | survey | third-party | quality | Y | Y |
| relevance | output | summary | survey | third-party | quality | Y | Y |
| coherency | output | summary | survey | third-party | quality | Y | Y |
| edit distance | output | summary | auto | third-party | quality | Y | Y |
| edit distance | process | change | auto | third-party | quality | N | Y |
| Metaphor generation | | | | | | | |
| queries | process | sentence | auto | third-party | quality | N | Y |
| acceptance | process | sentence | auto | third-party | quality | N | Y |
| edit | process | sentence | auto | third-party | quality | N | Y |
| time | process | sentence | auto | third-party | quality | N | Y |
| aptness | output | sentence | survey | third-party | quality | Y | Y |
| specificity | output | sentence | survey | third-party | quality | Y | Y |
| imageability | output | sentence | survey | third-party | quality | Y | Y |
| fluency | output | session | survey | first-person | quality | N | Y |
| satisfaction | output | session | survey | first-person | preference | N | Y |
| ease | process | session | survey | first-person | quality | N | Y |
| helpfulness | process | session | survey | first-person | quality | N | Y |
| enjoyment | process | session | survey | first-person | preference | N | Y |
| ownership | process | session | survey | first-person | preference | N | Y |
| reuse | process | session | survey | first-person | preference | N | Y |

Table 21: Full list of metrics and their mappings to the three dimensions.

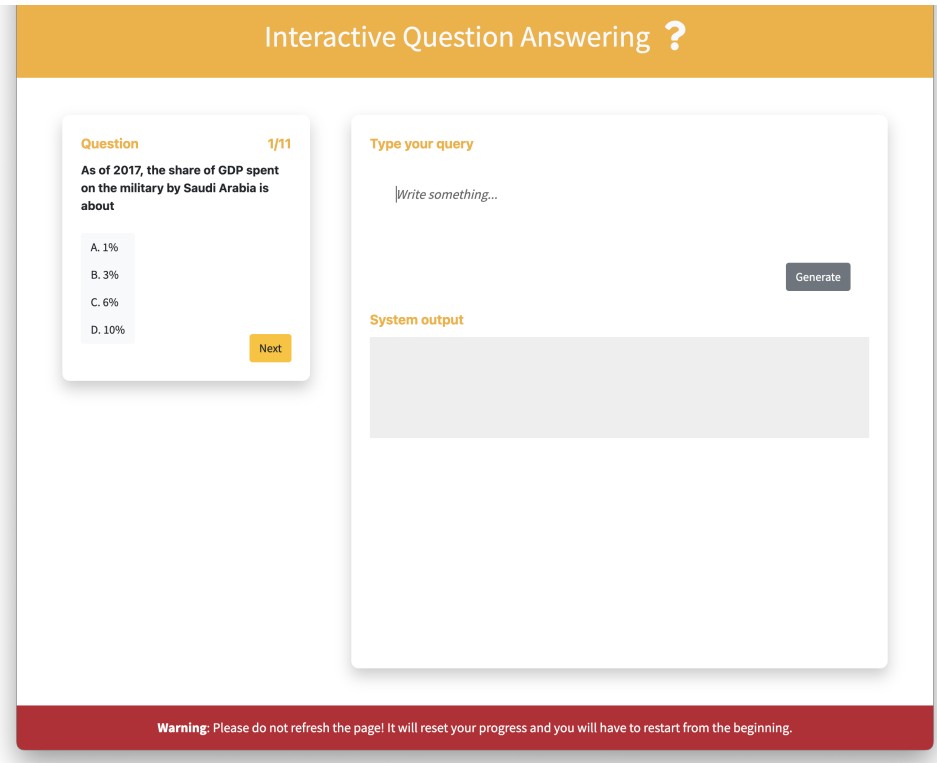

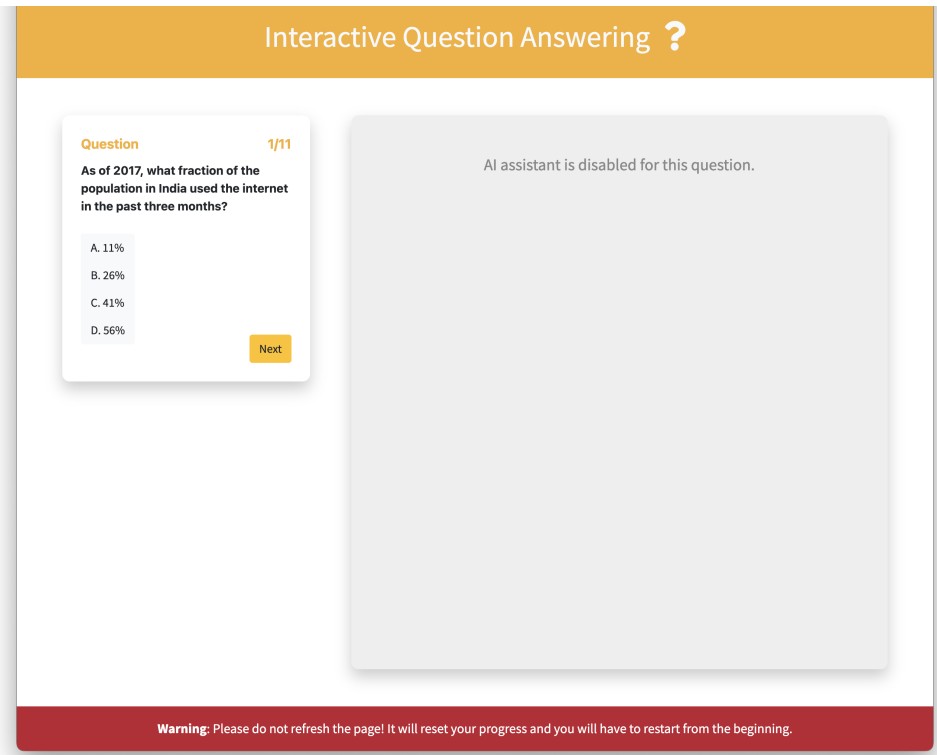

Figure 14: [**Question answering**] The interface enables AI assistance for half of the questions in a quiz (top) and disables it for the other half (bottom) in which case users have to answer the question without any assistance.

**How to interact with the AI teammate:** We encourage you to chat with an AI teammate, shown on the right side of the interface (see figure below).

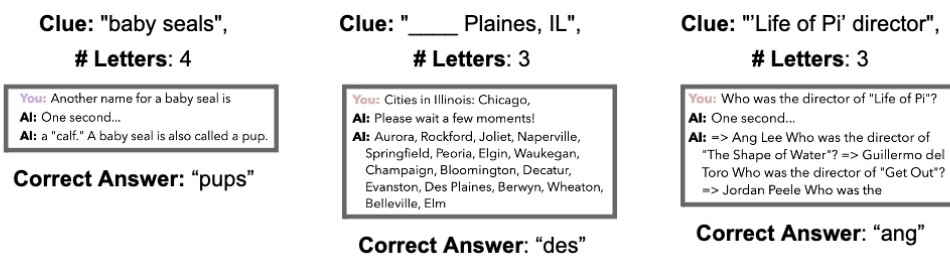

*After selecting a clue,* you can prompt the AI with text: for example a question or writing an unfinished sentence(s). The AI will attempt to respond with a fluent continuation of your prompt: e.g., an answer to your question, or a completion of your sentence.

**\*Note that the AI Teammate does not know the clue answers itself, but may have general knowledge that may be helpful\***

Below are some examples of ways users chose to interact with their AI teammate. However, you are free to interact with the AI in any way you find helpful, and we encourage you to be creative!

**Clue:** "baby seals",
**# Letters:** 4

> **You:** Another name for a baby seal is
> **AI:** One second...
> **AI:** a "calf." A baby seal is also called a pup.

**Correct Answer:** "pups"

**Clue:** "____ Plaines, IL",
**# Letters:** 3

> **You:** Cities in Illinois: Chicago,
> **AI:** Please wait a few moments!
> **AI:** Aurora, Rockford, Joliet, Naperville, Springfield, Peoria, Elgin, Waukegan, Champaign, Bloomington, Decatur, Evanston, Des Plaines, Berwyn, Wheaton, Belleville, Elm

**Correct Answer:** "des"

**Clue:** "'Life of Pi' director",
**# Letters:** 3

> **You:** Who was the director of "Life of Pi"?
> **AI:** One second...
> **AI:** => Ang Lee Who was the director of "The Shape of Water"? => Guillermo del Toro Who was the director of "Get Out"? => Jordan Peele Who was the

**Correct Answer:** "ang"

Figure 15: [**Crossword puzzles**] Instructions provided for users.

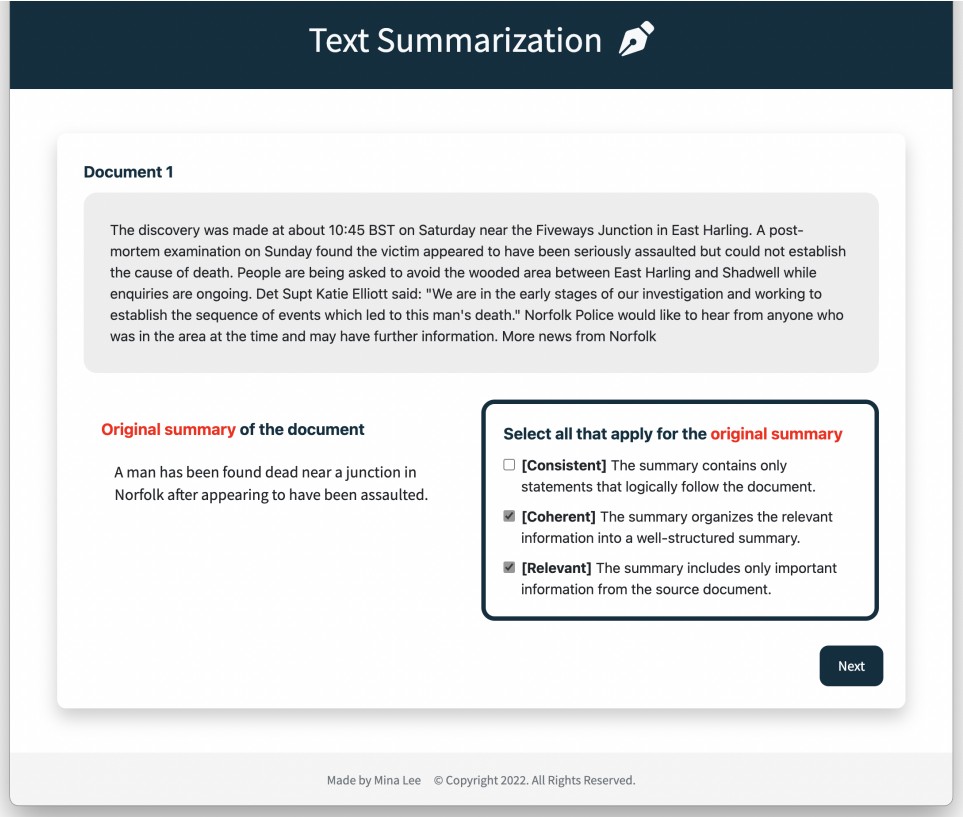

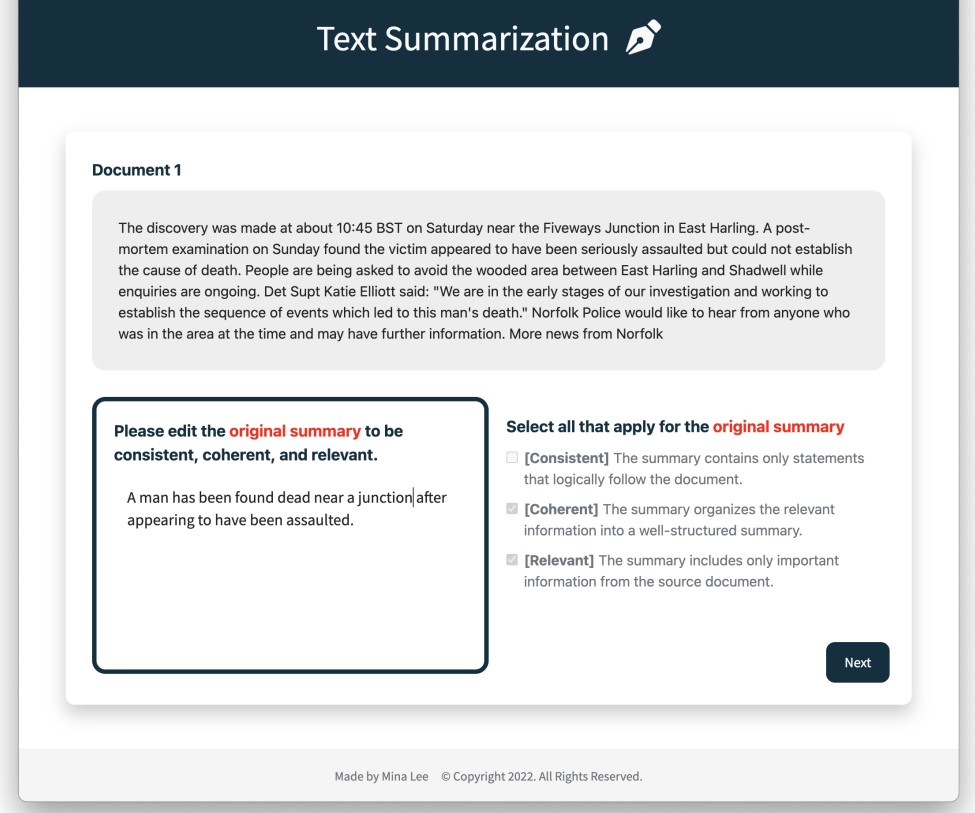

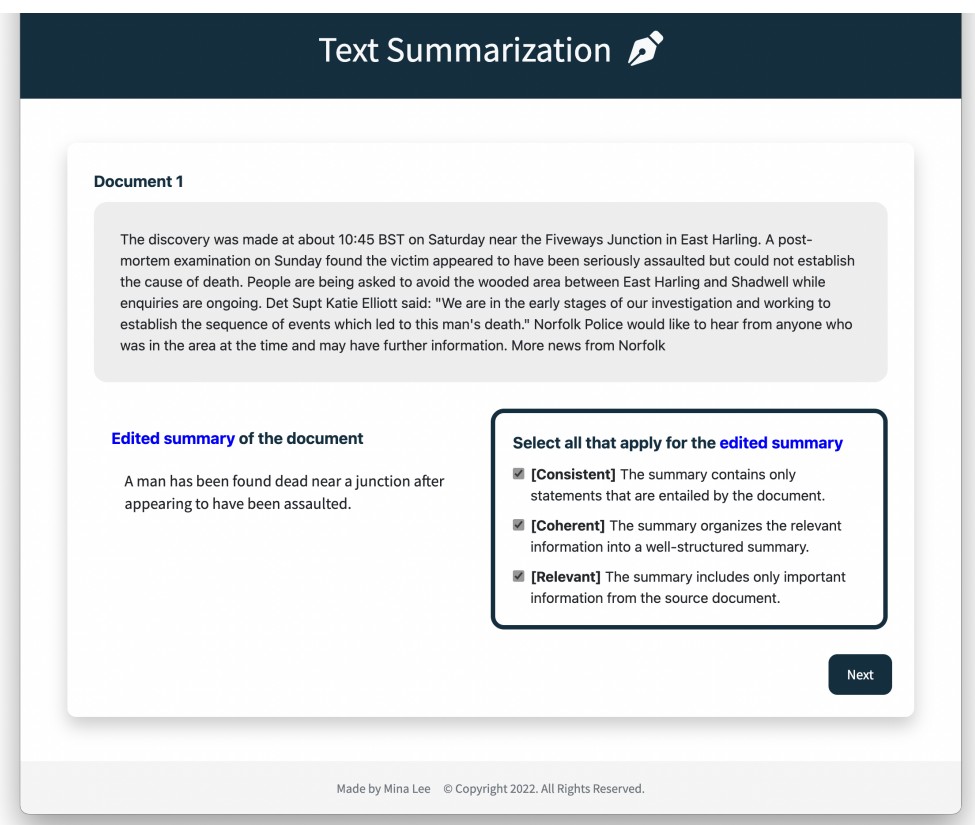

Figure 17: [**Text summarization**] We first ask users to rate the original summary generated by the system (top). Then, users edit the original summary to be more consistent, coherent, and relevant (middle). Finally, users rate their own edited summary (bottom).

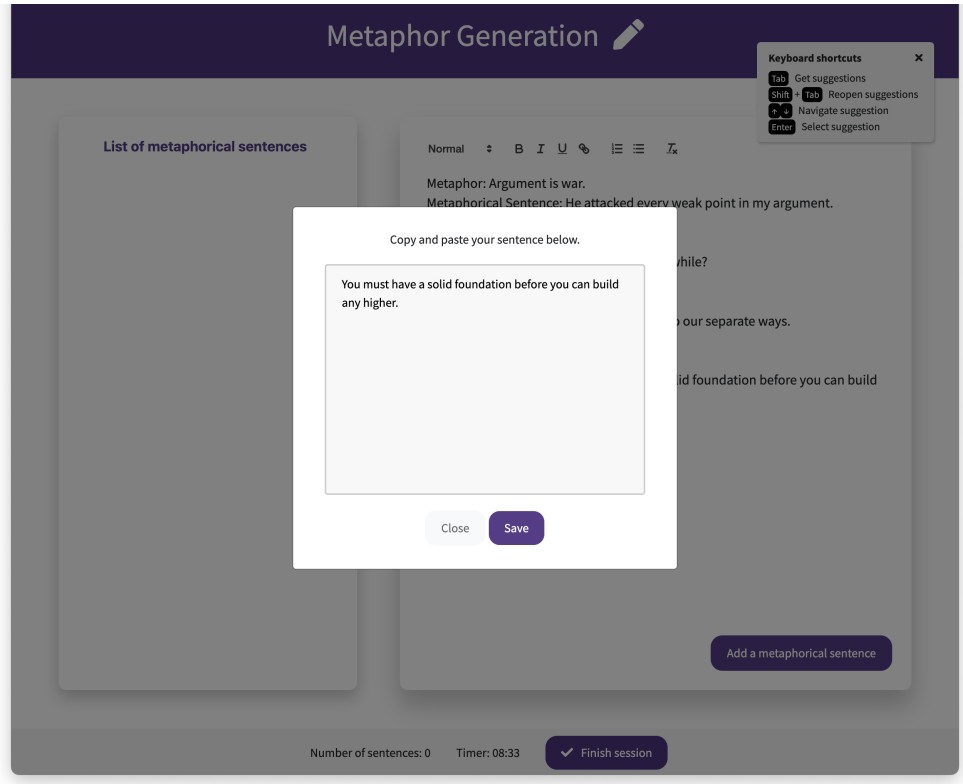

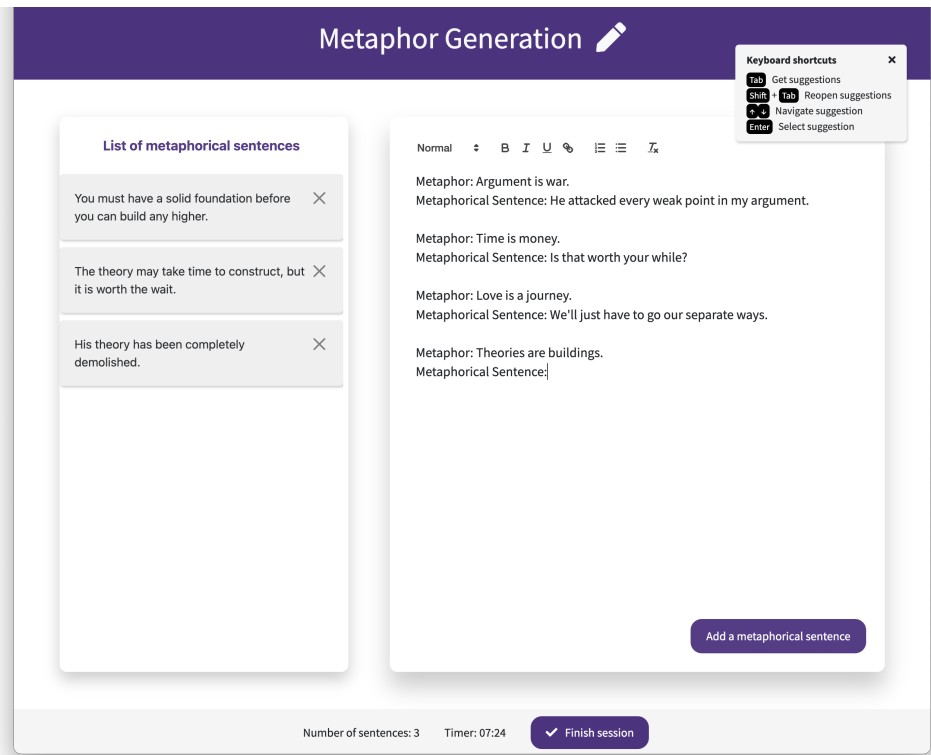

Figure 18: [**Metaphor generation**] Once users write a metaphorical sentence in the text editor, they can add it to the left-hand side of the interface (top) which keeps the list of metaphorical sentences written (bottom).

