# OpenReview forum: "Evaluating Human-Language Model Interaction"
_TMLR — Accepted by TMLR_

### Review · Reviewer_jbuE · 2023-07-31

**Summary Of Contributions:**

This paper presents a new framework, HALIE, to think about evaluation dimensions for settings that fundamentally involve human-LM interaction. It evaluates factors like the process itself rather than the output, the experience of the user (a "first-person" perspective), and aspects of the output beyond just quality (e.g., preference). Ultimately, the Cartesian product of these properties yields 8 categories of evaluation (e.g., "output first-person preference"), each of which is instantiated on at least one of the target tasks.

The paper specifically frames its approach as evaluating a system which sits *on top of* the language model and contains additional UI elements and UX design decisions. Relatively simple system logic is presented, largely focused around querying an LM, although there is flexibility in the implementation of the components of this process.

The paper evaluates five tasks designed to cover different forms of interaction. Several different LMs, including variants of GPT-3 and AI21's Jurassic-1 model, are evaluated. Detailed system descriptions and results are presented for each task.

Several research questions are explicitly asked and answered. First, do the "strongest" models work best in interactive settings?  The answer is: not always. Second, does perception of system utility match actual utility? That is, do people get better at tasks when they feel that LLM systems are helping them? Answer: not always (e.g., in crosswords, seemingly helpful feedback doesn't necessarily help accuracy that much). Finally, accommodation (users adapting to the affordances of the LLMs they're interacting with) is studied across two of the tasks in particular.

Finally, Section 5 discusses limitations of the study and highlights future work.

**Audience:**

Yes

**Claims And Evidence:**

Yes

**Requested Changes:**

I don't have any particular changes to request. Considering the weaknesses I listed above, I think there are a couple of minor framing things that can be improved for this paper. It would be nice to see a clearer claim about what this paper's contribution is. But I think as it is, the paper is quite well-written and will be valuable to the TMLR community.

**Strengths And Weaknesses:**

STRENGTHS

This paper is extremely timely. Systems like ChatGPT have simultaneously enabled a whole host of new applications for LLMs and also obviated many of the existing benchmarks.  Tasks like story generation fundamentally need to be evaluated with paradigms like this one. As a collection of case studies for how to do this across five different settings, this paper is quite valuable.

The general idea of the three axes and eight categories of evaluation presented is intuitively clear.

The breadth of the five tasks covered in this paper is impressive. I think seeing this range of applications allows the paper to give some authoritative takes on its research questions in ways that can't necessarily be addressed by more siloed evaluation.

The paper is very well-written. Each section is clearly structured and the takeaway points about the results make it easy to understand the high-level messages.

The accommodation studies (3.3-3.4, bullet point 4 in each) are quite interesting and I'm not personally aware of much other (public, academic) work studying these effects in LM usage.

The use of CreatePrompt specifically for summarization, where it takes the user's own edited summaries as training examples, is quite interesting. Again, I haven't seen many studies engaging with a workflow like this.

WEAKNESSES

I think the strengths for this paper generally outweigh the weaknesses. That said, I think there are a few weak points in the paper.

CLAIMS: TMLR's review criteria ask reviewers to evaluate the claims a paper is making and whether those claims are supported. For this paper, the main contribution seems to be a framework. However, the actual framework itself is pretty high-level and almost a kind of position statement. I'm not sure what the central claim is. Is it, "This is a good way to conceptualize human-LM evaluation"?  In that case, I'm not quite sure how to evaluate it with respect to alternatives.  The actual dimensions of the cube in Figure 1 don't seem like news to most researchers working on interactive systems.  And the real work seems to be figuring out how to instantiate this framework for each task of interest.

As a result, I felt that the whole was somewhat the sum of its parts here. There are a lot of good experiments conducted, and the conclusions are interesting, but they don't point in very particularly specific directions. The main exception to this is part of the Future Directions discussing accommodation, which is a very nice aspect of the study.

SET OF LANGUAGE MODELS: This paper is unfortunately a bit vulnerable to the temporal drift in the field over the last year. The language models very clearly reflect the state of things ~1 year ago rather than now.  The systems here differ in capabilities and scale, but I think having more systems at the top end (similar capabilities to TextDavinci) would probably give the most relevant specific conclusions in 2023.  I would imagine that in the July 2023 world of Llama 2 Chat, Claude 2, Bard, GPT-4, etc., there would be significantly more options for SOTA systems.  The validity of the research questions and the conclusions remains, but I would expect, for example, that the answer to RQ1 might be substantially more nuanced than what is found here.

As a result, the specifics of this paper are unfortunately probably not going to be that helpful to practitioners, compared to the high-level messages.

PERFORMANCE VS. INTERACTION: As a more minor quibble related to RQ1, I think the way of measuring non-interactive performance is slightly misleading.  For example, for the QA systems, it seems the users are using the LLM more as a retriever than as a QA system.  As a result, the divergence in the two green bars for Babbage in foreign policy is perhaps not that surprising: all of the systems do decently at fact retrieval even though they vary in terms of QA ability (full auto performance).

I think it's fair to see that as an indictment of optimizing for end-to-end leaderboard performance in the auto setting. But I think it's important to distinguish the difference in use cases here: it's more that the models are being used differently in the interactive setting, so the end-to-end performance is less relevant.  I felt this point could've been emphasized more.

(The summarization study is a bit more straightforward, but I would say serves more of an indictment of standard summary evaluation.  Revision distance probably *shouldn't* correlate because a small but insidious error is likely worse than a slightly off-topic but correct summary.)

---

> ### Author Response · Authors · 2023-08-13
> **Response to Review jbuE**
>
> We sincerely thank the reviewer for the insightful reviews and suggestions, especially given the length of our paper. Here, we list our responses to the weaknesses and the requested changes:
>
> ### Weakness 1 & Requested change 1: Clarity of the main contribution
>
> We appreciate the reviewer's constructive feedback regarding the clarity of our main contribution. We believe our main contribution is to look at disparate tasks and bring them under one roof to deepen our understanding of human-LM interaction, in line with the reviewer’s comment (“I think seeing this range of applications allows the paper to give some authoritative takes on its research questions in ways that can't necessarily be addressed by more siloed evaluation”). For example, dialogue evaluation clearly has been interactive and well-studied, but we generalize beyond dialogue and consider some novel and unconventional tasks (e.g., crossword puzzles and metaphor generation); it is because we believe that studying the full spectrum of tasks (from creative to factual) can help us better understand human-LM interaction in general.
>
> To improve upon this point, we will revise the introduction and outline the following key components of our contribution in our paper.
>
> * We propose a framework to evaluate human-LM interaction, which consists of three components: an interactive task, an interactive system, and a set of dimensions for metrics. By delineating the components, we provide a structured way to explore the multidimensional space of human-LM interaction.
> * We instantiate our framework in the five tasks. These instantiations showcase the adaptability and versatility of our approach across various scenarios.
> * The results obtained through the framework challenge conventional evaluation metrics and highlight the need for tailored assessment methodologies in the context of human-LM interaction.
>
> > I'm not sure what the central claim is. Is it, "This is a good way to conceptualize human-LM evaluation"? In that case, I'm not quite sure how to evaluate it with respect to alternatives.
>
> To directly answer the last question, we admit that "this is a good way to conceptualize, structure, and execute human-LM evaluation" is the main message we strive to convey throughout the paper, and we substantiate the claim through five case studies. In terms of evaluating it against other alternatives, it is unclear what other alternatives we should compare to, as this is the first work to formalize human-LM evaluation to our knowledge. On the other hand, some of the other alternatives we considered during the formulation of our framework include interactability (Hornbæk and Oulasvirta (2017)), cognitive dimensions (Green and Peter (1996); Green (2000)), facet theory (Guttman (1959)), and design space (Card et al. (1990), MacLean et al. (1991)), which we would be happy to discuss if desired.
>
> We note that our approach has elements reminiscent of those of a design space. By delineating interactive tasks, interactive systems, and dimensions for metrics, we provide a structured way to explore the multidimensional space of human-LM interaction. Each instantiation within our framework represents a specific point in this design space, and the design space offers researchers a practical means to navigate, comprehend, and compare the various possibilities of human-LM interaction scenarios.
>
> > The actual dimensions of the cube in Figure 1 don't seem like news to most researchers working on interactive systems.
>
> We agree with the reviewer and emphasize the fact that we do not claim novelty for each dimension. In fact, the dimensions have been partially and/or extensively studied in subareas within NLP (e.g., dialogue systems). However, to the best of our knowledge, the dimensions have not been presented altogether as orthogonal dimensions and for various tasks as a way to facilitate a holistic design of evaluation metrics for interactive systems. In case we have overlooked, we would be happy to add pointers to prior work.
>
> > And the real work seems to be figuring out how to instantiate this framework for each task of interest.
>
> We hope that our framework is less of a burden, but more of a useful resource for researchers to refer to when designing the evaluation of human-LM interaction for a new scenario. In particular, we believe that the framework helps researchers consider various elements and dimensions of human-LM interaction that are easy to overlook (e.g., design considerations associated with CreatePrompt, QueryLM, and ShowCompletions in the transition function, as described in Section 2.2). In line with the response to the reviewer ouU5’s Requested change #1, we will add general guidelines for applying the framework to a new scenario.

---

> ### Author Response · Authors · 2023-08-13
> **Response to Review jbuE (continue)**
>
> **References**
> * Hornbæk and Oulasvirta (2017). What Is Interaction?
> * Green and Peter (1996). Usability Analysis of Visual Programming Environments: A 'Cognitive Dimensions' Framework
> * Green (2000). Instructions and Descriptions: Some Cognitive Aspects of Programming and Similar Activities
> * Guttman (1959). Introduction to Facet Design and Analysis
> * Card et al. (1990). The Design Space of Input Devices
> * MacLean et al. (1991). Questions, Options, and Criteria: Elements of Design Space Analysis
>
> ### Weakness 2: Choice of LMs
>
> > This paper is unfortunately a bit vulnerable to the temporal drift in the field over the last year. [...] The validity of the research questions and the conclusions remains, but I would expect, for example, that the answer to RQ1 might be substantially more nuanced than what is found here. [...] As a result, the specifics of this paper are unfortunately probably not going to be that helpful to practitioners, compared to the high-level messages.
>
> As the reviewer pointed out, we believe the validity of our research questions and conclusions remain, and the significance of our work lies in its ability to offer a conceptual framework and insights that extend beyond the specific models under consideration.
>
> That being said, the choice of LMs in our experiments is certainly susceptible to rapid technological advancements. Since we conducted our experiments, there have been many new models that are notably much more capable and safer.
>
> In light of these newer models, we acknowledge the reviewer's observation regarding the nuanced nature of our findings in response to RQ1. Particularly, we observe that chatbot interfaces have grown significantly in popularity since our study. This dominance has prompted adaptations that prioritize interactive aspects, resulting in a more aligned optimization goal than before. Consequently, we speculate that the gap between non-interactive and interactive performance might have been partially bridged due to these adaptations.
>
> In short, we recognize the value of investigating the implications of these newer models on our research questions, which we will discuss in the paper. We appreciate the reviewer's insights in helping us contextualize our work within this ever-changing landscape.
>
> ### Weakness 3: Difference in use cases in interactive vs. non-interactive evaluation
>
> > I think it's important to distinguish the difference in use cases here: it's more that the models are being used differently in the interactive setting, so the end-to-end performance is less relevant.
>
> This is a great point. We will revise our paper to explicitly mention this difference.

---

### Review · Reviewer_mQ4i · 2023-08-04

**Summary Of Contributions:**

## Summary

- The paper proposes a new framework called HALIE (Human-AI Language-based Interaction Evaluation) for evaluating human-language model interaction (Section 2).

- HALIE expands on non-interactive evaluation of language models along three dimensions (Section 2.3):

    - Targets: Evaluates the full interaction process, not just the final output (Section 2.3)

    - Perspectives: Captures first-person user experience, not just third-party assessment (Section 2.3)

    - Criteria: Includes preference metrics like enjoyment, not just quality metrics like accuracy (Section 2.3)

- The authors design and build interactive systems for five diverse tasks (Section 3): social dialogue (Section 3.2), question answering (Section 3.3), crossword puzzles (Section 3.4), text summarization (Section 3.5), and metaphor generation (Section 3.6).

- They evaluate four language models (GPT-3 TextDavinci, TextBabbage, Davinci, and Jurassic-1 Jumbo) on these tasks through **user studies** on Amazon Mechanical Turk (Section 3).  The user survey covers three dimensions mentioned above (Tab. 1)

- Social dialogue (Section 3.2):

    - Evaluated chatbots for open-ended dialogues about social scenarios

    - Key findings:

        - Instruction tuning improves performance on most quality metrics, but not specificity (Section 3.2)

        - Users may prefer interacting with more specific LMs (Section 3.2)

- Question answering (Section 3.3):

    - Evaluated LMs as interactive assistants for multiple choice QA

    - Key findings:

        - Non-interactive performance doesn't always lead to better human-LM interaction (Section 3.3)

        - Instruction tuning improves human-LM interaction (Section 3.3)

        - Users accommodate behaviors over time (Section 3.3)

- Crossword puzzles (Section 3.4):

    - Evaluated LMs as assistants for solving crossword puzzles

    - Key findings:

        - User-perceived helpfulness exceeds accuracy improvements (Section 3.4)

        - Users demonstrate diverse engagement behavior (Section 3.4)

        - Users accommodate behaviors over time (Section 3.4)

- Text summarization (Section 3.5):

    - Evaluated LMs for iteratively improving summaries

    - Key findings:

        - Discrepancy between third-party and first-person metrics (Section 3.5)

        - Some LMs learn better from user examples (Section 3.5)

- Metaphor generation (Section 3.6):

    - Evaluated LMs for suggesting metaphors

    - Key findings:

        - User effort doesn't always correlate with output quality (Section 3.6)

        - User satisfaction may not indicate output quality (Section 3.6)

- Main contributions:

    - Proposes the HALIE framework that expands the scope of LM evaluation (Section 2).

    - Highlights gaps between non-interactive and interactive evaluations (Section 3.1).

    - Provides insights into designing tasks and metrics to assess human-LM interaction (Section 5).

- Insights on evaluation (Section 5):

    - General challenges in human evaluation (Section 5.1)  :

        - Cost of human evaluation is high

        - Individual differences lead to subjectivity and lack of reproducibility

    - Challenges specific to interactive evaluation (Section 5.2):

        - Need to account for individual differences and sequential effects in study design

        - Model latency impacts user experience

        - Mitigating potential harms from unsafe model responses

    - Limitations of authors' experiments (Section 5.3):

        - Tasks selected may not generalize to other domains

        - User interface design can affect human-LM interaction

        - Prompting strategies impact model performance

        - Crowdsourced users may not represent real users

    - Future work:

        - Study longer-term interactions and user accommodation over time (Section 5.4)

        - Investigate potential lasting impacts of human-LM interaction (Section 5.4)

**Audience:**

Yes

**Broader Impact Concerns:**

1. The experiments in this paper that involve human subjects obtained IRB approval.
2. The holistic evaluation approach proposed in this paper could be beneficial for mitigating ethical concerns surrounding the deployment of large language models.


**Claims And Evidence:**

Yes

**Requested Changes:**

There are several ways to improve this paper:

### Collect general domain tasks and design more general metrics
This paper focuses on 5 specific tasks for evaluating human-LM interaction. The authors could expand the scope by collecting more diverse and general domain tasks that better represent real-world applications of LMs.
They could then try to design evaluation metrics that work across multiple tasks, rather than task-specific metrics. Developing a standard set of metrics for human-LM interaction could make the framework more broadly applicable.
### Add automatic evaluation to reduce cost and increase availability
The user studies and surveys employed in this paper are costly and time-consuming. The authors could explore supplementing the human evaluation with automatic metrics (e.g. using LLMs to self- or cross-evaluate) that approximate user judgments.
This could help scale up evaluation and make it more accessible to future researchers. Of course, care would need to be taken to ensure the automatic metrics reliably reflect subjective user experiences. But adding automatic evaluation alongside human assessment could combine the benefits of both approaches.

**Strengths And Weaknesses:**

This is a *very long* paper (29 pages in the main text) with rich content. For every dimensions, there are significant benefits and risks. It is not easy to form two separate lists of strengths and weaknesses as reviewers usually do. Here I compose a few dimensions along which we should gauge this paper.

1. **The overall idea of HALIE** The key benefit of HALIE seems to be providing a more *comprehensive picture* of LM capabilities by focusing on real-world human-LM interaction. However, it also comes with challenges around cost (rely on user survey, not trivial to automate), reproducibility, and safeguards that need to be addressed. Overall, HALIE offers a promising direction to complement existing methods, but risks would need to be mitigated for responsible application.

2. **The design of the 5 tasks** The main issues seem to be around the limited scope/scale and subjectivity inherent in the tasks. The scenarios are contrived, documents simplified, and metaphors few. Individual differences in skills and preferences also make the evaluations more variable. The studies offer useful insights but may lack generalization. One way to improve is to conduct a general domain task through collecting tasks and goals LM users have in the real life. That being said, the tasks do touch all three dimensions mentioned before as shown in Tab. 1.

3. **Insights and oversights** For a benchmark paper, one of the most important contributions should be insights for future models. This paper presents insights around specificity, user effort/engagement, accommodation, and mitigating unsafe responses. These insights would be very useful for improving future chatbots to be more engaging partners. A problem with framing this paper as evaluating *"human-LM interaction** is that this paper studies specific sets of metrics for each task, which needs extra effort to curate for new tasks.

---

> ### Author Response · Authors · 2023-08-13
> **Response to Review mQ4i**
>
> We sincerely thank the reviewer for the insightful reviews and suggestions, especially given the length of our paper. Here, we list our responses to the weaknesses and requested changes:
>
> ### Weakness 1: Cost, reproducibility, and safeguard of human evaluation
>
> > However, it also comes with challenges around cost (rely on user survey, not trivial to automate), reproducibility, and safeguards that need to be addressed. [...] risks would need to be mitigated for responsible application.
>
> We appreciate the reviewer’s balanced view on human evaluation and acknowledge its potential limitations and challenges regarding cost, reproducibility, and safeguards. Below, we provide the details on how we tried to address them in HALIE, but admit that these are not extensive measures and there is room for improvement.
>
> **Cost.** While we acknowledge that cost is often a bottleneck for conducting experiments involving users, we believe it is a necessary step towards understanding the real-world implications of LMs given that a substantial number of individuals engage with LMs in their daily lives, as suggested by statistics (e.g., Duarte, 2023). As the reviewer suggested, devising cheaper surrogates (potentially automatic metrics) can be one viable, cost-effective way to extend this work. However, we emphasize that we need to do the evaluation with human users at least once to learn what the ground truth is and see which automatic metrics correlate with the real user evaluation. With good surrogates, we agree that one may not need to run the full human-LM interactive evaluation on every single model.
>
> **Reproducibility.** According to Belz et al. (2023), many papers lack fundamental details such as the number of participants, details of training, instructions, and set of model outputs, severely hindering their reproducibility. We have detailed the methodology, data collection processes, and experimental design in our paper, and released all interaction traces (the link was redacted for anonymity), which contain all model outputs along with events triggered by users. This information is aimed at facilitating other researchers in replicating our study or building upon our findings; the data can be further used to do additional derived analyses as well. Another challenge in the reproducibility of human evaluation is due to the subjectivity of tasks and/or users, which we respond to under Weakness #2.
>
> **Safeguards.** The reviewer aptly points out the necessity of safeguards for deploying LMs in real-world applications in a responsible manner. In our case, where the primary goal is to evaluate LMs, striking the right balance between comprehensive evaluation and stringent safeguards poses a unique challenge. While HALIE includes a mild measure (i.e., blocklist) to mitigate potential risks, we acknowledge that a more extensive approach could be pursued in future work, and should be pursued for productionized applications. Given the evolving nature of both NLP research and societal considerations, continued exploration of enhanced safeguards should be a collective effort within the community.
>
> **References**
> * Duarte (2023). [Number of ChatGPT Users (2023)](https://explodingtopics.com/blog/chatgpt-users)
> * Belz et al. (2023). Missing Information, Unresponsive Authors, Experimental Flaws: The Impossibility of Assessing the Reproducibility of Previous Human Evaluations in NLP
>
> ### Weakness 2: Subjectivity and ecological validity of the five tasks
>
> > The main issues seem to be around the limited scope/scale and subjectivity inherent in the tasks. The scenarios are contrived, documents simplified, and metaphors few.
>
> Broadly speaking, there are two evaluation approaches: field evaluations and lab evaluations (McGrath, 1995), each with its own set of trade-offs. Field evaluations encompass broader scope and ecological validity, albeit often presenting challenges in scalability and precise quantification of relevant factors. On the other hand, lab evaluations, as demonstrated in our work, may possess lower ecological validity but offer more precise estimations on a larger scale.

---

> ### Author Response · Authors · 2023-08-13
> **Response to Review mQ4i (continue)**
>
> Our approach was to strike the balance between incorporating the elements that exist in real-world applications/usage and conducting experiments around them that can facilitate relatively large-scale user studies. The current set of five tasks (social dialogue, question answering, crossword puzzles, text summarization, and metaphor generation) was chosen based on the common query types from the OpenAI report (Table 1 in Ouyang et al., 2022), which provided valuable insights on how real users query LMs. We acknowledge that these query types are abstractions on actual tasks (e.g., “brainstorming” as opposed to “come up with 10 different ideas for …”), which we then concretized into actual tasks (e.g., “metaphor generation” for “brainstorming”). In doing so, our goal was to capture a diverse set of interactions that users have with LMs while making our task as concrete as possible. Therefore, while they may not fully generalize, we believe they do unearth a range of conditions seen in the real interactive use of LMs.
>
> It holds true that certain tasks, such as metaphor generation, exhibit a higher degree of subjectivity compared to others like question answering. This deliberate choice was guided by our aim to encompass both open-ended, less constrained tasks, and goal-oriented, more constrained tasks (Gero et al., 2022). Regarding subjectivity in human evaluation, we contend that all human evaluation possesses an element of subjectivity to some degree. To ensure the robustness of our findings, we recruited a large number of users whenever possible and employed statistical analyses to quantify and analyze their responses. While it is undeniable that subjectivity remains a challenge in assessing certain tasks, the inclusion of a diverse pool of evaluators helps mitigate bias and provides a more comprehensive perspective. (Appendix D).
>
> > Individual differences in skills and preferences also make the evaluations more variable.
>
> We acknowledge that individual differences can make evaluation more variable. One way to mitigate this variability is to perform within-subject studies. However, to compare four models, each participant would need to repeat an experiment four times, which significantly increases the workload. For instance, in text summarization, each experiment (or “HIT” in Amazon Mechanical Turk) asks a participant to evaluate and edit a model summary 10 times; with four models, it will be 40 times (instead of 10), taking 2 hours (instead of 30 minutes) in total. Therefore, to streamline our experiments, we made use of between-subject studies as a practical choice and recruited a large number of users to reduce the effect from individual differences whenever possible.
>
> We note that while generalization is a customary desideratum for ML and NLP evaluations, it is also important to consider and be able to distinguish individual users (e.g., many works in HCI deploy relatively small-scale semi-structured interviews for nuanced and contextualized understanding). In this context, variability is not necessarily undesirable; people have different experiences and these are the real effects we want to capture for understanding the impact of LMs on various users. HALIE captures these individual differences in the form of interaction traces.
>
> **References**
> * McGrath (1995). Methodology Matters: Doing Research in the Behavioral and Social Sciences
> * Ouyang et al. (2022). Training Language Models to Follow Instructions with Human Feedback
> * Gero et al. (2022). A Design Space for Writing Support Tools Using a Cognitive Process Model of Writing
>
> ### Weakness 3 & Requested change 1: General domain tasks and metrics
>
> We appreciate this suggestion and acknowledge its importance in clarifying the motivation behind our choice of the tasks.
>
> > One way to improve is to conduct a general domain task through collecting tasks and goals LM users have in the real life.
>
> We are cautious in terms of collecting tasks and goals directly from individuals (especially when the number of individuals we can interview is small), because what people profess to be their goals may not be predictive of their actual goals, as documented thoroughly in the field of psychology (e.g., stated vs. revealed preferences). Furthermore, user goals often require a nuanced understanding of the user as well as the context in a specific time and place, which poses a further challenge in incorporating them into a benchmark. Therefore, we instead referred to the real user usage of OpenAI models (Ouyang et al., 2022) and designed our five tasks based on them.

---

> ### Author Response · Authors · 2023-08-13
> **Response to Review mQ4i (continue)**
>
> > Collect general domain tasks and design more general metrics
>
> We interpreted "general domain tasks" as tasks that encompass a wide range of everyday activities, interactions, and problem-solving scenarios that users might engage in while interacting with LMs. These tasks could span various domains, such as education, entertainment, communication, and more.
>
> With this interpretation, we believe that our tasks contain general elements that are reflective of these characteristics. See below for the distribution of use case categories from OpenAI’s API prompt dataset (Ouyang et al., 2022) and our tasks associated with each category.
>
> Generation (45.6%) - Question answering, Metaphor generation
> Open QA (12.4%) - Question answering, Crossword puzzles
> Brainstorming (11.2%) - Crossword puzzles, Metaphor generation
> Chat (8.4%) - Social dialogue, Crossword puzzle
> Rewrite (6.6%)
> Summarization (4.2%) - Text summarization
> Classification (3.5%)
> Other (3.5%)
> Closed QA (2.6%) - Question answering
> Extract (1.9%)
>
> We agree that by selecting specific instances for each task (e.g., questions from MMLU for question answering) and building an interface for the task, we make the task environment less “general”, but we argue that this is necessary to define an evaluation goal and conduct large-scale user studies. If we are misinterpreting the reviewer’s use of the term “general domain task”, we would appreciate clarification and examples for general domain tasks and would be happy to follow up on the discussion.
>
> Regarding designing metrics that work across multiple tasks, we note that there are already metrics shared across multiple tasks in the paper. For example, the number of queries and elapsed time are used for question answering, crossword puzzles, and metaphor generation, and reuse and/or helpfulness are asked via user surveys for all tasks. Although providing generic metrics for human-LM interaction could be beneficial for other researchers, we think it is unlikely that a fixed set of metrics are going to be applicable for all tasks. This is because many aspects of user experience are fundamentally dependent on both a specific task at hand as well as a specific user interface used to support the task, making any metric hard to be one-size-fits-all and context-agnostic. Therefore, we support the design of metrics that reflect the realities of conducting evaluation in specific contexts of interest, rather than promoting the use of generic metrics.
>
> > A problem with framing this paper as evaluating "human-LM interaction* is that this paper studies specific sets of metrics for each task, which needs extra effort to curate for new tasks.
>
> We recognize the value of exploring a broader array of tasks and metrics to better represent real-world user interactions and the need for guidelines in applying the framework to them. In line with the response to the reviewer ouU5’s Requested change #1, we will add a section discussing how future research can apply the framework to new scenarios.
>
> ### Requested change #2: Automatic evaluation
>
> > Add automatic evaluation to reduce cost and increase availability
>
> While exploring alternatives such as automatic metrics or using LMs as proxies is a valid and promising avenue for future research (as we acknowledge in our response to Weakness 1), these possibilities are currently beyond the scope of this work. As our work aims to set the foundations for human-LM interaction evaluation, we strive to study and establish the ideal setting, leaving room for future works to strike more pragmatic compromises of evaluation complexity/cost and fidelity/quality.

---

### Review · Reviewer_ouU5 · 2023-08-05

**Summary Of Contributions:**

This work aims to tackle the limitations of non-interactive evaluation on some real-world applications of large language models. The proposed framework, HALIE, covers three dimensions: targets, perspectives, and criteria. For each dimension, the evaluation is extended to evaluate one aspect of the human-language model interaction. For example, the perspective dimension aims to capture the first-person experience, instead of a third-party evaluation. The proposed evaluation framework is built on interactive state tracking. The experiments analyze the results from five tasks with four models.

**Audience:**

Yes

**Claims And Evidence:**

Yes

**Requested Changes:**

I think the previously mentioned weaknesses are resolvable, therefore the requested changes are
- provide a discussion on some general guidelines for designing HALIE for a new task
- provide a brief discussion between the system logic and other interaction procedure, for example, the one used in RL

**Strengths And Weaknesses:**


**Strengths**

- This work presents a comprehensive evaluation framework for human-language model interaction evaluation.
- For each selected task, this paper presents sufficient details on task description, system logic, user study procedure, and survey questions. Besides, the result analysis provides insights for readers to understand the language models, and the limitations of existing evaluation strategies.
- The discussion section, particularly section 5.3 (limitations specific to our experiment design) and section 5.4 (future work) hopefully will stimulate more research on the direction of human-language model interaction
- Last, the writing is clear — the paper is easy to read and follow.

**Weaknesses**

- Although this paper has demonstrated the application of the proposed framework on five different tasks, it is still unclear to me how to design the evaluation if I have a new task. In other words, what are the general guidelines for realizing the framework for a new task? Can any future task be considered similar to one of these five examples? I strongly recommend addressing these questions in the next version of this paper.
- This is more like a minor issue — the description of system logic in section 2.2 kept reminding me about the interaction procedure employed in reinforcement learning. Of course, the description in section 2.2 does not concerns with making decisions or predicting actions, but I am wondering whether some brief discussion can be offered to differentiate it from RL.

---

> ### Author Response · Authors · 2023-08-13
> **Response to Review ouU5**
>
> We sincerely thank the reviewer for the insightful reviews and suggestions, especially given the length of our paper. Here, we list our responses to the weaknesses and requested changes:
>
> ### Weakness 1 & Requested change 1: Guidelines for the framework
>
> > What are the general guidelines for realizing the framework for a new task? Can any future task be considered similar to one of these five examples?
>
> We agree with the reviewer that it would be beneficial for readers to have guidelines for applying the framework to a new (interactive) task. Below, we provide the guideline for (i) determining whether the framework is applicable as well as (ii) applying the framework to a new task with interactive story writing as a running example.
>
> **Guideline for determining whether the framework is applicable**
> * HALIE is designed to support evaluation of human-LM interaction. Therefore, we assume the presence of a human user, an LM, interaction between the user and LM, and an evaluation goal. For instance, a scenario where an LM is evaluated by another LM (as a proxy for human judgment) does not have human users, thus considered out of scope for the framework. Similarly, a scenario where there is a user, but the user has no way to interact with models (e.g., a user merely reads model outputs and evaluates them, as opposed to influencing model generation by providing inputs or modifying model outputs), we consider it as human evaluation, rather than human-LM interaction, and therefore consider it out of scope.
> * HALIE is designed for scenarios where a single user interacts with an LM. Therefore, when there are multiple users and/or LMs, it may require extending (but not replacing) HALIE.
> * We speculate that the use of HALIE is particularly helpful for tightly-scoped tasks with specific goals, where specifying the states, actions, and transition functions can be very informative for readers to understand the tasks, systems, and expected interactions between a user and LM.
>
> **Guideline for applying the framework to a new task**
>
> To instantiate the HALIE framework, the researchers need to specify an interactive task, an interactive system, and a set of metrics. To be concrete, we will use the task of interactive story writing in Clark et al. (2018) as a running example and describe how one might apply HALIE to describe and evaluate the task. Please note that this is a hypothetical scenario and we highlight parts of the work (but not all) for brevity.
>
> **Defining an interactive task.** To define an *interactive* task (as opposed to a *non-interactive* task), it is essential to communicate the nature of the interaction between a user and an LM that is being studied. This involves detailing the space of possible user input and model output, as well as any intermediate steps or back-and-forth exchanges that constitute the task.
>
> For interactive story generation, let’s assume the researchers decide to consider a setting where an LM writes every other sentence in a story. Concretely, they define the task to be writing a ten sentence story by writing sentence by sentence linearly, while every other sentence is first generated by an LM and optionally edited by the user. This turn-taking continues until a story reaches ten sentences. In this task, the space for user input and model output is restricted to an English sentence. The intermediate steps are defined as strict turn-taking. It’s worth noting that there are no back-and-forth exchanges in this task, as editing sentences is not allowed in this task once each sentence is submitted by the users.
>
> If a task is relatively new and it is unknown how users might interact with LMs, we recommend running small pilots to gain insights into user goals and preferences before defining a task. In Clark et al. (2018), the researchers decided to restrict the input and output space to be sentence-level based on the preliminary study.
>
> **Constructing an interactive system.** Once a task is defined, the researchers need to define states, actions, and transition functions for the task and build an interactive system based on them. By specifying these components, the researchers can easily communicate how their interactive systems operate not just on the surface (i.e., frontend), but also behind the scenes (i.e., backend, where, even with demos, it is often hard to communicate this aspect without a formal specification).

---

> ### Author Response · Authors · 2023-08-13
> **Response to Review ouU5 (continue)**
>
> In the case of interactive story generation, suppose the researchers decide to show sentences from previous turns in separate boxes (story history) and provide an input box for a user to write and edit a sentence (user input) in the user interface. These elements become states of the system (i.e., {story history, user input}). Then the researchers design available actions, such as clicking the “Add Line to Story” button and the “Submit Story” button. With the states and actions, they define transition functions, which specify how each (state, action) pair connects to the next state (e.g., when a user clicks the “Add Line to Story” button, the system adds the current sentence to the story, updating the story history, and empties the user input box for the next sentence).
>
> **Designing evaluation metrics.** When designing metrics, we encourage researchers to first consider all combinations of the three dimensions presented in the framework (evaluation targets, perspectives, and criteria), select dimensions and design metrics that are relevant to the task, and then detail how each metric will be measured in their systems while taking into account the unique characteristics of the task and the corresponding user interactions.
>
> For interactive story writing, the researchers come up with an idea to measure the usefulness of a model output, when thinking about a metric for process (target), first-person (perspective), and both quality and preference (criteria). Concretely, they decide to measure the usefulness by asking users while conducting open-ended interviews (an alternative for crowdsourcing can be to have a Likert scale next to each turn for a model output). To account for the unique characteristics of the task, they decide to see the relationship between the usefulness of a model output and the location of the sentence within a story where the model output is presented (e.g., in the 2nd turn vs. 10th turn).
>
> We will update the paper with these guidelines.
>
> **References**
> * Clark et al. (2018). Creative Writing with a Machine in the Loop: Case Studies on Slogans and Stories
>
> ### Weakness 2 & Requested change 2: Comparison between system logic and RL
>
> > The description of system logic in section 2.2 kept reminding me about the interaction procedure employed in reinforcement learning. Of course, the description in section 2.2 does not concerns with making decisions or predicting actions, but I am wondering whether some brief discussion can be offered to differentiate it from RL.
>
> This is a great point. Indeed, our system logic framework was inspired by Markov decision processes: we have states, actions, and traces (episodes). The main difference to standard RL is that we are not attempting to optimize any policies (i.e., the way a user interacts with an LM); rather we are evaluating naturally occurring patterns in these policies (i.e., the way a user interacts with and accommodates to an LM over time). Also, in standard RL, there is a single reward function that matters. In our framework, we evaluate a policy in a multi-dimensional way accounting for different evaluation targets, perspectives, and criteria. We will add this discussion to the paper for clarity.

---

### Author Response · Authors · 2023-08-13
**Author Response to All Reviews**

We thank all the reviewers for their valuable feedback on our paper.

The reviewers commended the importance of interactive evaluation (ouU5, mQ4i, jbuE), the thoroughness and detailed descriptions of our experiments (ouU5, mQ4i, jbuE), insightful and interesting results and analyses (ouU5, mQ4i, jbuE), and clear writing of the paper (ouU5, jbuE).

On the other hand, the reviewers pointed out challenges in instantiating the framework for a new task (ouU5, jbuE), challenges with regard to cost, reproducibility, and safeguards (mQri), the subjectivity and ecological validity of the five tasks (mQri), and need for clarifying the main contribution (jbuE).

In response to the raised concerns and requested changes, we have provided clarifications and perspectives in the comment section for each reviewer. We will update a revised version of the paper based on the suggestions before the end of the rebuttal period (August 18) so that the reviewers and AE can confirm that the requested changes and suggestions are addressed in the paper.

---

> ### Author Response · Authors · 2023-08-18
> **Revision**
>
> We uploaded a revised version of the paper and described the changes in "Changes Since Last Submission." Again, we thank all reviewers for their valuable suggestions!

---

### Decision · Action_Editors · 2023-09-06

**Recommendation:** Accept as is

**Comment:**

Reviewers unanimously feel their concerns are addressed by the revision.

The paper is well written, well received, and timely contribution to a burgeoning field.

**Audience:**

Broadly of interest to the LLM community, but specifically contexts in which automated agents are being used alongside or in lieu of humans.  The set of people who will find this work relevant will grow quickly over the next few years.

**Claims And Evidence:**

The paper introduces a framework for evaluating interactions between a human and language model.  The framework is then demonstrated on a diverse set of tasks (Dialogue, QA, Summarization, ...), from different perspectives, and across several strong models. Each task has to define criteria of interest (e.g. Helpfulness, Fluency, Knowledge, ...)  which can be measured and compared quantitatively.  The instances of these tasks are often constrained in their demonstration here, but the results for the use cases still prove interesting and the paper's aim is a framework to aid in the creation of such evaluations -- which it delivers.

The goals of the paper are thus achieved, and the main concern (how to extend the framework to novel tasks) has been addressed in the revision.